# Single-cell multiomics reveals ENL mutation perturbs kidney developmental trajectory by rewiring gene regulatory landscape

Lele Song[1,2,13], Qinglan Li[1,2,13], Lingbo Xia[1,2,3], Arushi Eesha Sahay [1,2], Qi Qiu [4,5], Yuanyuan Li[6,7], Haitao Li [6,7], Kotaro Sasaki[8,9,10], Katalin Susztak [4,11,12], Hao Wu[4,5] & Liling Wan [1,2,5,9]

How disruptions to normal cell differentiation link to tumorigenesis remains incompletely understood. Wilms tumor, an embryonal tumor associated with disrupted organogenesis, often harbors mutations in epigenetic regulators, but their role in kidney development remains unexplored. Here, we show at single-cell resolution that a Wilms tumor-associated mutation in the histone acetylation reader ENL disrupts kidney differentiation in mice by rewiring the gene regulatory landscape. Mutant ENL promotes nephron progenitor commitment while restricting their differentiation by dysregulating transcription factors such as *Hox* clusters. It also induces abnormal progenitors that lose kidney-associated chromatin identity. Furthermore, mutant ENL alters the transcriptome and chromatin accessibility of stromal progenitors, resulting in hyperactivation of Wnt signaling. The impacts of mutant ENL on both nephron and stroma lineages lead to profound kidney developmental defects and postnatal mortality in mice. Notably, a small molecule inhibiting mutant ENL's histone acetylation binding activity largely reverses these defects. This study provides insights into how mutations in epigenetic regulators disrupt kidney development and suggests a potential therapeutic approach.

Normal cellular differentiation is a tightly regulated process that relies on finely tuned epigenetic and transcriptional control[1–3]. Germline or somatic mutations affecting epigenetic/transcriptional regulators can disrupt these mechanisms, leading to developmental disorders and malignancies[2,4,5]. This phenomenon is particularly evident in pediatric cancers[6–8], where the roots are suspected to be intertwined with aberrant developmental trajectories. However, in most tissue types, the dynamic chromatin and gene regulatory events that orchestrate cell fate determination and how alterations in these events impact normal development and drive the diseased state remain unclear. Unraveling such insights has the potential to identify vulnerable cell types or developmental

[1]Department of Cancer Biology, University of Pennsylvania, Philadelphia, PA 19104, USA. [2]Abramson Family Cancer Research Institute, Perelman School of Medicine, University of Pennsylvania, Philadelphia, PA 19104, USA. [3]Department of the School of Engineering and Applied Science, University of Pennsylvania, Philadelphia, PA 19104, USA. [4]Department of Genetics, University of Pennsylvania, Philadelphia, PA 19104, USA. [5]Penn Epigenetics Institute, University of Pennsylvania, Philadelphia, PA 19104, USA. [6]MOE Key Laboratory of Protein Sciences, Beijing Frontier Research Center for Biological Structure, School of Medicine, Tsinghua University, Beijing 100084, China. [7]Tsinghua-Peking Center for Life Sciences, Beijing 100084, China. [8]Department of Biomedical Sciences, University of Pennsylvania, School of Veterinary Medicine, Philadelphia, PA 19104, USA. [9]Institute for Regenerative Medicine, Perelman School of Medicine, University of Pennsylvania, Philadelphia, PA 19104, USA. [10]Department of Pathology and Laboratory Medicine, University of Pennsylvania, Perelman School of Medicine, Philadelphia, PA 19104, USA. [11]Renal, Electrolyte, and Hypertension Division, Department of Medicine, University of Pennsylvania, Perelman School of Medicine, Philadelphia, PA 19104, USA. [12]Institute for Diabetes, Obesity, and Metabolism, University of Pennsylvania, Perelman School of Medicine, Philadelphia, PA, USA. [13]These authors contributed equally: Lele Song, Qinglan Li. ✉e-mail: Liling.Wan@Pennmedicine.upenn.edu

states as the disease origins and pave the way for precision medicine.

Wilms tumor, the most common pediatric kidney tumor[9,10], is linked to disrupted embryonic kidney development[10–12]. As such, it serves as a paradigm for understanding the interplay between development and tumorigenesis. The mammalian kidney emerges from the intermediate mesoderm through reciprocal interactions between two tissues, the ureteric bud (UB) and the metanephric mesenchyme[13–15]. At the initiation of nephrogenesis, signals from the mesenchyme induce reiterative branching of the UB. UB-derived signals in turn induce a subset of nephron progenitor cells (NPCs) located in the cap mesenchyme (CM) surrounding the UB tip to commit and undergo mesenchymal-to-epithelial transition (MET). This transition gives rise to an intermediate condensed structure known as the peritubular aggregate (PA), which then progresses into an epithelial structure termed the renal vesicle (RV). Subsequent segmentation and elongation of the renal vesicle give rise to a variety of epithelial nephron structures, including glomerular podocytes, proximal tubules (PT), loops of Henle (LOH), and distal tubules (DT), while the UB becomes the collecting duct[13–15] (Fig. 1a). The metanephric mesenchyme also gives rise to the renal stroma[16], which plays an important role in proper differentiation of the UB and nephron[17–19].

Histologically, Wilms tumor closely resembles the embryonic kidney and is often marked by rudimentary structures[12]. Known molecular drivers of Wilms tumor commonly involve disruptions to key transcription factors (TFs) (e.g., *WT1*, *SIX2*) and signaling proteins (e.g., *IGF2*, *WTX*, β-catenin) crucial for nephrogenesis[5,20]. To date, Wilms tumor studies have primarily focused on phenotypic characterization of mouse models for a few established molecular players[21–23], leaving the underlying cellular and molecular mechanism incompletely understood. In addition, the precise causes of two-thirds of Wilms tumor remain unclear. Recent genomic characterization of high-risk Wilms tumor has revealed previously unidentified mutations in epigenetic regulators in 30-50% of cases[20], underscoring unexplored roles of epigenetics dysregulation in this disease. Furthermore, while recent single-cell profiling studies on both normal kidneys and Wilms tumors[24–28] have offered valuable insights into kidney development and support a fetal origin for Wilms tumor, significant gaps remain in our understanding of how perturbations in chromatin mechanisms by disease mutations impact cell fate determination during kidney development and pathogenesis, particularly at single-cell resolution.

Our current study aims to address these fundamental questions by focusing on the epigenetic reader protein eleven-nineteen-leukemia (ENL). ENL, also known as MLLT1, exerts its function by binding to histone acylation through its conserved YEATS (Yaf9, ENL, AF9, Taf14, Sas5) domain and recruiting elongation factors to promote transcription[29,30]. ENL plays a crucial role in maintaining subsets of acute myeloid leukemia (AML)[29,30]. Our group has recently developed a small-molecule inhibitor designed to target ENL's acyl-binding activity, which has shown promising efficacy against AML in animal models[31]. More recently, a series of hotspot mutations within ENL's YEATS domain have been identified in 5–9% of Wilms tumor patients[32], making ENL the most frequently mutated epigenetic regulator in this disease[20,32]. Wilms tumors harboring *ENL* mutations often display intralobular nephrogenic rests, which stem from early kidney development and are associated with a high risk[20,32]. Previous studies in the human embryonic kidney cell line HEK293 have revealed that these mutations confer gain-of-function properties to ENL, enabling it to drive aberrant transcription activation through the formation of condensates at specific target gene loci[33,34]. Moreover, introduction of these *ENL* mutations into mouse embryonic stem cells has led to the formation of Wilms tumor-like blastema structures in an in vitro-directed differentiation assay[33], suggesting their potential biological significance. However, the precise functions of *ENL* mutations on

kidney development and tumorigenesis in vivo, as well as the underlying mechanisms, have remained unknown.

Here, by integrating genetic mouse modeling, histological characterizations, and single-cell transcriptomics and chromatin accessibility profiling, we reveal ENL mutation-induced alterations in cellular composition, differentiation trajectories, and gene regulatory landscapes during the development of the mouse kidney. These cellular and molecular alterations result in impaired nephrogenesis and postnatal mortality in mice. Furthermore, we demonstrate that transient inhibition of the acyl-binding activity of mutant ENL can effectively abolish its chromatin function in cellular systems and more importantly, rescue transcriptomic and developmental defects induced by the mutant in embryonic kidneys in vivo. This study provides functional and mechanistic insights into the impact of Wilms tumor-associated *ENL* mutations on kidney development and offers a proof-of-concept for the use of epigenetics-targeted agents in the correction of developmental defects.

## Results
### Mutant ENL disrupts embryonic kidney development and leads to postnatal mortality in mice

To investigate the role of ENL mutants in nephrogenesis, we generated a conditional knock-in mouse model for the most prevalent *ENL* mutation found in Wilms tumor[32] (p.117_118insNHL, referred to as ENL-T1) using an inversion strategy (Supplementary Fig. 1a). Before induction of Cre recombinase activity, the targeted allele is expressed as *Enl*-WT (Supplementary Fig. 1a). After two steps of Cre-mediated recombination, the inverted exon 4 containing the T1 mutation is flipped to the correct direction and expressed, and WT exon 4 is excised, thus leading to the expression of the targeted allele as *Enl*-T1 (Supplementary Fig. 1a). Genotyping PCR was used to successfully distinguish the wildtype and the targeted allele before Cre-mediated recombination (Supplementary Fig. 1b). To induce *Enl*-T1 expression in the developing kidney, we crossed *Enl*[flox-T1/+] mice with *Wt1*[GFPCre/+] mice (Fig. 1b) to generate *Enl*[flox-T1/+]*Wt1*[GFPCre/+] (hereafter referred to *Enl*-T1) and *Enl*[+/+]*Wt1*[GFPCre/+] (hereafter referred to *Enl*-WT) offspring. *Wt1* (Wilms tumor 1) is a transcription factor expressed within the intermediate mesoderm[35,36], the origin of the metanephric kidney[12]. The *Wt1*[GFPCre] knock-in allele in mice expresses an EGFPCre fusion protein from the *Wt1* promoter/enhancer elements and concomitantly inactivates the endogenous *Wt1* gene[37]. Given the well-characterized expression pattern and function of *Wt1* in kidney development, the *Wt1*[GFPCre] strain is widely used for genetic studies in kidney biology and diseases[21]. To validate *Enl*-T1 expression following Cre-mediated recombination, mRNA was extracted from E15.5 *Enl*-WT and *Enl*-T1 kidneys for reverse transcription, PCR amplification, and next-generation sequencing to determine the relative abundance of *Enl*-WT and *Enl*-T1 mRNA (Supplementary Fig. 1c). As expected, only *Enl*-WT mRNA was present in *Enl*-WT kidneys. In *Enl*-T1 kidneys, 58% and 41% of the sequencing reads corresponded to *Enl*-WT and *Enl*-T1 cDNA, respectively (Supplementary Fig. 1d). These results confirm the successful induction of *Enl*-T1 and demonstrate comparable expression levels of the *Enl*-WT and *Enl*-T1 alleles in *Enl*[flox-T1/+]/*Wt1*[GFPCre/+] kidneys.

While *Enl*-T1 and *Enl*-WT pups derived from the breeding scheme (Fig. 1b) were born at a mendelian ratio, all *Enl*-T1 newborns exhibited early postnatal mortality (Supplementary Fig. 1e). Further examination of embryonic kidneys harvested at embryonic day 15.5 and 18.5 (E15.5 and E18.5) revealed markedly reduced size of *Enl*-T1 kidneys compared to their WT counterparts (Fig. 1c). Histologically, *Enl*-WT kidney harbored expected structures characteristic of the developing kidney including ureteric buds (UB) invading the cap mesenchyme (CM), intermediate differentiating nephron structures (comma/S-shape bodies), and fully differentiated nephron structures (tubules and glomeruli) (Fig. 1d, f). These results, aligning with earlier findings[38], confirm that the loss of one allele of *Wt1* has minimal impacts on

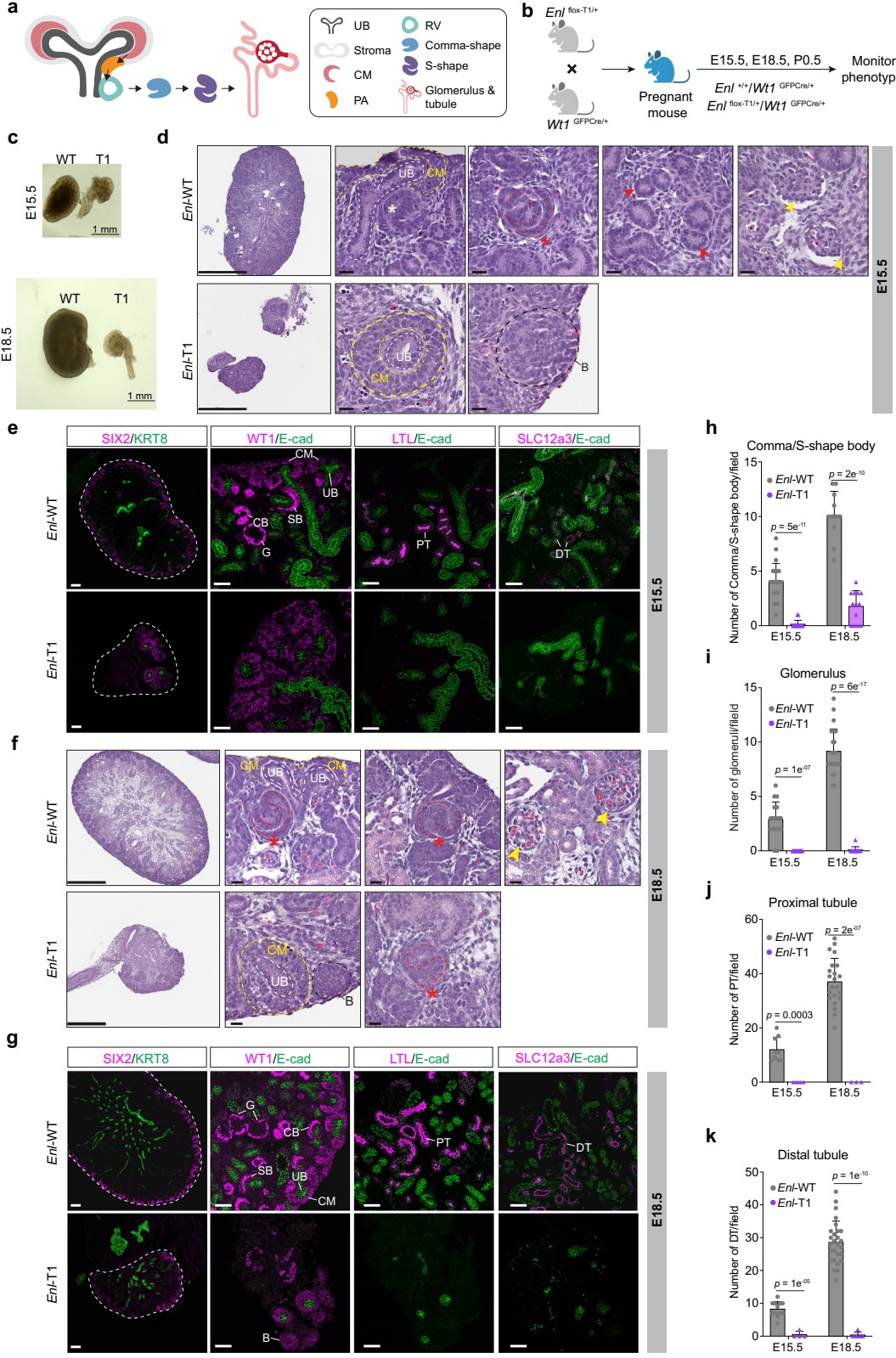

embryonic kidney development, supporting these mice as suitable controls for our study. In *Enl*-T1 kidneys, although the CM and UB structures were discernible, the CM was characterized as multilayers of cells enveloping a non-branching UB (Fig. 1d, f). In addition, some *Enl*-T1 kidneys exhibited proliferative structures with elevated Vimentin expression that morphologically resembled undifferentiated blastema components observed in Wilms tumors[12] (Fig. 1d, f, Supplementary

Fig. 1f). The CM abnormality in *Enl*-T1 kidneys was further validated through immunostaining for SIX2 (a marker for NPCs)[39], KRT8 (a marker for the ureter epithelium)[40], WT1 (expressed in NPCs and podocytes)[35], and E-cadherin (epithelial cells)[41] (Fig. 1e, g). Moreover, *Enl*-T1 kidneys displayed fewer differentiating/differentiated structures such as the comma/S-shape bodies (Fig. 1d–h), glomeruli (Fig. 1d–g, i), and elongating tubules including the PT (LTL⁺) and DT

**Fig. 1 | Heterozygous expression of mutant ENL disrupts embryonic kidney development and leads to postnatal mortality in mice. a** Schematic of nephrogenesis. Cap mesenchyme (CM), ureteric bud (UB), peritubular aggregate (PA), renal vesical (RV). **b** Schematic of the mouse breeding and experimental strategy. **c** Brightfield images of the E15.5 (top) and E18.5 (bottom) kidneys. Scale bar, 1 mm. **d, f** Hematoxylin and eosin-stained sections showing the histology of E15.5 (**d**) and E18.5 (**f**) kidneys from *Enl*-WT and *Enl*-T1 embryos. The red star indicates S-shape body, the red arrows indicate tubules, the black dashed circle indicates a region of blastema-like structure (B), and the yellow arrows indicate glomeruli structures. Scale bar in the first column images, 250 μm (**d**) and 500 μm (**f**); scale bar in the zoom-in images, 20 μm. **e, g** Immunostaining for SIX2, KRT8, WT1, E-cadherin (E-cad), LTL, and SLC12a3 at E15.5 (**e**) and E18.5 (**g**) kidney sections. Scale bar in the first column images, 150 μm (**e**) and 100 μm (**g**); scale bar in the zoom-in images, 50 μm. SB, S-shape body; CB, Comma-shape body; G, glomerulus; PT, proximal tubule; DT, distal tubule. **h–k** Quantification for the numbers of nephron structures per field. **h** Comma/S-shape body (E15.5, *n* = 9 WT and 10 T1 kidneys; E18.5, *n* = 6 WT and 6 T1 kidneys); **i** glomerulus (E15.5, *n* = 7 WT and 9 T1 kidneys; E18.5, *n* = 9 WT and 7 T1 kidneys); **j** proximal tubule (E15.5, *n* = 7 WT and 4 T1 kidneys; E18.5, *n* = 7 WT and 3 T1 kidneys); **k** distal tubule (E15.5, *n* = 7 WT and 4 T1 kidneys; E18.5, *n* = 6 WT and 5 T1 kidneys). Dots represent the number of indicated structures per field. Date represent mean ± s.d.; two-tailed unpaired Student's *t*-test. Source data are provided as a Source Data file.

(SLC12A3⁺) (Fig. 1d–g, j, k). These defects in the developing kidney of *Enl*-T1 mice persisted postnatally (Supplementary Fig. 1g–l), which likely explains the observed postnatal mortality (Supplementary Fig. 1e). Thus, heterozygous expression of *Enl*-T1 initiated in *Wt1*⁺ metanephric mesenchyme precursors substantially disrupts kidney development primarily through a gain of aberrant CM/UB structures and a reduction in differentiated nephron structures. This perturbation ultimately results in kidney agenesis and postnatal lethality in mice.

Given that *Wt1*$^{GFPCre}$ induces *Enl*-T1 expression in both nephron and stromal progenitors[13,42], we next asked whether *Enl*-T1 expression in both lineages contributes to the developmental defects observed in *Enl*$^{flox-T1/+}$*Wt1*$^{GFPCre/+}$ mice. To this end, we crossed *Enl*$^{flox-T1/+}$ mice with *Six2*$^{GFPCre/+}$ mice to induce *Enl*-T1 expression specifically in *Six2*⁺ NPCs and their progeny[39] (Supplementary Fig. 2a). Similar to findings with the *Wt1*$^{GFPCre/+}$ strain, *Enl*-T1 expression caused early postnatal lethality in the *Six2*$^{GFPCre/+}$ strain (Supplementary Fig. 2b). However, the phenotypic manifestations of *Enl*-T1 expression differed between the two Cre strains. Specifically, *Enl*-T1 expression in *Six2*⁺ NPCs had minimal impact on the overall kidney size and tubular structures, but significantly affected the development and maturation of glomeruli in embryonic (E15.5 and E18.5) (Supplementary Fig. 2c–f) and neonatal (P0.5) (Supplementary Fig. 2g, h) kidneys, as indicated by reduced numbers as well as shrunken and fragmented glomeruli. These results suggest that expression of *Enl*-T1 in the *Wt1*$^{GFPCre/+}$ model likely impairs kidney development through its effects in both nephron and stroma compartments.

## Altered cellular composition and differentiation trajectories in *Enl*-mutant kidneys

To dissect the mechanism by which mutant ENL perturbs embryonic kidney development, we isolated and sequenced a total of 10, 000 cells from whole kidney suspensions derived from 2 *Enl*-WT and 4 *Enl*-T1 E15.5 embryos. Following stringent filtration, 9376 individual cells (5232 for WT and 4144 for T1) were retained for further analysis (Supplementary Fig. 3a–c and Supplementary Data 1). We first focused on developing a single-cell transcriptomics map of *Enl*-WT kidneys. Unbiased clustering coupled with the expression patterns of lineage-specific marker genes identified four major lineage compartments: nephron (*Six2*⁺ and *Cited1*⁺), stroma (*Pdgfra*⁺ and *Col1a1*⁺), UB/ureteric epithelium (UE) (*Ret*⁺ and *Calb1*⁺), and endothelial cells (*Pecam1*⁺ and *Cdh5*⁺) (Supplementary Fig. 3d, e)[43,44]. Within the nephron, cells were further grouped into 10 clusters (Supplementary Fig. 3f) representing various cell types, including nephron progenitors (C0/1/4: *Six2*⁺), distal tubule precursor (C6-DT pre: *Gata3*⁺, *Sim1*⁺), distal tubule (C8-DT: *Gata3*⁺), LOH precursor (C5-LOH pre: *Sim1*⁺), LOH (C9-LOH: *Slc12a1*⁺), proximal tubule (C7-PT: *Slc34a1*⁺), and podocyte (C3-podo: *Nphs1*⁺, *Mafb*⁺) (Supplementary Fig. 3g). Within nephron progenitors, we identified two major cell states: Cluster 0 represents self-renewing NPCs with high expression levels of *Six2* and *Cited1* (C0-NP1: *Six2*⁺/*Cited1*⁺), while clusters 1/4 represent NPCs primed for differentiation (C1/4-NP2: *Six2*$^{low}$/*Wnt4*⁺) that are

often found within the pretubular aggregate (Supplementary Fig. 3g). In addition, we identified a cluster representing an intermediate cell state (C2-IM: *Cdh6*⁺, *Six2*), which loosely corresponds to cells in the renal vesicle (RV), the precursor of nephron[15]. Developmental trajectory analysis of *Enl*-WT nephrons revealed three distinct differentiation paths originating from NPCs, leading to the formation of podocytes, PT, and DT/LOH, respectively (Supplementary Fig. 3h). These results align well with prior characterizations and single cell analyses on normal mouse kidneys[26,28,43,45]. Thus, despite the heterozygosity of *Wt1* in *Enl*-WT kidneys, their cellular composition and differentiation pattern closely resemble those of a typical, healthy kidney.

To assess the consequences of the *Enl* mutation, we analyzed the integrated scRNA-seq datasets from *Enl*-WT and *Enl*-T1 kidneys. Using single-cell transcriptomes of *Enl*-WT cells as a reference map for cell type annotations, we showed that *Enl*-T1 cells were similarly grouped into 4 lineage compartments (Fig. 2a). A quantitative analysis of the cellular distribution among these compartments revealed a relative decrease in the nephron and an increase in the stroma in *Enl*-T1 kidneys (Fig. 2b). These observations, coupled with the fact that *Wt1*-cre expression is restricted to the nephron and stroma (Supplementary Fig. 3i, j), prompted us to focus our subsequent analyses on the role of mutant ENL in these two lineages.

In the integrated nephron, we identified 12 distinct cell clusters that correspond to 9 different cell types (Fig. 2c). There was a noticeable depletion of differentiated cell types, such as PT (C8), LOH (C11), DT (C9), and podocytes (C6) in *Enl*-T1 nephrons compared with wildtype. Conversely, undifferentiated cell types, such as NP2 (C1/5/7), displayed a slight enrichment in *Enl*-T1 nephrons (Fig. 2c–f). Furthermore, two clusters, namely C4 and C10 (hereafter referred to as T1-abnormal, or T1-ab), were markedly enriched in *Enl*-T1 nephrons (Fig. 2c–f). Pseudotime trajectory analysis suggested a potential perturbation in nephron differentiation by *Enl*-T1, leading to two possible developmental paths from uncommitted NPCs (C0-NP1) (Fig. 2g and Supplementary Data 2). One path resembles the differentiation trajectory typically observed in *Enl*-WT nephrons. However, most *Enl*-T1 cells may not progress past the committed NPCs (NP2) phase in this trajectory. An alternative path shows that NP1 cells might transition directly to T1-abnormal clusters (C4 and C10), with these clusters inferred to be poorly differentiated based on pseudotime scores (Fig. 2g). Our results reveal the impact of *Enl*-T1 on the composition of nephron cells and their possible differentiation paths. We propose that *Enl*-T1 results in arrested nephrogenesis at early progenitor stages and the emergence of abnormal, undifferentiated progenitors (Fig. 2n), a hypothesis that requires further validation through lineage tracing experiments.

## Transcriptional changes induced by mutant ENL in the developing kidney

To identify transcriptional changes induced by the mutant ENL in nephrons, we performed differential gene expression analysis between *Enl*-WT and *Enl*-T1 in cell populations that are reasonably abundant in

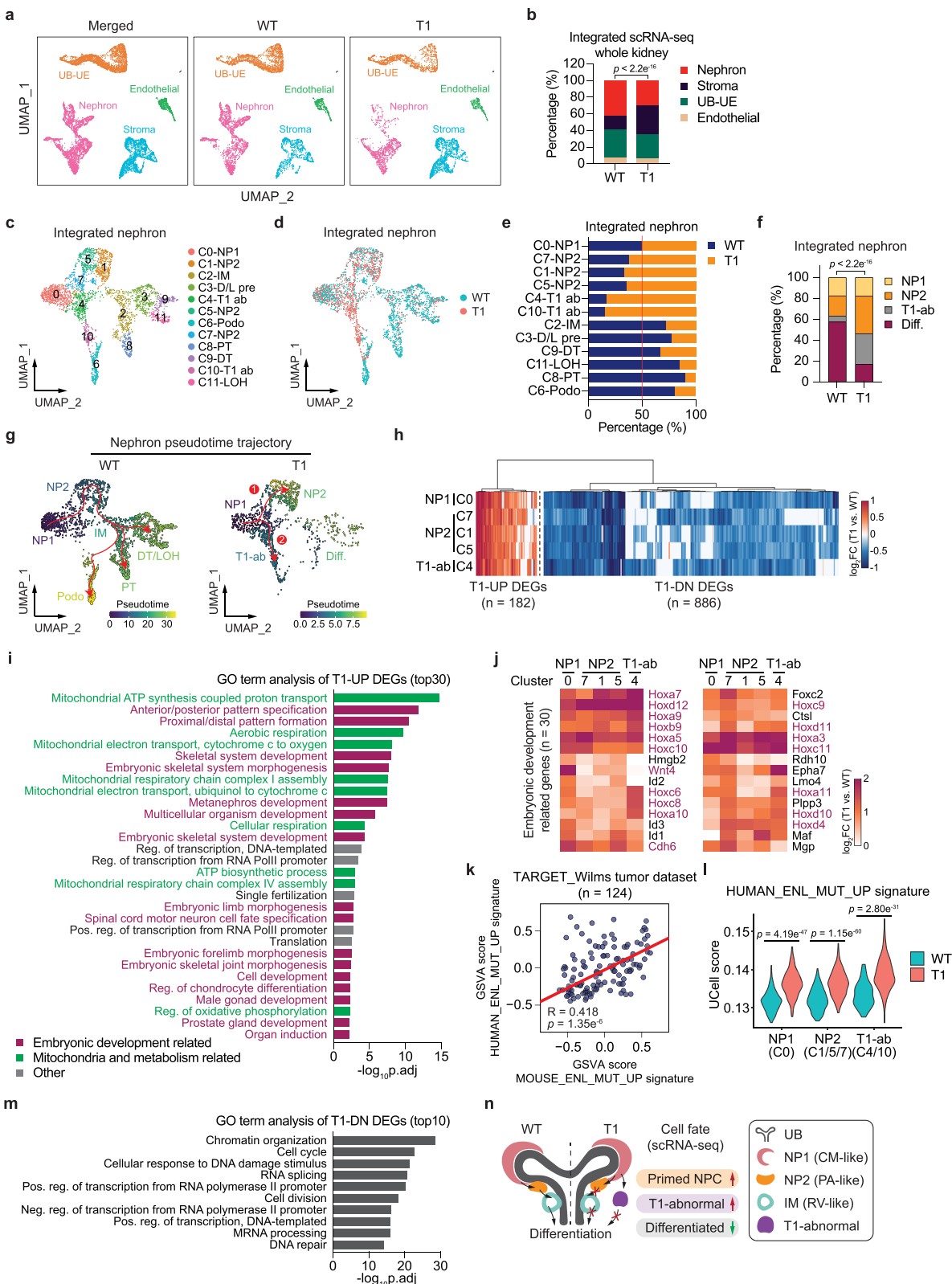

the *Enl*-T1 nephron (NP1, NP2, and T1-ab). We confined our analysis of T1-abnormal cells to cluster 4, as cluster 10 was omitted due to a paucity of *Enl*-WT cells in this cluster (*n* < 50). We identified 1068 differentially expressed genes (DEGs, fold-change > 1.4), with a significant portion of DEGs shared among all five clusters examined (Fig. 2h). Interestingly, while there were more downregulated than upregulated genes in T1 cells (Supplementary Fig. 4a), the upregulated DEGs

exhibited a higher degree of changes in the percentage of cells expressing these genes (Supplementary Fig. 4b). Additionally, T1-upregulated (T1-UP) DEGs exhibited a higher degree of overlap across all five clusters (59 out of 182 DEGs) compared to downregulated (T1-DN) ones (122 out of 886 genes) (Supplementary Fig. 4c, d), suggesting that the impact of the mutation on the upregulated genes is more consistent.

**Fig. 2 | Mutant ENL alters the cellular composition, differentiation trajectories, and gene expression programs in the developing kidney. a** UMAP embedding of scRNA-seq data labeled with four main embryonic kidney lineages. UE, ureteric epithelium. **b** The percentage of four main embryonic kidney compartments within samples. **c, d** UMAP embedding of integrated scRNA-seq cells from *Enl*-WT and T1 nephrons. Cells are colored and labeled by annotated cell types (**c**) or samples (**d**). D/L pre, distal tubule/loop-of-Henle precursor; T1 ab, T1-abnormal; Podo, podocyte; LOH, loop-of-Henle. **e, f** The relative distribution of indicated cell clusters between *Enl*-WT and T1 nephrons (**e**) or within *Enl*-WT or T1 nephron (**f**). Diff., differentiation, including RV, Podo, PT, D/L pre, DT, and LOH cell types. **g** UMAP embedding represents *Enl*-WT and T1 scRNA-seq nephron differentiation trajectory. Cells are colored by pseudotime scores. Trajectories are depicted in red. Two distinct trajectories in *Enl*-T1 nephron are highlighted as "1" and "2". **h** The log2 fold change of differentially expressed genes (DEGs) between *Enl*-WT and T1 within the indicated nephron cell types. T1-UP (or T1-DN), upregulated (or downregulated) in *Enl*-T1 cells. **i, m** Gene ontology (GO) term analysis for the union T1-UP (**i**) and DN (**m**) DEGs shown in (**h**). **j** The log2 fold change of embryonic development-related T1-UP DEGs identified in (**i**) within the indicated nephron cell types. **k** Pearson correlation analysis of the Gene Set Variation Analysis (GSVA) scores evaluated by human and mouse ENL_MUT_UP signatures for the Wilms tumor patients from the TARGET dataset. See Supplementary Data 4. Each dot represents the GSVA scores of one TARGET_Wilms tumor patient. R, Pearson correlation coefficient. **l** The UCell score evaluated by human ENL_MUT_UP signature for the indicated nephron cell types within *Enl*-WT and T1 datasets, respectively. **n** Schematic illustrating the impaired nephrogenesis in the *Enl*-T1 nephron revealed by scRNA-seq. **b, f** Two-tailed Chi-Square test. **i, m** One-sided Fisher's Exact test adjusted by Benjamini–Hochberg procedure. **k** Two-side *t*-test. **l** Two-side Wilcoxon rank-sum test.

Gene ontology (GO) analysis revealed that T1-UP DEGs were enriched in pathways associated with embryonic development, nephron development, and mitochondrial metabolism (Fig. 2i). Among T1-UP DEGs involved in development, we observed a marked upregulation of numerous *Hox* genes across all five clusters (Fig. 2j). In *Enl*-WT nephrons, *Hoxa* and *Hoxb* genes are ubiquitously expressed in the entire nephron, while *Hoxc* and *Hoxd* genes exhibit more cell-type specificity, with *Hoxc* expressed in NPCs and podocytes and *Hoxd* expressed in NPCs and DT (Supplementary Fig. 4e). *Hoxb* genes exhibited a relatively modest increase compared with the other three subfamilies, possibly due to high basal level in *Enl*-WT nephrons (Supplementary Fig. 4f). Among the *Hox* genes that are abnormally upregulated by *Enl*-T1, *Hox9, Hox10, Hox11* genes have been established as critical regulators of NPC self-renewal and precise execution of differentiation in genetic knockout studies[46–48]. This, together with our results, indicate that proper levels of *Hox* genes are crucial for kidney development. Consistent with scRNA-seq, RT-qPCR analyses of whole E15.5 kidneys revealed a pronounced upregulation of *Hox9/10/11* gene expression in *Enl*-T1 kidneys compared to *Enl*-WT counterparts (Supplementary Fig. 4g). Importantly, *ENL*-mutant human Wilms tumors exhibit elevated levels of certain *HOX* genes (e.g., *HOXA13*) compared to *ENL*-WT tumors[32], supporting the clinical relevance of our findings and underscoring a potential role of *HOX* genes in Wilms pathogenesis. Intriguingly, genes involved in NPC commitment were also upregulated in *Enl*-T1 cells. For instance, *Wnt4*, which is typically expressed in the committed NPCs and functions as an inducer of MET[49,50], had elevated expression in *Enl*-T1 NP1 (C0) and NP2 (C1/5) cells (Fig. 2j). Furthermore, *Cdh6*, an epithelial marker gene usually expressed in the RV[41], was aberrantly upregulated by *Enl*-T1 in all five clusters analyzed, including the NPCs (Fig. 2j). These results suggest that *Enl*-T1 induces transcriptional changes involved in both self-renewal and cell fate commitment in nephron progenitors.

Quite interestingly, we also observed upregulation of genes related to mitochondria and metabolism in the nephron (Fig. 2i and Supplementary Fig. 4h). Among these genes, the *Nduf*, *Atp*, and *Cox* gene families are linked to the respiratory electron transport chain (ETC)[51]. It has been shown that the ETC pathway becomes increasingly activated as nephron progenitors differentiate in the fetal kidney[52], suggesting the possibility that *Enl*-T1 may promote premature commitment of NPCs in part through augmenting ETC activation. This hypothesis merits further exploration.

To determine whether the gene signatures are induced by *Enl*-T1 specifically in the nephron, we scored their expression across all major lineages using UCell, a tool for interrogating gene signatures in single-cell datasets[53]. Among *Enl*-T1-upregulated genes, the development-related signature was enhanced by *Enl*-T1 predominantly in the nephron and stroma, while the mitochondria-related signature was increased throughout the entire kidney (Supplementary Fig. 4i). Given that *Enl*-T1 expression was restricted to *Wt1*+ cells and their progeny in the nephron and stroma, these results suggest that the alteration of development-associated signatures is more likely a direct effect of *Enl*-T1 compared to the mitochondria-related signature.

We next assessed the clinical relevance of *Enl*-T1-induced gene signatures identified in our mouse model. We obtained RNA-seq datasets for Wilms tumor samples from the TARGET dataset[20] and identified genes that are upregulated in *ENL*-mutant compared to *ENL*-wildtype tumors (Supplementary Data 3). We then performed gene set variation analysis (GSVA) to score both human and mouse ENL-mutant-UP signatures across all Wilms tumor samples and observed a positive correlation, suggesting their functional associations in human Wilms tumors (Fig. 2k and Supplementary Data 4). Next, we assessed the expression of human ENL-mutant-UP signature in specific nephron cell types using our scRNA-seq datasets. *Enl*-T1 cells in the NP1, NP2, and T1-ab clusters exhibited higher expression levels of this signature compared to their wildtype counterparts, with the most significant increase occurring in T1-ab populations (Fig. 2l). Collectively, these findings support the clinical relevance of our models in uncovering mutant ENL-induced transcriptional changes with implications for human Wilms tumors.

Next, we turned to genes that were downregulated in *Enl*-T1 nephrons. GO analysis revealed that T1-DN DEGs were associated with chromatin organization and transcription-related functions (Fig. 2m and Supplementary Fig. 4j). Notably, *Dnmt1*, a methyltransferase essential for maintaining DNA methylation[54], is known to be enriched in the nephrogenic zone of the developing kidney[55]. Studies have demonstrated that the loss of *Dnmt1* in nephron progenitors resulted in reduced self-renewal, leading to impaired renal differentiation and postnatal mortality in mice[55,56]. Additionally, key epigenetic regulators involved in histone methylation, such as *Ash2l* and *Ezh2*, which are expressed at higher levels in the *Six2*+ cap mesenchyme, were downregulated in *Enl*-T1 cells[57]. Ezh2 serves as the dominant H3K27 methyltransferase in *Six2*+ NPCs and is essential for their proper proliferation[58]. The reduced expression of these and other epigenetic regulators in *Enl*-T1 nephron precursors likely contributes to the observed nephrogenesis defects in *Enl*-T1 kidneys, a hypothesis that warrants further investigation. Like the *Enl*-T1-induced mitochondria-related signature, *Enl*-T1-downregulated genes found in the nephron showed a similar degree of decrease across different lineage compartments within the kidney (Supplementary Fig. 4k), suggesting a potential secondary effect of *Enl*-T1 expression.

## Chromatin accessibility dynamics and regulatory landscape during normal nephrogenesis

Dynamic control of chromatin accessibility plays a crucial role in cell differentiation by regulating the access of cell-type specific transcription factors (TFs) to their regulatory genomic regions[2,59]. To probe whether *Enl*-T1 affects the chromatin landscape in specific cell types, we performed single-nuclei Assay for Transposase Accessible

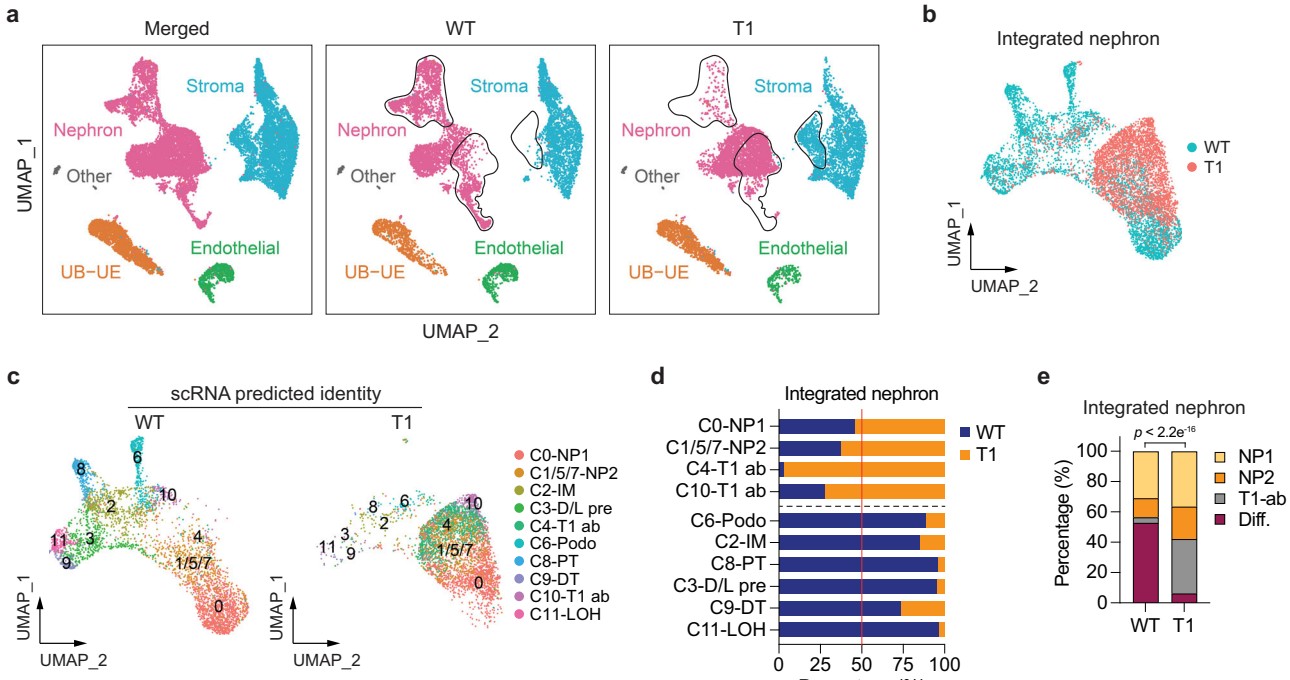

**Fig. 3 | snATAC-seq reveals changes in the open chromatin landscape in *Enl*-mutant kidneys during nephrogenesis. a** UMAP embedding of snATAC-seq data labeled with four main embryonic kidney lineages. The dramatically altered clusters between *Enl*-WT and T1 kidneys are highlighted with black circles. **b** UMAP embedding of integrated snATAC-seq cells from *Enl*-WT and T1 nephrons, colored by sample. **c** UMAP embedding of integrated snATAC-seq cells from *Enl*-WT (left)

and T1 (right) nephrons, respectively. Cells are colored and labeled by the cell types predicted by corresponding scRNA-seq data. **d** Stacked bar plot showing the percentage of scRNA-seq predicted cell types between *Enl*-WT and T1 nephrons. **e** Stacked bar plot showing the percentage of scRNA-seq predicted cell types within *Enl*-WT and T1 nephrons. Two-tailed Chi-Square test *p*-value is shown. Diff., differentiation, including RV, Podo, PT, D/L pre, DT, and LOH cell types.

Chromatin sequencing (snATAC-seq) on E15.5 *Enl*-WT and T1 kidneys. After applying a series of quality control metrics, we retained 17102 high-quality cells for subsequent analysis (Supplementary Fig. 5a–e and Supplementary Data 1). We first mapped the open chromatin landscape of *Enl*-WT kidneys. Based on the chromatin accessibly around the transcription start site (TSS) and gene body regions of well-known cell type-specific marker genes, we identified four major lineage compartments (Supplementary Fig. 5f) as in our scRNA-seq dataset. Within the nephron, cells were further grouped into several major cell types, including NPCs, intermediate cells, podocytes, PT, and DT/LOH (Supplementary Fig. 5f, g). Differential analysis identified 56,258 cell-type specific ATAC peaks across all eight major clusters within the kidney, with a majority exhibiting high specificity for a single cluster (Supplementary Fig. 5h and Supplementary Data 5). Motif enrichment analysis nominated top TFs that occupy these cell type-specific open regulatory elements, many of which also exhibited cell type-specific expression patterns (Supplementary Fig. 5i, j). Our results, together with recent scATAC-seq profiling of late-stage embryonic and postnatal kidneys[60,61], define the cell type-specific chromatin and TF regulatory landscape during nephrogenesis in the mouse kidney.

To further understand the dynamic changes in chromatin accessibility that occur during cell fate transition, we defined major cell state transitions in *Enl*-WT nephrons based on differentiation trajectory using our snATAC-seq dataset (Supplementary Fig. 6a–c). We then identified differentially accessible regions (DARs) between descendant cell states along the trajectory (Supplementary Fig. 6d and Supplementary Data 6). Surprisingly, the transition from NP1 to NP2, signifying the commitment of NPCs, was characterized by a substantial gain in chromatin accessibility. Conversely, the transition from NP2 to podocytes showed a more pronounced loss in chromatin accessibility. The most substantial remodeling in chromatin accessibility occurred during the transition from NP2 to IM, coinciding with MET, a critical

step for committed NPC to differentiate into various tubule structures[13–15].

Motif enrichment analysis on the DARs identified candidate TFs that may regulate each step of cell fate transitions (Supplementary Fig. 6e). Upon examining the expression level and motif dynamic score using chromVAR[62], we nominated 7 transcription factors (TFs), namely *Six2*, *Hoxc9*, *Wt1*, *Tcf21*, *Lhx1*, *Hnf1b* and *Hnf4a*, as key regulators orchestrating cell fate transitions during nephrogenesis with distinct dynamics (Supplementary Fig. 6f, g). For instance, *Six2* and *Hoxc9*, two NPC-specific TFs[39,60], exhibited a gradual decrease in both their expression and binding site accessibility during the transition from the NP2 to subsequent stages. *Tcf21*, a key TF for glomerulogenesis[63], was de novo activated during the NP2 to podocyte transition. *Lhx1* and *Hnf1b* were both highly expressed in IM cells, but they exhibited a mutually exclusive pattern of activity in subsequent lineages, with *Lhx1* activated in podocytes and DT and *Hnf1b* activated in PT. Thus, while an earlier study focusing on postnatal and adult kidneys highlighted the closing of open chromatin regions as the primary event during nephron differentiation[60], our findings uncover chromatin dynamics and key regulatory TFs during cell fate transitions in the developing mouse kidney.

### *Enl*-mutant kidneys exhibit an altered open chromatin landscape during early nephrogenesis

Having established the open chromatin landscape of the *Enl*-WT embryonic kidneys, we investigated how this landscape is impacted by *Enl*-T1. Through the integration of snATAC-seq datasets obtained from *Enl*-WT and T1 kidneys, we found that *Enl*-T1 kidneys exhibited a reduction in specific nephron clusters and an increase in certain stromal clusters (Fig. 3a, b). Through label transfer[64], we used scRNA-seq annotations as a reference to assign cell type identity to nephron cells in snATAC-seq datasets (Fig. 3c). This analysis revealed several

intriguing observations. First, while *Enl*-WT nephron cells displayed a variety of chromatin states corresponding to cell types found in nephrogenesis, *Enl*-T1 cells exhibited a marked reduction in chromatin state diversity. In particular, *Enl*-T1 cells showed a substantial loss of chromatin states associated with differentiated nephron structures (e.g., PT, DT, Podo) (Fig. 3d, e) and a more uniform chromatin state of clusters representing undifferentiated cells. Second, while *Enl*-T1 NP1 cells clustered with *Enl*-WT counterparts based on gene expression (Fig. 2c, d), they exhibited a shift towards NP2 cells (C1/5/7) in the snATAC-seq dataset (Fig. 3c). This observation was further supported by a lower number of DARs between NP1 and NP2 in *Enl*-T1 (NP2 vs. NP1: 2153 gained and 267 lost) compared to the DARs in *Enl*-WT nephrons (NP2 vs. NP1: 11020 gained and 547 lost). These findings suggest that *Enl*-T1 NP1 cells may have substantial chromatin alterations before overt transcriptional changes occur. Furthermore, although scRNA-seq analysis indicated that NP2 (C1/5/7) and T1 abnormal clusters (C4/10) in *Enl*-T1 nephrons follow distinct differentiation paths originating from NP1 (Fig. 2g), a higher degree of similarity in chromatin accessibility was observed between these clusters (Fig. 3c). These results reveal dynamic changes in chromatin accessibility induced by the *Enl* mutation, shedding light on aspects that would otherwise remain masked in scRNA-seq analysis alone.

## Mutant ENL promotes premature commitment of nephron progenitors while restricting their differentiation through misregulation of specific TF regulons

To understand the regulatory mechanism underlying altered chromatin states in *Enl*-T1 uncommitted NPCs, we compared chromatin accessibility between WT and T1 NP1 cells. We identified 8958 gained and 5616 lost DARs in *Enl*-T1 cells (Fig. 4a and Supplementary Data 7), the majority of which were in distal intergenic regions or gene bodies, likely representing enhancers (Supplementary Fig. 7a). GREAT analysis (Fig. 4b, Supplementary Fig. 7b, and Supplementary Data 7) revealed that T1-gained, but not T1-lost, DARs were strongly associated with functional terms related to nephrogenesis, such as metanephric nephron morphogenesis and renal vesicle morphogenesis, indicating that *Enl*-T1 NP1 cells are more primed to commitment and differentiation at the chromatin level. Motif enrichment coupled with gene expression analysis identified several *Hox* TFs (*Hoxa9*, *Hoxc9*, *Hoxa11*, and *Hoxd11*) as top regulators of gained DARs (Fig. 4c). Notably, *Hoxc9* has been identified as a key TF driving the NP1 to NP2 transition during normal nephrogenesis (Supplementary Fig. 6f). In contrast, expression levels and motif accessibility of several TFs important for NPC self-renewal, such as *Six2*[39] and *Wt1*[35], were decreased in *Enl*-T1 NP1 cells, with *Six2* standing out as the most significant hit (Fig. 4c). These results suggest that mutant ENL may disrupt the balance of self-renewal and commitment of NPCs through misregulation of key TF regulons and prime these cells for differentiation at the chromatin level. Furthermore, we found that 20% of *Enl*-T1 gained DARs in NP1 cells coincided with open chromatin regions associated with the NP1 to NP2 transition during normal nephrogenesis (Supplementary Figs. 6d, 7c and Supplementary Data 8), suggesting that *Enl*-T1 promotes premature commitment of NP1 cells through the acquisition of normal development-associated as well as de novo open chromatin regions.

Despite the premature commitment of *Enl*-T1 NP1 cells to a NP2-like state, these cells failed to give rise to differentiated kidney structures. To explore the underlying mechanisms, we compared the chromatin accessibility between *Enl*-WT and *Enl*-T1 NP2 cells. As in the case of T1-induced DARs in NP1 cells, DARs identified in NP2 cells were primarily located in distal regulatory regions (Fig. 4d and Supplementary Fig. 7d). However, GREAT analysis revealed that these DARs were not directly related to nephrogenesis but instead, they exhibited strong enrichment in mesenchyme development pathways (Fig. 4e and

Supplementary Fig. 7e). Given that normal NP2 represents primed NPCs in the pretubular aggregates that undergo MET, a prerequisite for nephron differentiation, our results suggest that *Enl*-T1 NP2 cells may exhibit a defect in differentiation due to aberrant maintenance of a mesenchymal chromatin state. Furthermore, an integrative analysis of TF motif enrichment and expression revealed similar regulators in *Enl*-T1 NP2 cells as those identified in NP1 cells (Fig. 4f), underscoring the sustained impact of the mutant on the chromatin state of self-renewing and committed nephron progenitors during early nephrogenesis. Interestingly, our scATAC-seq on *Enl*-WT embryonic kidneys revealed that both the expression levels and activity of *Hoxc9* gradually decrease during the transition from NP2 to IM (Supplementary Fig. 6f, g), a precursor stage for podocyte and tubule differentiation (Supplementary Fig. 3h). Therefore, we speculated that the persistently elevated expression and activity of *Hoxc9* and possibly other *Hox* TFs contribute to the impeded differentiation of *Enl*-T1 NP2 cells. To address this hypothesis, we performed a regulon analysis to identify putative target genes of *Hox* TFs in NP1 and NP2 cells. The results showed that > 80% of *Enl*-T1-gained DARs contained at least one motif sequence of identified *Hox* TFs (Supplementary Fig. 7f). We then identified genes associated with *Hox* motif-containing DARs and examined their overlap with T1-upregulated DEGs. We found that 24% of T1-upregulated DEGs in NP1 cells and 13% in NP2 cells have the potential to be activated by *Hox* TFs (Fig. 4g and Supplementary Fig. 7g, h). Notably, among these targets, *Wnt4*, essential for NPC commitment and MET initiation[49,50], and *Cdh6*, an epithelial marker in RV[41], showed aberrant activation in *Enl*-T1 NP1 and NP2 cells, respectively (Fig. 4h). Over 55% of *Enl*-T1 NP1 cells expressed *Wnt4*, compared to only ~10% of *Enl*-WT cells (Fig. 4i). Similarly, the percentage of *Cdh6*+ cells in NP2 cells was increased from 17.1% in *Enl*-WT to 58.4% in *Enl*-T1 (Fig. 4j). Furthermore, we observed an increase in chromatin accessibility at the *Wnt4* and *Cdh6* gene loci, which likely contributes to their elevated expression (Fig. 4k, l).

To validate transcriptional changes identified by scRNA-seq, we performed RNA in situ hybridization (ISH) and/or immunostaining for several key genes in E15.5 kidneys. First, we focused on *Hoxc9*, a top transcription factor candidate predicted to mediate increased chromatin accessibility in *Enl*-T1 *Six2*+ NPCs (Fig. 4c and f). *Hoxc9* mRNA ISH coupled with SIX2 protein co-staining showed that *Hoxc9* was highly expressed in SIX2+ NPCs in *Enl*-WT kidneys (Fig. 4m), consistent with our scRNA-seq data (Supplementary Fig. 4e). Notably, *Hoxc9* ISH signals were elevated in SIX2+ NPCs in *Enl*-T1 kidneys (Fig. 4m). We then aimed to confirm the upregulation of *Wnt4* and *Cdh6* in *Enl*-T1 NP1 and NP2 cells. Imaging data showed that in *Enl*-WT kidneys, *Wnt4* was present in SIX2low cells within the PA structure, likely corresponding to NP2 cells, and absent in SIX2high cells in the CM structure, likely corresponding to NP1 cells (Fig. 4n and Supplementary Fig. 7i). This pattern aligns well with our scRNA-seq data (Fig. 4h) and previous findings[50]. We observed increased *Wnt4* ISH signals in *Enl*-T1 SIX2+ NP1 and NP2 cells relatively to *Enl*-WT (Fig. 4n and Supplementary Fig. 7i). During normal kidney development, *Cdh6* expression begins in the intermediate (IM) cells within the renal vesicle and subsequently appears in differentiated cells (Fig. 4h). CDH6 immunostaining confirmed that in *Enl*-WT kidneys, CDH6 was not detected in NPCs located in the CM and PA structures, but it was present in cells within the RV structure, which likely represent the IM cells identified in our scRNA-seq analysis. In contrast, in *Enl*-T1 kidneys, *Cdh6* was aberrantly expressed in NP2 cells, as revealed by scRNA-seq (Fig. 4h), and CDH6 protein was detected within the PA structure (Supplementary Fig. 7j). These in situ validation results lend further support to findings from scRNA-seq. We propose that the *Hox*-mediated increase in chromatin accessibility and gene expression of key developmental genes, such as *Wnt4* and *Chd6*, may underlie the aberrant commitment of *Enl*-T1 NPCs (Fig. 4o).

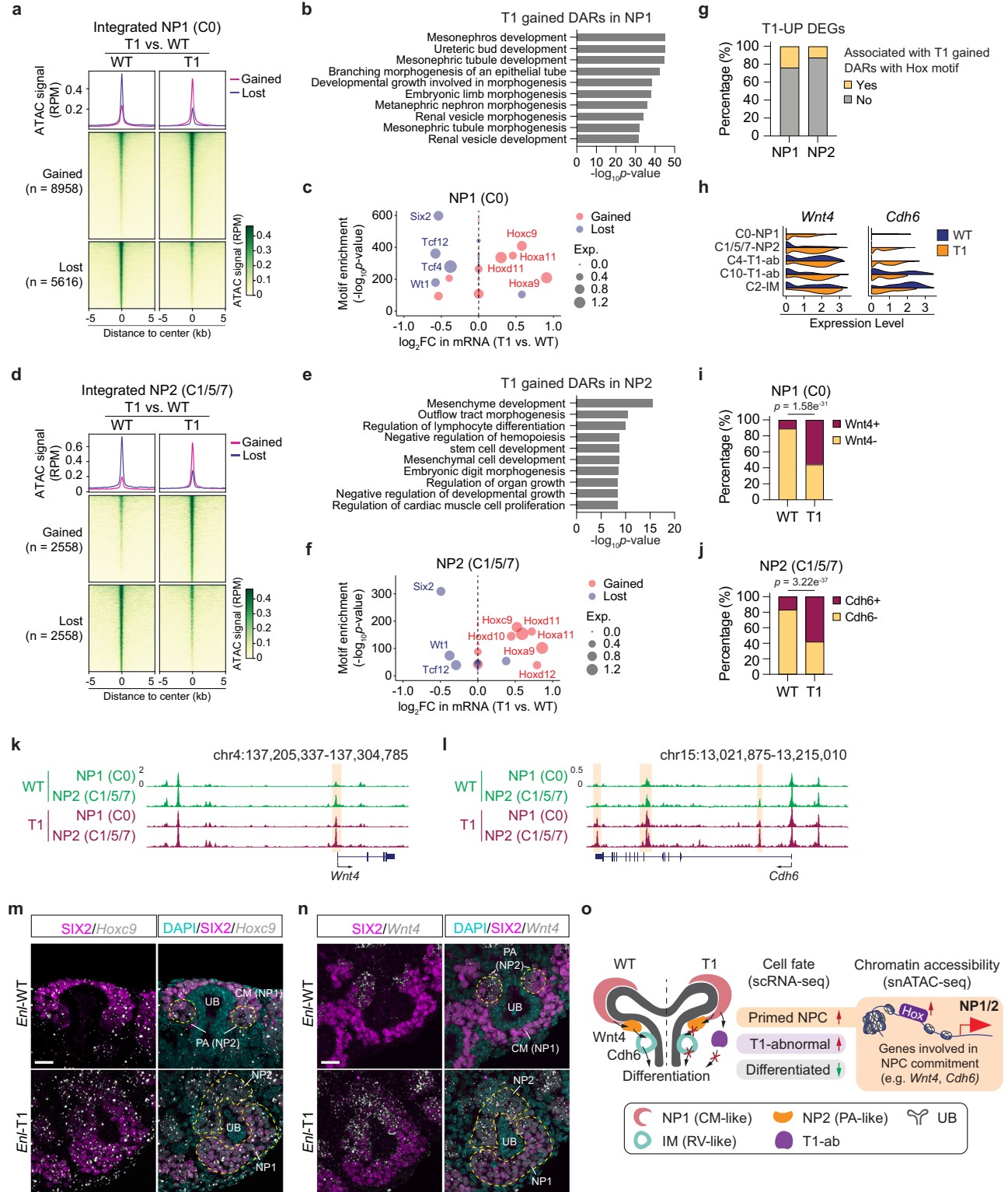

### An abnormal progenitor state losing nephron chromatin identity appears in *Enl*-mutant kidneys

Next, we aimed to characterize the abnormal cell cluster (C4) gained in *Enl*-T1 nephron. Trajectory analysis on the scATAC-seq dataset placed NP2 and T1-ab cells downstream of NP1 cells, and NP2 and T1-ab cells exhibited similar differentiation scores (Fig. 5a and Supplementary Fig. 8a). These results might suggest that NP2 and T1-ab cells represent subsets of primed NPCs. Moreover, trajectory analysis based on

scRNA-seq dataset points to a potential divergence from the self-renewing NP1 to either NP2 or T1-ab cells, highlighting differences in the cellular state between these two populations (Fig. 2g and Supplementary Fig. 8b, c). To explore the mechanism underlying the emergence of T1-abnormal cells, we identified DARs between the predominant T1-ab cluster (C4) and NP1 (C0) or NP2 (C1/5/7) in *Enl*-T1 nephrons. The analysis revealed substantial differences in open chromatin landscapes between T1-ab and NP1, with 1109 gained and 2518

**Fig. 4 | Mutant ENL promotes premature commitment of nephron progenitors while restricting their differentiation through misregulation of specific TF regulons. a, d** snATAC data in NP1 (**a**) and NP2 (**d**) are plotted as average occupancies and heatmap across the differentially accessible regions (DARs) between *Enl*-WT and T1 cells. The ATAC signal is normalized by reads per million (RPM). All regions are defined as gained (upregulated in *Enl*-T1) and lost (down-regulated in *Enl*-T1). See Supplementary Data 7. **b, e** GREAT analysis of T1-gained DARs in NP1 (**b**) and NP2 (**e**). Binomial test *p*-values are shown. **c, f** Dot plot showing the top 20 most significant TFs identified from the motif enrichment analysis for T1 gained or lost DARs in NP1 (**c**) and NP2 (**f**). The size of the circles represents the expression level in *Enl*-WT (blue) or *Enl*-T1 (red) cells. Binomial test *p*-values are shown. FC, fold-change. Exp., expression. **g** Stacked bar plots indicating the percentage of T1-UP DEGs associated with Hox motif enriched T1 gained DAR. **h** Violin plots showing the expression level of *Wnt4* and *Cdh6* for NP1, NP2, T1-ab, and IM cell types in *Enl*-WT

or T1 nephrons. **i** Stacked bar plots indicating the percentage of NP1 cells with or without *Wnt4* expression in *Enl*-WT or T1 nephrons. One-side Fisher's exact test *p*-value is shown. **j** Stacked bar plots indicating the percentage of NP2 cells with or without *Cdh6* expression in *Enl*-WT or T1 nephrons. One-side Fisher's exact test *p*-value is shown. **k, l** The genome browser view of ATAC signals at *Wnt4* (**k**) and *Cdh6* (**l**) gene loci in indicated cell types from *Enl*-WT (top) or T1 (bottom) nephrons. **m** Representative images of SIX2/*Hoxc9* mRNA co-staining in E15.5 kidneys. CM, cap mesenchyme; UB, ureteric bud; PA, pretubular aggregate. Scale bar = 20 μm. **n** Representative images of SIX2/*Wnt4* mRNA co-staining in E15.5 kidneys. CM cap mesenchyme, UB ureteric bud, PA peritubular aggregate. Scale bar = 20 μm. Data shown in (**m, n**) are representative of three *Enl*-WT or T1 kidneys. **o** Schematic illustrating impaired nephrogenesis phenotypes and potential molecular mechanism we identified in the *Enl*-T1 NP1 and NP2.

lost DARs in T1-ab cells (Fig. 5b and Supplementary Data 9). In contrast, when comparing T1-ab and NP2, we observed a decrease in ATAC signals in only 629 chromatin regions (Fig. 5c and Supplementary Data 9). These results suggest a more similar chromatin state between T1-ab and NP2 cells compared to the differences observed between T1-ab and NP1 cells. GREAT analysis on these DARs showed that T1-ab cells lost the potential for nephrogenesis compared with NP1 (Fig. 5d) and NP2 cells (Fig. 5e). Specifically, when compared to NP2 cells, DARs lost in T1-ab cells were enriched for GO terms related to renal vesicle morphogenesis and mesenchymal to epithelial transition, indicating a more pronounced defect in their differentiation into renal vesicles (Fig. 5e). Intriguingly, T1-ab cells gained the potential at the chromatin level to differentiate into other organs (Fig. 5f). These results suggest that the mutant ENL induces an abnormal progenitor state characterized by a loss of open chromatin regions encoding nephron identity or differentiation potential.

To identify TFs that regulate chromatin changes in T1-ab cells, we performed motif enrichment analyses on identified DARs. There was a strong enrichment of motifs for several TFs involved in NPC self-renewal and maintenance, such as *Six2*[39] and *Tcf21*[65], in T1-ab lost DARs when compared with NP1 cells (Fig. 5g). Similar results were observed in T1-ab vs. NP2 comparison, albeit to a lesser extent, which may be attributed to a lower number of DARs (Fig. 5h). Among these TFs, *Six2* exhibited the most significant gene expression alteration among NP1, NP2, and T1-ab cells, with T1-ab cells showing the lowest expression levels (Fig. 5i and Supplementary Fig. 8d). As the *Six2* gene locus displayed minimal changes in ATAC signals across these three populations (Supplementary Fig. 8e), downregulation of its upstream regulators or changes in other chromatin features may underlie the observed expression changes. Nevertheless, the loss of *Six2* binding regions observed in T1-ab cells likely contributes, at least in part, to the reduced potential of T1-ab to specify the kidney lineage.

On the other hand, several TFs, including *Hoxc9*, *Hoxb13*, *Cdx2*, *Pbx2*, and *Cdx4*, were identified as top candidates that potentially regulate T1-ab gained DARs when compared with NP1 cells (Fig. 5g), despite their similar expression levels in T1-ab and NP1 cells (Fig. 5i). Intriguingly, motifs for these same TFs were also enriched, albeit to a lesser extent, in T1-ab lost DARs (vs. NP1) (Fig. 5g). These TFs may cooperate with distinct partners to induce the opening or closing of different chromatin regions. Among these TFs, *Hoxc9* ranked as the top TF candidate occupying T1-ab gained DARs. We have demonstrated that *Hoxc9* is a key TF driving NPC commitment (NP1 to NP2 transition) during normal nephrogenesis (Supplementary Fig. 6e, g). As both T1-ab and T1-NP1 cells exhibited aberrantly high levels of *Hoxc9*, we speculated that *Hoxc9* may promote the opening of additional chromatin regions in T1-ab cells that regulate other lineage specifications, a hypothesis requiring further investigation.

Having determined the open chromatin landscape in T1-ab cells, we next investigated the transcriptional changes underlying their aberrant cell state. We identified DEGs across T1-ab, NP1, and NP2 cells

in *Enl*-T1 nephrons (Supplementary Fig. 8f and Supplementary Data 10) and performed GO term analyses for these DEGs. Genes that were downregulated in T1-ab cells compared to NP1 were enriched in pathways related to kidney development (Fig. 5j). Notably, the expression levels of several well-known NPC markers were markedly decreased in T1-ab cells, consistent with these cells losing the NPC state (Fig. 5l). Additionally, compared to NP2, T1-ab cells exhibited a defect in proliferation, a cellular process required during normal NPC commitment[66] (Fig. 5k and Supplementary Fig. 8g). Furthermore, genes upregulated in T1-ab cells compared to either NP1 or NP2 are associated with the development of other tissues/organs or in metabolism (Fig. 5m and Supplementary Fig. 8h, i). Integrated analysis of scRNA-seq and snATAC-seq datasets revealed that only a subset of DEGs was linked to changes in chromatin accessibility (Supplementary Fig. 8j and Supplementary Data 11), suggesting additional mechanisms involved in transcriptional regulation. Taken together, our chromatin and transcriptional analyses indicate that T1-abnormal cells are arrested in an aberrant progenitor state characterized by a loss of kidney lineage identity and a gain of potential to develop into other lineages (Fig. 5m).

## Mutant ENL alters the open chromatin landscape of *Foxd1*+ stromal progenitors

The kidney stroma plays a crucial role in renal morphogenesis through interactions with the nephron and ureteric bud[17–19,43]. However, the precise origins, developmental hierarchy, and regulatory mechanisms of the stroma remain poorly understood. Our scRNA-seq analyses have shown that *Enl*-T1-induced development-related gene signatures were also found in the stroma (Supplementary Fig. 4i). Moreover, *Enl*-T1 stroma exhibited substantial changes in the open chromatin landscape compared to *Enl*-WT stroma (Fig. 3a). Therefore, we sought to investigate how alterations in the stroma may contribute to the observed nephrogenesis defects in *Enl*-T1 kidneys.

Unbiased clustering of stroma cells extracted from our scRNA-seq datasets resulted in 10 clusters (Fig. 6a, b) representing 6 distinct cell types, including *Foxd1*+ stromal progenitor (SP) (*Foxd1*+, *Dlk*+), proliferating cortical stroma (CS) (*Top2a*+), CS (*Clca3a1*+), medullary stroma (MS) (*Alx1*+, *Wnt4*+), ureteric stroma (US) (*Myh11*+), and renin/mesangial cells (Ren/Mes) (*Ren1*+, *Akr1b7*+) (Supplementary Fig. 9a). The three major stroma subdomains, CS, MS, and US, which exhibit a dorsoventral distribution pattern (CS, dorsal; MS and US, ventral) in the developing kidney[44], were well separated in our scRNA-seq datasets (Fig. 6a, b), indicating distinct gene expression and functions of these subdomains. Trajectory analysis based on scRNA-seq datasets on *Enl*-WT stroma (Supplementary Fig. 9b) suggested that *Foxd1*+ SP (C0/C8) is the origin of CS/MS populations, which constitute most of the kidney interstitium[18]. In contrast, the US cells (C5) represented a distinct population derived from *Tbx18*+ SP, consistent with previous lineage tracing studies[67]. Mutant ENL did not significantly alter the developmental trajectories of stroma cells.

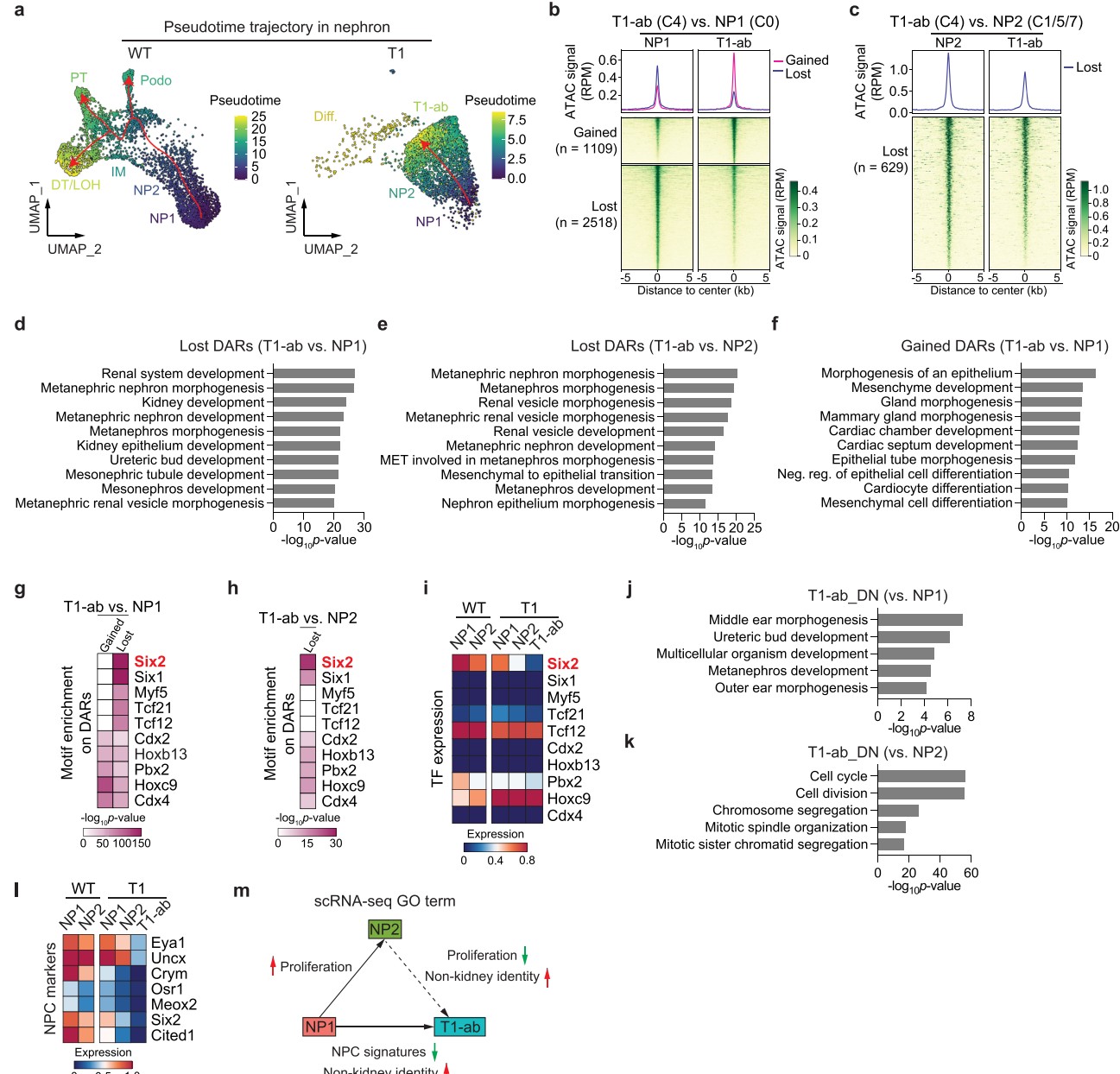

**Fig. 5 | An abnormal progenitor state losing nephron chromatin identity emerges in *Enl*-mutant kidney. a** UMAP embedding of integrated *Enl*-WT (left) or T1 (right) snATAC-seq nephron differentiation trajectory. Cells are colored by pseudotime. Trajectories are depicted by red arrows. **b** snATAC data in *Enl*-T1 for NP1 (left) and T1-ab (right) are plotted as average occupancies (top) and heatmap (bottom) across the DARs between NP1 and T1-ab cells. The ATAC signal is normalized by RPM. All regions are defined as gained (up-regulated in T1-ab, red) and lost (down-regulated in T1-ab, blue), and the corresponding numbers are shown on the left. See Supplementary Data 9. **c** snATAC data in *Enl*-T1 for NP2 (left) and T1-ab (right) are plotted as average occupancies (top) and heatmap (bottom) across the T1-ab lost DARs. The ATAC signal is normalized by RPM. The corresponding DAR numbers are shown on the left. See Supplementary Data 9. **d**–**f** Bar plots showing GREAT analysis for T1-ab vs. NP1 lost (**d**), T1-ab vs. NP2 lost (**e**), and T1-ab vs. NP1 gained (**f**) DARs in *Enl*-T1 cells. **g**, **h**, Heatmap showing the motif enrichment *p*-values of the top TF candidates identified from the motif analysis for the DARs indicated in (**b**, **c**). Binomial test *p*-values are shown. **i** Heatmap showing the expression level of TFs identified in (**g**, **h**) in *Enl*-WT and *Enl*-T1 NP1, NP2, and T1-ab scRNA-seq cells. **j**, **k** Bar plots showing GO term analysis for T1-ab vs. NP1 DN DEGs (**j**) and T1-ab vs. NP2 DN DEGs (**j**) in the *Enl*-T1 nephron. One-sided *p*-values for Fisher's Exact test are shown. **l** Heatmap showing the expression levels of NPC markers in indicated cell types of *Enl*-T1 nephron. **m** Schematic summarizing the GO term analyses performed in (**j**, **k**), and Supplementary Fig. 8g–i.

To gain a comprehensive understanding of chromatin changes induced by mutant ENL in the stroma, we assigned cell types to snATAC-seq datasets based on the corresponding annotations from the scRNA-seq dataset. Several notable chromatin changes in *Enl*-T1 stroma were identified, including an embedding shift of the *Foxd1*⁺ SP cells (C0) and an expansion of the US cluster (C5) in the UMAP (Fig. 6c, d). Histological studies on the mouse embryonic kidney[68] showed that *Foxd1*⁺ SP, but not *Tbx18*⁺ SP, constitutes the stroma

population surrounding the uncommitted NPC-containing cap mesenchyme (CM) (Fig. 6e). This observation, coupled with reported interactions between *Foxd1*⁺ SP and CM, led us to speculate that abnormal *Foxd1*⁺ SP cells in *Enl*-T1 kidneys may impact nephrogenesis by influencing stroma-nephron interactions. Re-clustering of extracted *Foxd1*⁺ SP cells (C0) from the integrated stroma snATAC-seq datasets confirmed that, while a small portion of *Enl*-T1 *Foxd1*⁺ SP cells clustered with *Enl*-WT counterpart, the majority exhibited a distinct open

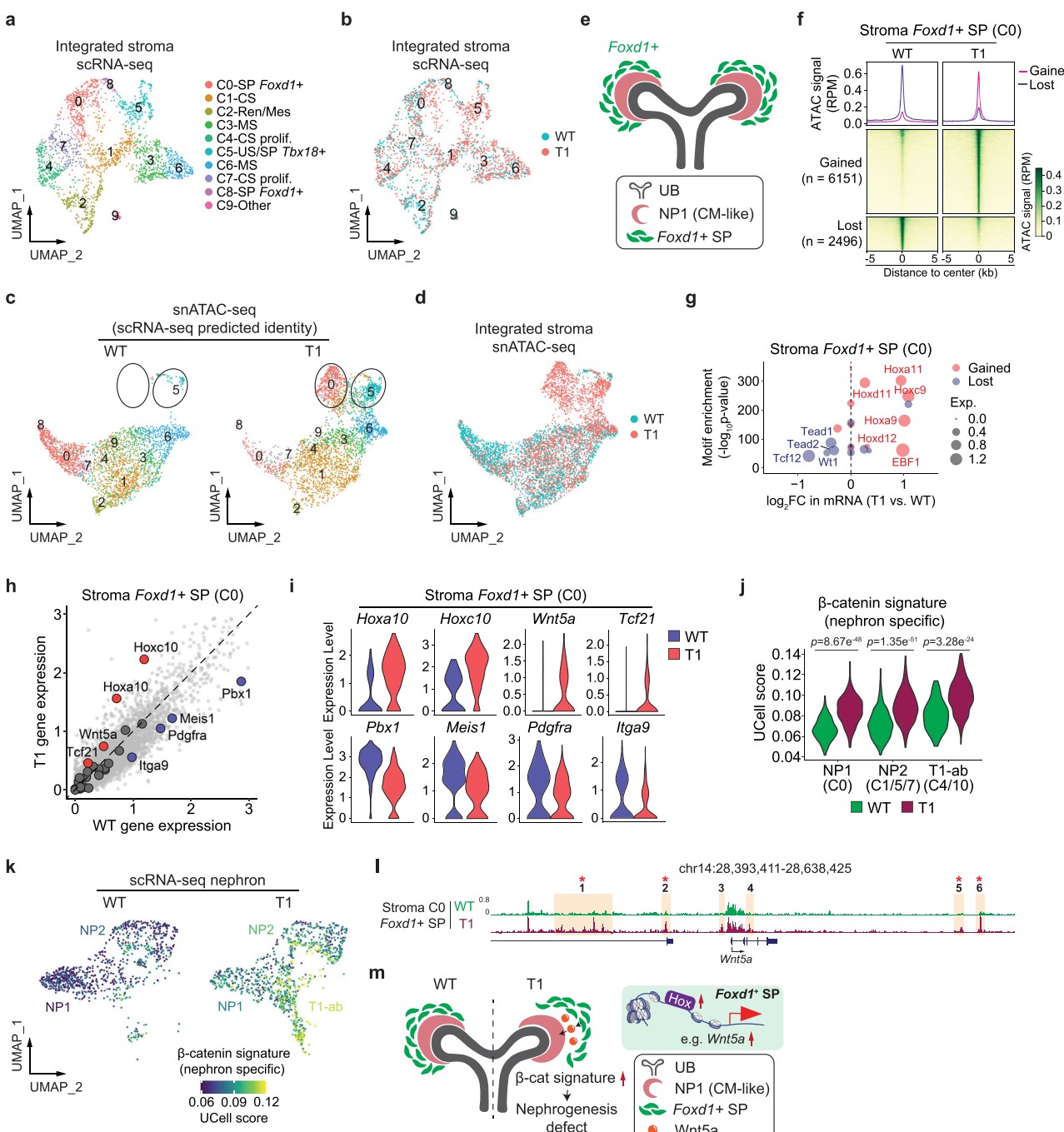

**Fig. 6 | *Enl*-mutant *Foxd1*⁺ stromal progenitors exhibit altered chromatin accessibility and might affect stroma-nephron interactions through aberrant activation of Wnt signaling. a** UMAP embedding of integrated scRNA-seq cells from *Enl*-WT and T1 stroma. SP, stromal progenitor; CS, cortical stroma; Ren/Mes, renin/mesangial cells; MS, medullary stroma; prolif., proliferation; US, ureteric stroma. **b**–**d** UMAP embedding of integrated scRNA-seq (**b**) and snATAC-seq (**c**, **d**) cells from *Enl*-WT and T1 stroma. In (**c**) C0 and C5 are highlighted with black circles. **e** Schematic illustrating the spatial arrangement of *Foxd1*⁺ SP and CM-like population NP1 within the kidney. **f** snATAC data in *Enl*-WT and T1 stroma C0 are plotted as average occupancies and heatmap across the DARs between *Enl*-WT and T1 cells. The ATAC signal is normalized by RPM. All regions are defined as gained and lost. See Supplementary Data 12. **g** Dot plot showing the top 20 most significant TFs identified from the motif enrichment analysis for T1 gained or lost DARs in stroma C0. The size of the circles represents the expression level in *Enl*-WT (blue) or *Enl*-T1 (red) cells. Binomial test *p*-values are shown. FC, fold-change. Exp., expression.

**h** Scatter plot showing the expression level of the whole mouse genome in *Enl*-WT and T1 stroma C0. 28 stroma-nephron interaction-related genes implicated or predicted previously are highlighted. See Supplementary Data 13. **i** The expression level of indicated DEGs highlighted in (**h**) in *Enl*-WT and T1 stroma C0. **j** The UCell score evaluated by nephron specific β-catenin activation signature for the integrated cell types of NP1, NP2, and T1-ab within *Enl*-WT and T1 nephrons. Two-side Wilcoxon rank-sum test *p*-values are shown. **k** UMAP embedding of scRNA-seq NP1, NP2, and T1-ab cells showing the nephron specific β-catenin signature in *Enl*-WT or T1. Cells are colored by the UCell score of the signature. **l** ATAC signals at *Wnt5a* gene locus in stroma C0 from *Enl*-WT or T1. T1-gained DARs are highlighted and numbered, in which DAR containing Hox TF motif is indicated with star.
**m** Schematic illustrating the model that the nephrogenesis defects in *Enl*-T1 kidney may be partially attributed to the hyper-activation of nephrogenic β-catenin due to Hox-driven upregulation of *Wnt5a* in *Foxd1*⁺ SP.

chromatin landscape (Supplementary Fig. 9c). Interestingly, we did not observe a significant separation of these cells from *Enl*-WT counterparts based on scRNA-seq data (Fig. 6b), suggesting that chromatin changes in these cells precede overt transcriptional alterations. Differential analysis of *Foxd1*+ SP cells (C0) between T1 and WT identified 6151 T1-gained and 2496 T1-lost DARs (Fig. 6f and Supplementary Data 12). Interestingly, GREAT analysis of the gained DARs revealed an enrichment for GO terms associated with nephron and UB development (Supplementary Fig. 9d). Several well-known NPC and UB marker genes, such as *Cited1*[16] and *Ret*[69], exhibited increased chromatin accessibility (Supplementary Fig. 9e). These results suggest that *Enl*-T1 *Foxd1*+ SP cells may acquire lineage plasticity at the chromatin level. Furthermore, the DARs lost in *Enl*-T1 cells were associated with epithelial-to-mesenchymal transition (EMT) (Supplementary Fig. 9d), raising the possibility that these cells might have a compromised ability to maintain their mesenchymal state. Integrating motif and gene expression analysis, we identified *Hox9/11* TFs as top candidates that bind to T1-gained DARs and have higher expression levels in *Enl*-T1 *Foxd1*+ SP cells (Fig. 6g). Previous research has demonstrated the role of stroma-derived *Hox10* paralogs in renal morphogenesis by affecting the reciprocal interaction between *Foxd1*+ SP cells and nephrogenic mesenchyme[70]. While we found that *Hox10* genes, especially *Hoxa10* and *Hoxc10*, were highly expressed in *Enl*-T1 *Foxd1*+ SP cells, the expression levels of *Hox9/11* subfamilies were much more elevated by mutant ENL (Supplementary Fig. 9f). In line with these results, *Hoxc9* RNA ISH coupled with SIX2 protein co-staining revealed increased *Hoxc9* expression in the *Six2*-negative stroma cells at the periphery of the *Six2*+ cap mesenchyme (Fig. 4m), which likely correspond to the *Foxd1*+ stromal progenitors[68]. Altogether, these results suggest a potential role of *Hox9/11* in modulating the chromatin state of *Enl*-T1 *Foxd1*+ SP cells.

### Enl-mutant *Foxd1*+ stromal progenitors might affect stroma-nephron interactions through aberrant activation of Wnt signaling

The observed phenotypic differences between *Enl*-T1/*Wt1*GFPCre and *Enl*-T1/*Six2*GFPCre mouse models (Fig. 1 and Supplementary Fig. 2) could suggest a role for *Enl*-T1 stroma in contributing to developmental defects in the nephron. This prompted us to investigate whether *Enl*-T1 modulates the expression of genes involved in stroma-nephron interactions. We compiled a list of 28 genes previously shown or predicted to play a role in stroma-nephron interactions (Supplementary Data 13). Among them, *Pbx1*, *Meis1*, *Pdgfra*, and *Itga9* were down-regulated, while *Hoxa10*, *Hoxc10*, *Wnt5a*, and *Tcf21* were up-regulated in *Enl*-T1 *Foxd1*+ SP cells compared to *Enl*-WT (Fig. 6h, i). Notably, *Pbx1* and *Meis1* are two major co-factors of *Hox*[71]. Genetic loss of *Pbx1* in the developing kidney results in smaller kidneys with fewer nephrons and expanded mesenchymal condensates in the nephrogenic area, akin to the phenotypes observed in *Enl*-T1 kidneys. The downregulation of these co-factors, coupled with upregulation of distinct *Hox* in *Enl*-T1 stroma, may result in the dysregulation of normal transcriptional programs governed by *Hox* TFs.

Among the genes upregulated in *Enl*-T1 *Foxd1*+ SP cells, *Tcf21*[72] and *Wnt5a*[73] are involved in the Wnt signaling pathway. RNA ISH experiments confirmed the upregulation of *Wnt5a* in stroma cells at the peripheral of CM (Supplementary Fig. 9g), likely representing *Foxd1*+ SP cells[68]. Aberrant activation of the Wnt pathway has been linked to impaired nephrogenesis and the onset of Wilms tumor[20,74]. *Tcf21* can interact with β-catenin and enhance the activation of its target genes[72]. Meanwhile, *Wnt5a* could act as an upstream ligand that binds to cell surface receptors to initiate the Wnt signaling cascade[75]. *Wnt5a* is normally expressed in the MS in the kidney (Supplementary Fig. 9h)[76]. Although Wnt5a typically transmits non-canonical Wnt pathway[77,78], ectopic expression of *Wnt5a* can lead to abnormal activation of β-catenin/TCF signaling in calvarial mesenchyme in a transgenic mouse

model[79]. Thus, the upregulation of *Wnt5a* and *Tcf21* in *Enl*-T1 *Foxd1*+ SP cells could potentially result in hyperactivation of the Wnt pathway in *Foxd1*+ SP cells. To test this hypothesis, we examined the expression of previously defined lineage-specific β-catenin-induced gene signatures[76]. Interestingly, we found that the expression of a nephron-specific, but not stroma-specific, β-catenin signature was elevated in *Enl*-T1 *Foxd1*+ SP cells compared to *Enl*-WT (Supplementary Fig. 9i, j). These results suggest that *Enl*-T1 *Foxd1*+ SP cells aberrantly express a β-catenin signature typically associated with the nephron lineage.

Previous studies have demonstrated that proper activation of β-catenin in NPCs is crucial for initiating mesenchymal-to-epithelial transition (MET) during nephrogenesis[74,80], while its hyperactivation in *Six2*+ NPCs or *Foxd1*+ SP cells can lead to an accumulation of undifferentiated nephron structures and a failure of MET[76,80]. Interestingly, we observed a marked increase in the nephron-specific β-catenin target gene signature in NP1, NP2, and T1-abnormal cells in *Enl*-T1 kidneys compared to their *Enl*-WT counterparts (Fig. 6j, k). This elevation of β-catenin signature in *Enl*-T1 *Six2*+ NPCs may be linked to an intrinsic increase in *Wnt4* (Fig. 4h) and *Tcf21* (Fig. 5i) in these cells. On the other hand, given the role of Wnt5a as a paracrine signal that activates the Wnt pathway in adjacent cells and the reported potential for stroma signals to amplify Wnt/β-catenin activity in NPCs[74,75], the aberrant upregulation of *Wnt5a* in *Enl*-T1 *Foxd1*+ SP cells might also influence β-catenin activity in neighboring NPCs. Therefore, the nephrogenesis defects in *Enl*-T1 kidneys might be partially due to abnormal activation of the β-catenin/Wnt pathway in either or both the nephron and stroma (Fig. 6m).

Next, we investigated the mechanism underlying ectopic upregulation of *Wnt5a* in *Enl*-T1 *Foxd1*+ SP cells. Visualization of ATAC intensity revealed that *Enl*-T1 *Foxd1*+ SP cells gained six DARs associated with the *Wnt5a* gene locus, with two in the promoter/genebody and four in distant regulatory elements (Fig. 6l). There was a significant enrichment of *Hox9/11/12* motif sequences in all four distal DARs (Supplementary Fig. 9k), indicating a potential role of *Hox* TFs in the activation of *Wnt5a* through distal enhancers (Fig. 6m).

### Blocking the acyl-binding activity of mutant ENL compromises its function on chromatin

As a chromatin reader, ENL binds to acylated histones to regulate transcriptional processes[29,30]. Although inhibitors designed to disrupt the acyl-binding activity of WT ENL proteins have been developed[81–87], their effectiveness against ENL mutants has not been explored. Our previous work in HEK293 cells showed that *ENL* mutations found in Wilms tumor and AML (T1-T8) lead to enhanced self-association of ENL and the aberrant formation of condensates at select target genes, notably the *HOX* genes[33,34]. This leads to increased recruitment of ENL and its associated elongation factors, which in turn promotes the transcriptional elongation by RNA Polymerase II at these genes[34]. Disrupting condensate formation significantly impairs these mutants' ability to activate target genes[34]. Furthermore, we found that disrupting the acyl-binding activity of these ENL mutants (T1, T1, T3) by introducing a point mutation (Y78A) in the YEATS domain results in a loss of their gene activation function[33,34]. These data imply that blocking the reader function of ENL mutants could be a viable strategy to inhibit their function.

We have developed a potent ENL inhibitor, TDI-11055, which effectively displaces WT ENL from chromatin by competitively binding to the acyl-binding pocket and successfully blocks its oncogenic function in AML[31]. Since cancer-associated mutations do not alter ENL's acyl-binding pocket[33,34], we hypothesized that TDI-11055 could also act on these mutants. In support, isothermal titration calorimetry assays confirmed the direct binding of TDI-11055 to purified YEATS domains harboring three different Wilms-tumor associated *ENL* mutations (T1, T2, T3)[33,34], albeit with a slightly lower affinity than that observed with WT (Fig. 7a). To assess the target engagement of TDI-

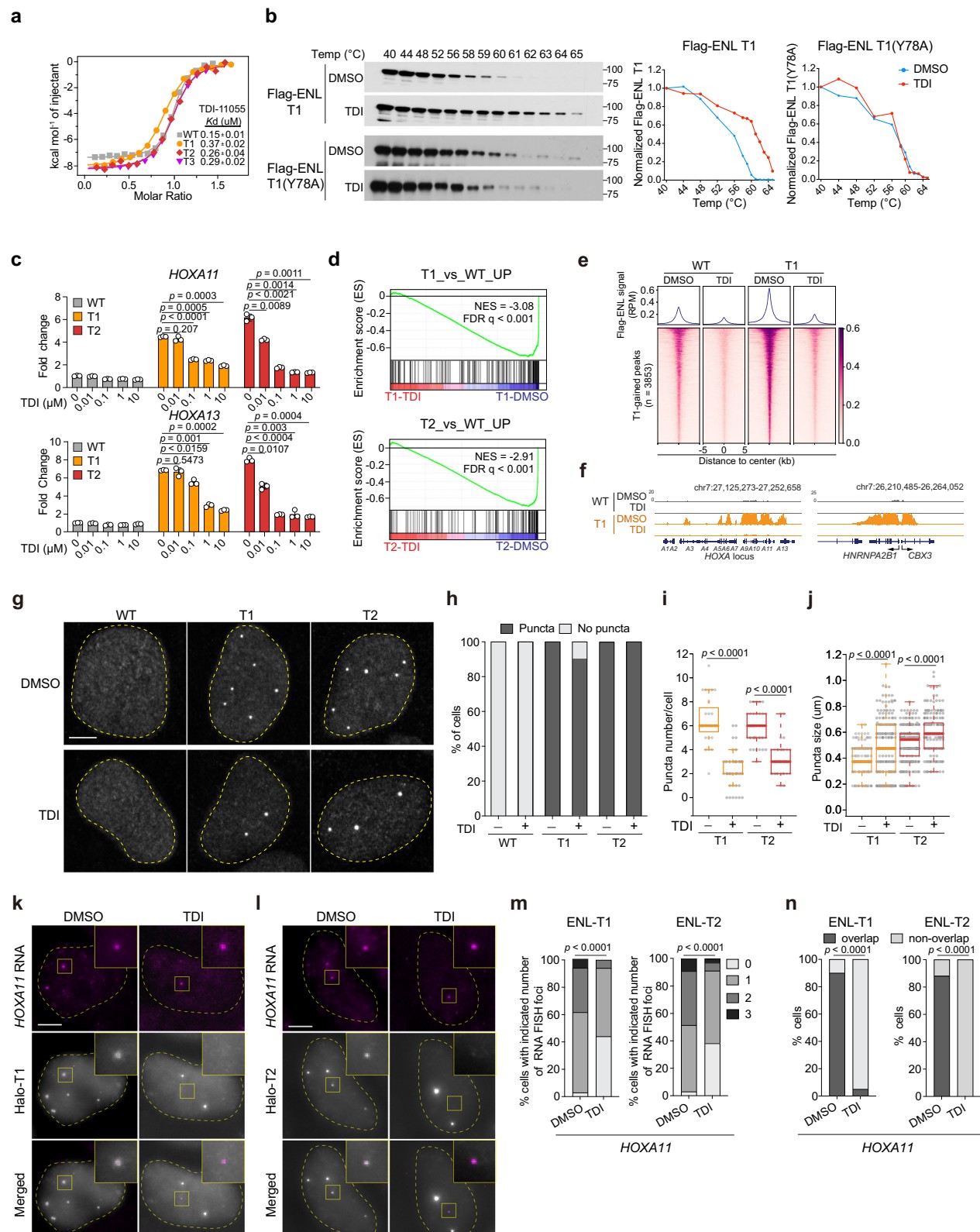

11055 in a cellular context, we performed cellular thermal shift assays[88] in HEK293 cells. TDI-11055 bound to and stabilized exogenously expressed Flag-tagged ENL-T1 and T2 proteins, but not to ENL-T1(Y78A) or ENL-T2 (Y78A) proteins that are deficient in binding acylated histones (Fig. 7b and Supplementary Fig. 10a), supporting a direct interaction of this compound with ENL-T1/T2's acyl-binding region.

Next, we evaluated the effect of TDI-11055 on the transcriptional activity of ENL mutants in HEK293 cells. Treatment with TDI-11055 for 24 h inhibited mutant ENL-induced increase in the expression of target genes *HOXA11* and *HOXA13* in a dosage-dependent manner (Fig. 7c). To assess the global impact of TDI-11055 on mutant ENL-induced transcription, we performed RNA-seq on HEK293 cells treated with either DMSO or TDI-11055 for 24 h. Transcriptional changes induced by TDI-

**Fig. 7 | Blocking the acyl-binding activity of mutant ENL via a small-molecule inhibitor compromises its function on chromatin. a** ITC assay showing TDI-11055 directly binds to ENL-WT and ENL mutants (T1-T3) YEATS domains. **b** Immunoblots and quantification showing the levels of Flag-ENL T1 and Flag-ENL T1(Y78A) after heat treatment in HEK293 cells at increasing temperatures. Temp, temperature. The experiments have been independently repeated three times with similar results. **c** mRNA expression (normalized to *GAPDH*) of *HOXA* genes in HEK293 cells expressing endogenous levels of Flag-ENL transgenes with indicated treatment. DMSO was used as the vehicle control. The experiments have been independently repeated three times with similar results. Date represent mean ± s.d.; two-tailed unpaired Student's *t*-test. **d** GSEA using custom gene set of upregulated genes in T1 or T2 vs. WT performed in cells treated with DMSO or TDI-11055. See Supplementary Data 14. **e** NGS plot and heatmap of Flag-ENL ChIP-seq signals at peaks with increased ENL-T1 occupancy in the ENL-T1 DMSO vs. ENL-WT DMSO comparison.

See Supplementary Data 16, 17. **f** The genome browser view of Flag-ENL signals at select ENL-T1 target genes under DMSO or TDI-11055 treatment in HEK293 cells. **g–j** IF staining of Flag-ENL (**g**) and quantification (**h–j**) in HEK293 cells under DMSO or TDI-11055 treatment. **h**, Percentage of nuclei with and without Flag-ENL condensates. **i** the number of condensates in each nucleus (left to right: $n = 29, 43, 44, 22$). **j** size of condensates (left to right: $n = 173, 180, 342, 210$). **i, j** Center lines indicate median and box limits are set to the 25th and 75th percentiles. Two-tailed unpaired Student's *t*-test. **k–n** Representative images of RNA-FISH (**k, l**) and quantification (**m, n**) showing the percentage of cells with indicated number of *HOXA11* nascent RNA FISH foci and the percentage of cells containing *HOXA11* nascent RNA FISH foci overlapped with Flag-ENL condensates (**n**). Two tailed Chi-square test (**m, n**). **g, k, l** Scale bar, 10 μm. Source data are provided as a Source Data file.

11055 were more pronounced in ENL mutant cells compared to WT counterparts (Supplementary Fig. 10b and Supplementary Data 14). Gene set enrichment analysis (GSEA) revealed that genes upregulated by ENL mutants were strongly suppressed by TDI-11055 (Fig. 7d and Supplementary Data 15). To investigate whether TDI-11055 inhibits mutant ENL-induced transcriptional changes by displacing it from chromatin, we performed chromatin immunoprecipitation followed by high-throughput DNA sequencing (ChIP-seq) for WT and T1 Flag-tagged-ENL in HEK293 cells treated with DMSO or TDI for 24 h. ENL-T1 exhibited increased chromatin occupancy at a subset of target genes when compared with ENL-WT (Fig. 7e and Supplementary Data 16), consistent with previous studies[33,34]. TDI-11055 treatment substantially decreased this enhanced chromatin binding by ENL-T1 at target genes (Fig. 7e and Supplementary Data 17), such as *HOXA* and *CBX3* (Fig. 7f). Together, these results demonstrate the efficacy of TDI-11055 in blocking mutant ENL' chromatin binding and transcriptional function in cells.

We previously revealed that ENL mutants form submicron-sized condensates at specific genomic targets in HEK293 cells, and these condensates are functionally required for hyperactivation of these targets, including *HOXA11/13*[34]. Interestingly, we found that TDI-11055 treatment did not abolish the formation of mutant ENL condensates (Fig. 7g, h), but rather decreased their number (Fig. 7i) and slightly increased their size (Fig. 7j). These effects phenocopy the acyl-binding defective Y78A mutation (Supplementary Fig. 10c–l). To understand how TDI-11055 impacts condensate localization to and expression of target genes, we performed IF staining for ENL mutants with concurrent nascent RNA FISH for *HOXA11* in HEK293 cells expressing Halo-tagged ENL-T1 (Fig. 7k) or T2 (Fig. 7l) proteins. TDI-11055 treatment reduced the number (Fig. 7m) and intensity (Supplementary Fig. 10m) of *HOXA11* RNA FISH foci, as well as the percentage of cells with one or more *HOXA11* FISH foci overlapping with a mutant ENL condensate (Fig. 7n). Similar changes were not observed for a negative control gene, *GAPDH* (Supplementary Fig. 10n–p). These results suggest that TDI-11055 treatment could dislodge mutant ENL condensates from genomic targets, thereby abolishing gene activation induced by these condensates. Collectively, our data indicate that targeting the acyl-binding activity of ENL mutants via small molecules can abolish their function on chromatin and gene regulation in a cellular context.

### Transient treatment with TDI-11055 rescues mutant ENL-induced developmental defects

Building on the potent effect of TDI-11055 in cell lines, we next examined its potential to rescue mutant ENL-induced defects in kidney development. We treated pregnant mice with vehicle or 100 mg/kg TDI-11055 via daily oral gavage from E10.5, when mouse nephrogenesis begins[13], to E14.5, followed by collection and histological characterization of kidneys on E15.5 (Fig. 8a). Of note, such a short-term treatment with TDI-11055 did not alter the gross morphology (Fig. 8b) nor cause noticeable changes in the histology of various nephron

structures in *Enl*-WT kidneys (Fig. 8c–g). Remarkably, TDI-11055 treatment restored the size of *Enl*-T1 kidneys to that of *Enl*-WT kidneys (Fig. 8b). Histologically, abnormal structures observed in *Enl*-T1 kidneys, such as CM-UB structures and blastema-like structures, were markedly reduced upon TDI-11055 treatment (Fig. 8c). Furthermore, TDI-11055 treatment partially rescued the decrease in differentiated nephron structures, including the Comma/S-shape bodies, glomeruli, and proximal and distal tubules (Fig. 8c–g).

To elucidate the cellular and molecular changes induced by TDI-11055 at single-cell resolution, we performed scRNA-seq on E15.5 *Enl*-T1 kidneys following treatment with TDI-11055 (Supplementary Fig. 11a, b). By integrating scRNA-seq datasets from untreated *Enl*-WT and *Enl*-T1 kidneys, along with *Enl*-T1 kidneys treated with TDI-11055, we classified all cells into four distinct lineages and assessed their distribution. TDI-11055 treatment largely reverted T1-induced alterations in the proportions of nephron and stroma cells (Supplementary Fig. 11c). Next, we identified distinct clusters in the nephron compartment representing 10 different cell types (Fig. 8h and Supplementary Fig. 11d). Compared to untreated *Enl*-T1 kidneys, TDI-11055-treated *Enl*-T1 kidneys showed a higher representation of various differentiated structures and a lower representation of T1-abnormal populations (C2 and C5) (Fig. 8i–k). Moreover, TDI-11055 treatment suppressed a significant portion of genes upregulated by mutant ENL in nephron progenitor clusters (NP1, NP2, T1-ab). Notably, this included several *Hox* genes, particularly *Hoxc* and *Hoxd* genes, as well as *Wnt4* (Fig. 8l). TDI-11055 treatment also partially reverted the increase in the percentage of *Wnt4*+ cells in *Enl*-T1 NP1 cells (Fig. 8m). In contrast, TDI-11055 treatment did not affect the expression levels of T1-upregulated genes associated with mitochondrial and metabolic pathways (Supplementary Fig. 11e). For T1-downregulated DEGs in the nephron, TDI-11055 treatment only partially restored the expression levels of a small subset of genes (Supplementary Fig. 11f). These included *Six2* and *Cited1* in *Enl*-T1 NP1 (Supplementary Fig. 11 g), suggesting that the self-renewal of *Enl*-T1 NPCs is partially recovered. Importantly, TDI-11055 treatment reduced the transcriptomic similarity of *Enl*-T1 nephron progenitor subsets to *ENL*-mutant Wilms tumor (Supplementary Fig. 11h). Furthermore, TDI-11055 treatment reduced *Wnt5a* expression levels in *Enl*-T1 *Foxd1*+ SP cells (Supplementary Fig. 11i). The β-catenin activation signature was downregulated by TDI-11055 treatment in both nephron (NP1, NP2, T1-ab) and stroma (*Foxd1*+ SP) (Supplementary Fig. 11j). Altogether, these results demonstrate that transient inhibition of the acyl-binding activity can partially rescue mutant ENL-induced transcriptomic and developmental alterations in vivo.

We next asked whether increasing the dosage of TDI-11055 or extending the treatment duration could further enhance the rescue effect. First, we administered a higher dosage of 200 mg/kg daily to pregnant mice from E10.5 to E14.5 and evaluated kidney phenotypes at E15.5. Our results showed that the 200 mg/kg dosage was as effective as 100 mg/kg in restoring kidney size and various nephron structures

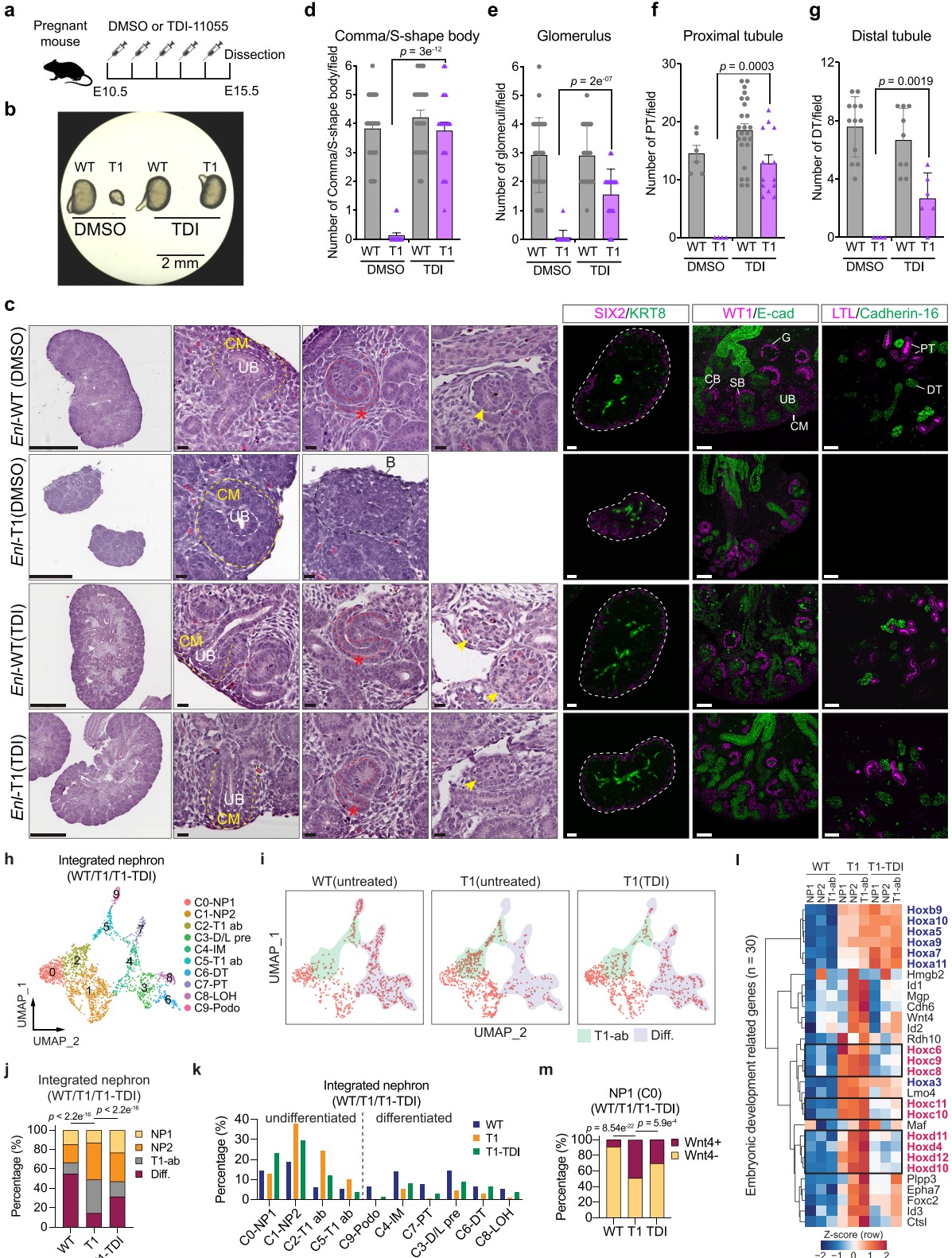

(Supplementary Fig. 12a, b), indicating a plateau in efficacy beyond a certain dosage. Additionally, we extended treatment from 4 days (E10.5-E14.5) to 8 days (E10.5-E18.5) and allowed the pups to be born (Supplementary Fig. 12c). This prolonged treatment significantly ameliorated the small kidney size, aberrant CM/UB structures, and the lack of differentiating and differentiated structures induced by *Enl*-T1 (Supplementary Fig. 12d–h), showing a slightly enhanced

rescue compared to the 4 day treatment (Fig. 8d–g). However, unexpectedly, both *Enl*-WT and *Enl*-T1 pups subjected to this extended treatment died shortly after birth. Given that TDI-11055 can inhibit the acyl-binding activity of wildtype ENL[31](Fig. 7a), the lethality observed in the *Enl*-WT pups likely indicates a previously unknown role of ENL in normal kidney development, which warrants further investigation.

**Fig. 8 | Transient treatment with TDI-11055 partially rescues mutant ENL-induced developmental and transcriptional defects in the developing kidney.**
**a** Schematic showing the experimental strategy. **b** E15.5 kidneys. Scale bar, 2 mm.
**c** Left, Histology of E15.5 kidneys as described in (**b**). The red star indicates S-shape body (SB) and the yellow arrows indicate glomerulus (G) structures. Right, Immunostaining for indicated proteins on E15.5 kidney sections. E-cad, E-cadherin. Scale bar in the first column images: H&E, 500 μm; IF, 100 μm. Scale bar in the zoom-in images, H&E, 20 μm, IF, 50 μm. **d**–**g** The number of nephron structures per field. **d** Comma/S-shape body (*n* of *Enl*-WT DMSO, *Enl*-T1 DMSO, *Enl*-WT TDI, and *Enl*-T1 TDI = 9, 10, 7, 6 kidneys); **e** glomerulus (*n* of *Enl*-WT DMSO, *Enl*-T1 DMSO, *Enl*-WT TDI, and *Enl*-T1 TDI = 7, 9, 7, 6 kidneys); **f** proximal tubule (*n* of *Enl*-WT DMSO, *Enl*-T1 DMSO, *Enl*-WT TDI, and *Enl*-T1 TDI = 4, 4, 7, 5 kidneys); **g** distal tubule (*n* of *Enl*-WT DMSO, *Enl*-T1 DMSO, *Enl*-WT TDI, and *Enl*-T1 TDI = 4, 4, 3, 3 kidneys). Dots represent

the number of indicated structure per field. Date represent mean ± s.d.; Two-tailed unpaired Student's *t*-test. **h**, **i** UMAP embedding of integrated scRNA-seq cells from *Enl*-WT, T1, and T1-TDI nephrons. **i** T1-ab clusters (C2/5) are highlighted in green; Differentiated structures (C3/4/6/7/8) are highlighted in purple. Diff., differentiated. **j** The percentage of main nephron cell types within *Enl*-WT, T1, and T1-TDI nephrons. Two-tailed Chi-Square test *p*-values are shown. **k** The percentage of all nephron clusters within *Enl*-WT, T1, and T1-TDI nephrons. **l** The expression of embryonic development-related T1-UP DEGs identified in Fig. 2k in NP1, NP2, and T1-ab cells from *Enl*-WT, T1, and T1-TDI nephrons. Gene expression is normalized by Z-score. *Hox* genes rescued by TDI-11055 treatment are highlighted in red text and those not rescued are highlighted in blue. **m** The percentage of NP1 cells with or without *Wnt4* expression in *Enl*-WT, T1, or T1-TDI nephrons. One-side Fisher's exact test *p*-values are shown. Source data are provided as a Source Data file.

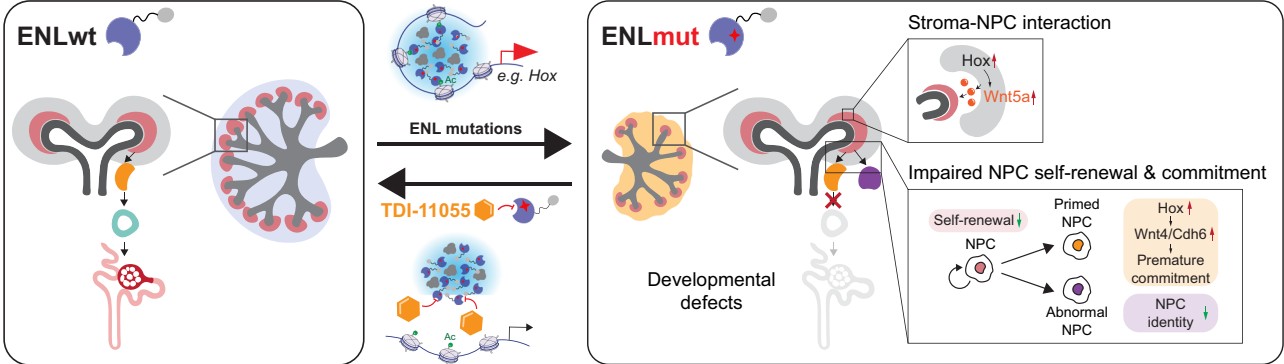

**Fig. 9 | Overall summary of the study.** Schematic showing normal nephrogenesis (left box) and that *ENL* mutation impairs kidney development trajectory by rewiring gene regulatory landscape (right box). The mutant ENL (ENLmut) disrupts kidney development by driving nephron progenitors (NPC) into a committed state while concurrently impeding their further differentiation. This dysregulation involves the misregulation of critical transcription factor regulons, particularly the *HOX* clusters. ENLmut forms transcriptional condensates at *HOX* clusters and hyperactivates *HOX* genes, which in turn, leads to increased expression of priming factors such as *Wnt4* and *Cdh6* in NPCs. Additionally, ENLmut induces the emergence of

abnormal NPCs that lose the chromatin identity typically associated with kidney development. Furthermore, ENLmut might disrupt stroma-nephron interactions through hyperactivation of paracrine Wnt5a signaling. These multifaceted effects resulting from the mutation lead to severe developmental defects in the kidney and early postnatal mortality in mice. Inhibition of the acetylation binding activity of ENLmut with a small molecule (TDI-11055) displaces ENLmut condensates from target genes and abolishes its gene activation function. This intervention effectively restores developmental defects in mice.

## Discussion

Despite the established link between disrupted development and cancer, our understanding of how specific cancer mutations impact gene regulatory landscapes to impair developmental programs in vivo remains rather limited. Our study provides insights into this fundamental question by focusing on ENL, the most frequently mutated epigenetic regulator in Wilms tumor[20]. Through the integration of genetic mouse modeling, histological characterizations, and single-cell analyses, we have uncovered the role of the *ENL*-T1 mutation in perturbing kidney development and elucidated the underlying mechanisms at single-cell resolution (Fig. 9). We have also demonstrated that transient inhibition of the chromatin reader activity of mutant ENL can effectively reverse these alterations, thus presenting a proof-of-concept for the potential use of epigenetics-targeted agents in correcting developmental defects.

Our integrated single-cell transcriptomics and chromatin accessibility profiling produced a cell atlas of mouse embryonic kidneys, elucidated key regulatory mechanisms for each cell type, and revealed dynamic changes in open chromatin accessibility during cell fate transitions. This work, in conjunction with previous studies that applied similar technologies to late-stage developing and adult kidneys, offers a comprehensive map for understanding normal kidney function and disease development. Building on this information, we demonstrated that a Wilms tumor-associated *ENL* mutation (T1) altered the cellular state and composition of the nephron. Specifically, *Enl*-T1 causes an accumulation of nephron progenitor populations and

a reduction of various differentiated cell types, indicating a block in renal differentiation. Intriguingly, *Enl*-T1 nephron progenitors (NP1) exhibit a more committed state, as evident by their chromatin accessibility landscapes and marker gene expression (e.g., downregulation of *Six2* and *Cited1* and upregulation of *Wnt4* and *Chd6*). While it has been proposed that nephron progenitors in Wilms tumor display elevated self-renewal ability[89], our findings highlight unexpected effects of cancer mutations on the balance of NPC self-renewal and commitment during nephrogenesis. Additionally, we observed the emergence of abnormal, poorly differentiated cells (T1-ab) in *Enl*-T1 nephrons with distinct transcriptomic and chromatin profiles compared to self-renewing (NP1) and committed (NP2) progenitors. These cells maintain a progenitor-like state but lose the potential for kidney specification, possibly due to a decrease in *Six2* expression and/or activity. Furthermore, compared to committed nephron progenitors (NP2), these cells exhibit a severe defect in proliferation, a property critical for NPC commitment and differentiation[13,14,90]. Future studies are warranted to investigate whether other Wilms tumor-associated mutations also induce such a de novo cell state in the developing kidney and whether these cells contribute to Wilms pathogenesis.

The developmental defects observed in *Enl*-mutant kidneys can be attributed to specific alterations in the transcriptome and chromatin accessibility. Notably, mutant ENL upregulates multiple *Hox* clusters across various cell types, including nephron progenitors (NP1, NP2, and T1-ab) and *Foxd1*⁺ stromal progenitors. This, along with our previous studies in HEK293 cells showing direct binding of ENL

mutants to *HOX* genes[33,34], establish *HOX* genes as direct targets of ENL mutants. Moreover, *Hox* genes are identified as top TF candidates responsible for the changes in chromatin accessibility induced by mutant ENL in nephron and stromal progenitors. These results suggest a critical role for *Hox* genes and their downstream targets, such as *Wnt4* and *Cdh6*, in mediating kidney phenotypes in *Enl*-T1 mice. Future studies should explore whether decreasing *Hox* gene expression in *Enl*-T1 kidneys can mitigate the nephrogenesis defects.

Among the *Hox* genes upregulated in *Enl*-T1 kidneys, the genetic loss of *Hox9/10/11* has been reported to result in lineage infidelity in the kidney[46], highlighting the role of these genes for proper lineage specification and maintenance. Our study suggests that persistent hyperactivation of certain *Hox* genes could promote immature commitment of NPCs while simultaneously restricting them from further differentiation. Importantly, *ENL*-mutant Wilms tumors express higher levels of certain *HOX* genes compared with *ENL*-WT tumors[32]. These findings suggest that maintaining proper expression levels of *HOX* genes is essential for the precise execution of kidney differentiation. Dysregulation of *HOX* genes caused by mutations in *ENL* and potentially other Wilms tumor-associated genes may contribute to Wilms tumor pathogenesis, an area that warrants further investigation.

Our study suggests a potential role for mutant ENL in perturbing the normal stroma-nephron interaction critical for nephrogenesis[17]. Specifically, *Enl*-T1 expression leads to substantial alterations in the open chromatin landscape of *Foxd1*+ stromal progenitors, which locate adjacent to the CM and are known to play a vital role in stroma-nephron interactions. Several genes involved in stroma-nephron interactions[70,72,73], such as *Hoxa10*, *Hoxc10*, *Wnt5a*, and *Tcf21*, are upregulated in *Enl*-T1 *Foxd1*+ SP cells. *Wnt5a* and *Tcf21* are linked to the Wnt signaling pathway. Proper activation of the Wnt pathway in both stroma and nephron is required for balanced NPC self-renewal and differentiation[49]. It is tempting to speculate that *Enl*-T1 *Foxd1*+ SP might contribute to the aberrant activation of the Wnt pathway in their adjacent NPCs by secreting de novo-gained *Wnt5a*, thereby influencing renal differentiation. Future studies are needed to thoroughly test this hypothesis. Interestingly, several enhancers associated with the *Wnt5a* locus display increased chromatin accessibility in *Enl*-T1 *Foxd1*+ SP cells and contain *Hoxa* binding sites, offering a potential mechanism for enhanced *Wnt5a* expression in these cells. Notably, *CTNNB1* is one of the most frequently mutated genes in Wilms tumors, and up to 50% of Wilms tumors exhibit nuclear accumulation of β-catenin indicative of constitutive activation of the β-catenin pathway[76]. Therefore, *ENL* mutations may represent a previously unknown mechanism to activate this pathway in the developing kidney. Unbiased grouping of human Wilms tumors based on their transcriptomes results in multiple clusters, with *ENL*, *CTNNB1*, and *WT1* mutations residing in similar clusters[20,32], supporting a molecular link between these proteins in kidney biology and diseases. Thus, insights gained from studying *ENL* mutations could have implications for a significant proportion of Wilms tumors.

Wilms tumors are highly heterogenous in histology and contain cells from diverse lineages, including the nephron, stroma, and muscle[5,10–12,36]. Thus, it remains a challenge to pinpoint cell types and/or developmental states that are susceptible to transformation. By utilizing the *Wt1*-cre strain to induce *Enl*-T1 expression in early progenitors that give rise to both nephron and stroma lineages, we have revealed the role of this mutant in remodeling transcriptional and chromatin accessibility landscapes within these two compartments. Future studies are needed to determine whether mutant ENL can also affect other lineages within the kidney, such as the ureteric bud/ureteric epithelium (UB/UE) and endothelial cells. Nevertheless, our study may suggest that the nephron and stroma lineages or their early progenitors are potential cellular targets for Wilms tumor-associated mutations. Lineage-specific investigations[21–23,76] of several Wilms tumor-associated mutations support this hypothesis. For instance,

gain-of-function β-catenin mutations have been detected in both the blastema and stromal components of human Wilms tumor[91]. While earlier studies suggest that β-catenin activation in the nephron is causal to Wilms tumor formation[23,92], recent work demonstrates that activation of β-catenin specifically in the stroma in mouse models non-autonomously prevents NPC differentiation and results in histological and molecular features that resemble human Wilms tumors[76]. Furthermore, the overexpression of *Lin28*, a gene encoding for an RNA-binding protein and amplified in Wilms tumor, results in Wilms tumors formation in mice only when introduced into Wt1+ early progenitors that give rise to both nephron and stroma lineages[21]. Given the histological and molecular diversity observed in Wilms tumors, it is conceivable that they differ in their developmental roots and mechanisms. As exemplified in our study of *ENL* mutations, the integration of genetic mouse modeling and single-cell technologies holds promise of elucidating these fundamental questions, thereby advancing our basic understanding of kidney development and pathogenesis at the cellular and molecular levels.

Recent genomics studies have revealed that a significant number of mutations identified in Wilms tumor impact genes involved in histone modifications and transcription elongation. These genes include *ENL (MLLT1)*, *BCOR*, *BCORL1*, *HDAC4*, *EP3OO*, *CREBBP*, *BRD7*, and *MAP3K4*[20]. Moreover, germline mutations that predispose individuals to Wilms tumor are also found in genes that converge into similar pathways, such as *CDC73* and *CTR9*[20]. Notably, ENL has been shown to interact with several of these factors, including BCOR[93], CDC73 and CTR9[94], suggesting potential functional interconnectedness. Our work showcases the significant impact of mutated epigenetic regulators on kidney development, serving as a catalyst for future investigations into other epigenetic factors. A limitation of our study, however, is that the profound development defects and early lethality caused by mutant ENL expression in *Wt1*+ precursors preclude a direct assessment of its role in Wilms tumor formation, which typically manifests months after birth. Given that *ENL* mutations often occur in Wilms tumor without other well-known genetic alterations, except in occasional instances with *CTNNB1* mutations[32], future studies should consider inducing *ENL* mutations, either alone or in combination with *CTNNB1* mutations, in a limited subset of kidney progenitors. This strategy would better recapitulate the sporadic nature of mutational events in human cancer and likely circumvent the detrimental developmental effects observed in the *Wt1*GFPCre/*Enl*-T1 model, thus allowing for a more targeted exploration of the potential role of *ENL* mutations in tumorigenesis.

Given the amenability of epigenetic mechanisms for therapeutic intervention, research in this area holds promise for uncovering novel treatment avenues. Here, we have demonstrated that a small-molecule inhibitor specifically targeting the acyl-binding activity[31] can rescue developmental and transcriptional defects induced by mutant ENL in embryonic kidneys. This serves as a compelling example of how targeting epigenetic mechanisms can mitigate developmental abnormalities caused by disease mutations. Our results also imply that inhibitors of the ENL YEATS domain may be promising therapeutic agents for treating cancers driven by these *ENL* mutations. Indeed, we have recently shown that TDI-11055 can block the onset and progression of *ENL* mutation-driven AML in mouse models[95]. Given that the acyl-binding pocket of ENL is largely unperturbed by the mutations, such inhibitors also affect the wildtype ENL. Our observation that 8 day treatment of *Enl*-WT embryos can lead to lethality suggests a critical role of wildtype ENL in early kidney development, which warrants further investigation. This finding underscores the need for a careful assessment of potential side effects of ENL YEATS inhibitors if this strategy is to be translated into clinical settings. Moreover, as exemplified by the study of EZH2 gain-of-function mutations in lymphoma[96], genetic mutations in specific genes may imply hitherto undiscovered functions of wildtype proteins in the same disease context. Future studies are needed to explore the potential involvement of wildtype

ENL in Wilms tumors lacking *ENL* mutations, as well as to explore the broader applicability of ENL inhibitors.

In summary, we have provided a direct examination of the molecular and cellular consequences of an *ENL* mutation associated with Wilms tumor during kidney development. These studies identified key epigenetic and transcriptional aberrations that disrupt cellular differentiation programs in vivo and may contribute to disease. Our work also highlights the power of combining genetic mouse models and emerging single-cell multi-omics approaches in understanding how disease mutations perturb normal development and exploring potential therapeutic avenues.

## Methods

This research complies with all relevant ethical regulations of the participating institutions that approved the study protocol.

### Mouse models

The animal protocols (#806874) were reviewed and approved by the Institutional Animal Care and Use Committees (IACUC) at the University of Pennsylvania. Mice were housed in a temperature-controlled specific-pathogen-free facility under 12 h light/dark cycles (lights on at 7:00 AM, off at 7:00 PM). Timed matings were conducted to obtain embryos of either sex at the desired developmental stages, specifically E15.5 and E18.5. Neonates <7 days of age, and embryos were euthanized via decapitation with sharp scissors. The pregnant mice were euthanized with $CO_2$. The sex of embryos and pups was not taken into consideration in the study design and therefore not determined. The *Wt1*GFPCre strain (The Jackson Laboratory #010911) contains the *Wt1*GFPCre knock-in allele, which abolishes *Wilms tumor 1* (*Wt1*) gene function and expresses an EGFPCre fusion protein directed by the *Wt1* promoter/ enhancer elements. The *Six2*GFPCre strain (kind gift from Susztak Lab) contains the *Six2*GFPCre knock-in allele, which abolishes SIX Homeobox 2 (*Six2*) gene function and expresses an EGFPCre fusion protein directed by the *Six2* promoter/enhancer elements.

The conditional knock-in mouse model (Ingenious Targeting Laboratory) was generated for ENL-T1, the most frequent *ENL* mutation found in cancer which involves the insertion of three amino acids (p.117_118insNHL) (cite), following the steps described below. Firstly, to target vector for conditional activation of the *Enl/Mllt1* mutations, we used an 8.9 kb genomic DNA sequence that was subcloned from a positively identified C57BL/6 fosmid clone (WI1-2250I3) to construct the targeting vector. The region was designed such that the long homology arm (LA) extends ~ 6.1 kb 5′ to the 5′ LoxP cassette, and the short homology arm (SA) extends about 2.1 kb 3′ to the insertion of the inversion cassette. Two mutant Lox sites (Lox71/66) were used to flank the inversion cassette containing mutant exon 4 (AACCACCTG duplication) and the flanking genomic sequences for correct splicing (Inv.saE4 * Sd), which was inserted in the reverse direction downstream of exon 4. The FRT-flanked Neo cassette was inserted immediately upstream of the inversion cassette and is 175 bp away from wildtype exon 4. The targeting region is 628 bp containing exon 4. Next, the targeting vector was linearized and then transfected into HF4 (129/ SvEv x C57Bl/6 J) (FLP Hybrid) embryonic stem cells. We selected with G418 antibiotic and identified recombinant ES clones by PCR analysis. After that, the Neo cassette was removed, and PCR and DNA sequencing confirmed the recombinant clones. Following that, we injected the correct ES clones into blastocysts and transferred the injected embryos to pseudo-pregnant recipient females. The chimeras were evaluated for germline transmission and bred to establish the knock-in mouse line.

*Wt1*-EGFPCre allele was genotyped using the following primers, which produce a 163 bp band for *Wt1* wildtype allele and a 170 bp band for *Wt1*-EGFPCre allele.

Wt1-WT F (5′-3′): CCTACCATCCGCAACCAAG
Wt1-WT R (5′-3′): CCCTGTCCGCTACTTTCAGA

Wt1-Mut F (5′-3′): ATCGCAGGAGCGGAGAAC
Wt1-Mut R (5′-3′): GAACTTCAGGGTCAGCTTGC

*Enl*-T1 allele was genotyped using the following primers, which produce a 411 bp band for *Enl* wildtype allele and a 456 bp band for *Enl*-T1 allele.

Enl-T1 primer 1 (5′-3′): CCCCAAGTCCCAGATGCTTATCTAATC
Enl-T1 primer 2 (5′-3′): ACATTGGGGAGTTCAAGGCCAGC

*Six2*-EGFPCre allele was genotyped using the following primers, which produce a 347 bp band for for *Six2*-EGFPCre allele.

Six2-1 F (5′-3′): ATGCTCATCCGGAGTTCCGTATG
Six2-2 R (5′-3′): CACCTTGTCGCCTTGCGTATAA

To confirm the expression of *Enl*-T1 in the kidney upon Cre recombinase, E15.5 *Enl*-WT and *Enl*-T1 kidneys were collected for RNA extraction with the RNeasy kit (Qiagen, #74106) and reverse transcribed with the high-capacity cDNA reverse transcription kit (Applied Biosystems, #4368814) following the manufacturer's instructions. PCR was performed using cDNA as template and following primers. Then PCR product was sent for NGS sequencing.

Enl-F (5′-3′): CGAGGAAGGTCTGCTTCAC
Enl-R (5′-3′): GTGGAATTGTGGGTAACATG

### Cell lines

In this study, HEK293 lenti-teton-3xflag-ENL-WT/T1/T1(Y78A) and HEK293 lenti-teton-3xflag-Halo-ENL stable cells were generated previously[34]. Forty-eight hours 4 ng/ml (HEK293 lenti-teton-3xflag-ENL cell lines) or 20 ng/ml (HEK293 lenti-teton-3xflag-Halo-ENL cell lines) doxycycline treatment was used to induce the transgene to express at near endogenous levels. The cells were maintained in EMEM with 10% FBS and 100 U/mL penicillin-streptomycin and were cultured in a humidified incubator at 37 °C with 5% CO2. All cell lines were mycoplasma-negative and were tested for authentication.

### H&E and immunostaining in kidneys

Kidneys were extracted at indicated time points and fixed in 10% formalin (Sigma-Aldrich, #HT5014) with gentle rocking overnight at room temperature (RT). Kidneys were then rinsed with PBS (Mediatech, #MT21-031-CV), sectioned at 4 μm thickness, and dehydrated for paraffin embedding on histological slides (Histowiz). Histology and immunohistochemistry (IHC) were performed by HistoWiz Inc. (histo-wiz.com) using a Standard Operating Procedure and fully automated workflow. Paraffin sections were rehydrated sequentially using xylene (Sigma-Aldrich, #534056), 100% ethanol (Decon Labs, #64-17-5), 95% ethanol, 70% ethanol, 60% ethanol, deionized water, and wash buffer (1 × PBS). Sections were then treated with antigen retrieval buffer (10 mM citrate acid (Sigma-Aldrich, #C1909), pH 6.0) under high pressure at 100 °C for 2–3 min. Sections were next cooled to RT, rinsed with deionized water and wash buffer, and blocked for 30 min at RT with blocking buffer (1% goat serum (CST, #5425 s) in PBS). Tissues were then incubated with primary antibodies diluted in incubation buffer (1% bovine serum albumin (Sigma-Aldrich, #A7906), 5% goat serum, 0.1% Tween-20 (Fisher Scientific, #BP337500) in PBS) overnight at 4 °C. Sections were washed with wash buffer 3 times for 5 min each and then were incubated with secondary antibody diluted in incubation buffer for 30–60 min at RT in the dark. Sections were again washed in wash buffer 3 times for 5 min each. Finally, tissues were mounted with 4′,6-diamidino-2-phenylindole (DAPI) containing mounting medium (Sigma-Aldrich, #F6057) and visualized using a confocal microscope (Zeiss, LSM880) with 10 x or 20 x objective in the Zeiss Zen Black software.

Primary antibodies (diluted with incubation buffer): WT1 (Abcam, #ab89901) diluted 1:50; E-cadherin (Fisher Scientific, #BDB610181) diluted 1:400; LTL (Vector laboratories, #B-1325) diluted 1:400; Cadherin-16 (Santa Cruz, #sc-393132) diluted 1:50; SIX2 (Proteintech, #11562-1-AP) diluted in 1:100; KRT8 (DSHB) diluted in 1:50; SLC12a3 (Abcam, #ab95302) diluted in 1:100; CDH6 (Sigma-Aldrich, #HPA007047) diluted in 1:50.

Secondary antibodies (diluted 1:200 with incubation buffer): Goat anti-Rabbit IgG (H + L) Alexa Fluor® 488 (Invitrogen, #A32732); Goat anti-Mouse IgG (H + L) Alexa Fluor® 488 (Invitrogen, #A32723); Goat anti-Rabbit IgG (H + L) Alexa Fluor® 568 (Invitrogen, #A11011); Goat anti-Mouse IgG (H + L) Alexa Fluor® 568 (Invitrogen, #A11031); Streptavidin-Alexa Fluor™ 568 conjugate (Thermofisher, #S11226).

The quantification of comma/S-shape and glomerulus was performed using the H&E staining images. The quantification of tubular structures was performed using the IF staining images.

### RNAscope in situ hybridization (ISH) and protein co-detection assay

RNAscope in situ hybridization (ISH) was conducted using RNAscope® Multiplex Fluorescent Reagent Kit v2 (Biotechne, #323280) following the manufacturer's protocol. Briefly, paraffin sections were rehydrated sequentially using fresh xylene (Sigma-Aldrich, #534056) and fresh 100% ethanol (Decon Labs, #64-17-5), followed by 10 min treatment with RNAscope® Hydrogen Peroxide at room temperature (RT) and twice wash with $H_2O$. After antigen retrieval buffer incubation at >98 °C for 15 min, sections were immediately rinsed twice with Milli-Q $H_2O$ for 2 min and ethanol for another 3 min. After drying slides, the sections were treated with protease plus at 40 °C for 30 min, then washed with fresh $H_2O$ twice. Following that, the sections were placed into a humidity control tray and incubated with pre-warmed probes in a Hybridization oven at 40 °C for 2 h and then washed twice with 1 x wash buffer for 2 min at RT. The probe signal was amplified using sequentially incubation with AMP1 for 30 min, AMP2 for 30 min, AMP3 for 15 min, HRP-C1 for 15 min, diluted TSA-647 (1:1500) for 30 min, and HRP-Blocker for 15 min. After rinsing in wash buffer and PBS, the sections were applied for protein detection as described in the method for immunostaining in kidneys. Finally, the sections were visualized using a confocal microscope (Zeiss, LSM880) with 10 x, 20 x, or 63 x objectives in the Zeiss Zen Black software.

Primary antibodies (diluted with incubation buffer): SIX2 (Proteintech, #11562-1-AP) diluted in 1:100. Secondary antibodies (diluted 1:200 with incubation buffer): Goat anti-Rabbit IgG (H + L) Alexa Fluor® 568 (Invitrogen, #A11011).

### Cellular immunofluorescence staining and confocal microscopy

Immunofluorescence staining and confocal microscopy were performed as previously described[34]. In brief, HEK293 cells stably expressing indicated ENL mutants were plated on cover slips in a 24-well plate. 4 ng/mL doxycycline was added to induce *ENL* mutant transgene expression at close to endogenous *ENL* levels. For TDI-11055 treatment, after 48 h 4 ng/mL doxycycline treatment, cells were treated with DMSO or TDI-11055 (1 µM) for another 24 h. Cells were fixed with 4% PFA in PBS for 15 min, permeabilized with 0.1% Triton-X 100 in PBS for 10 min and blocked with 10% goat serum in PBS for 30 min. Cells were then incubated with primary antibody (Flag, Sigma-Aldrich, #F1804-1MG) in the blocking buffer overnight. The next day, cells were washed 3 times with 0.5% Tween-20 in PBS. Cells were incubated with the secondary antibody (goat anti-Mouse IgG (H + L) Alexa Fluor 568 (Invitrogen, #A11031)) and washed 3 times with 0.5% Tween-20 in PBS. Cover slips were mounted using DAPI-containing mounting medium. Z-stack images were captured on a confocal microscope with 63x oil DIC objective in the Zeiss Zen Black software.

### RNA fluorescence in situ hybridization (FISH) and imaging

Briefly, HEK293 cells stably expressing Halo-tagged ENL-T1 or T2 were plated on cover slips in a 24-well plate and treated with 20 ng/mL doxycycline for 48 h followed by treatment with DMSO or TDI-11055 1uM for another 24 h. 150 nM JF549 Halo dye was then added to the media to label Halo-positive cells. Then, labeled cells were fixed and permeabilized. Following a 5 min incubation with Wash buffer A (Biosearch Technologies, #SMF-WA1- 60), cells were hybridized with

hybridization solution overnight at 37 °C. Cells were then washed with Wash buffer A (Biosearch Technologies, #SMF-WA1- 60) and Wash buffer B (Biosearch Technologies, #SMF-WB1-20) sequentially. Finally, cells were mounted with mounting medium. RNA FISH probes were labeled with Quasar 670 dye. Images were captured on a widefield Leica microscope with 63x oil objective and illuminated with a mercury lamp and standard filters for DAPI, Cy2, and Cy5.

### Image analysis

All analyses were performed in ImageJ. To measure the mean nuclear fluorescence intensity of Flag-ENL in individual cells, we included all the pixels within the cell nucleus in the single-cell image. To quantify the condensate size, we created a mask that covers all condensates in the single-cell image and extracted the fluorescence intensity of all the pixels within the condensate mask.

### Quantitative real-time PCR analysis

We performed RNA isolation and reverse transcription using RNeasy kit (Qiagen, #74106) and high-capacity cDNA reverse transcription kit (Applied Biosystems, #4368814) according to the manufacturer's instructions. SYBR Green PCR Master Mix (Fisher Scientific, #A25778) was used for quantitative real-time PCR with the ViiA 7 Real-time PCR System. The primers have been included in Supplementary Data 18.

### RNA-sequencing (RNA-seq)

We extracted total RNA using the RNeasy kit (Qiagen) and then performed library preparation with 500 ng total RNA using the poly(A) mRNA magnetic isolation module (NEB, #E7490) and RNA library kit (NEB, #7770) following the manufacturer's instructions. Samples were sequenced on an Illumina NextSeq 500. Raw reads were aligned to the human reference genome (hg19) using HISAT2 v 2.2.1 with default parameters[97]. The package featureCounts v 2.0.2 was used to count mapped reads[98]. We used transcript per million (TPM) to normalize gene expression and DESeq2 v1.38.3[99] to identify differentially expressed genes between conditions with the following criteria: adjusted *P*-value < 0.05 and fold change ≥ 1.5. Volcano plots were generated in R.

### Gene set enrichment analysis (GSEA)

GSEA was performed using the GSEA v4.1.0 software with 1,000 gene set permutations. Gene sets were manually curated from our own datasets. A detailed description of GSEA methodology and interpretation can be found at http://www.broadinstitute.org/gsea/doc/GSEAUserGuideFrame.html. We generated gene rank lists by ordering the expression fold change (T1 vs. WT and T2 vs. WT) from largest to smallest in indicated cells. Enrichment score (ES) reflects the degree to which a gene set is overrepresented at the top or bottom of a ranked list of genes. The normalized enrichment score (NES) is the enrichment score normalized across analyzed gene sets. The false discovery rate *q* value (FDR *q*-val) is the estimated probability that a gene set with a given NES represents a false positive finding. All gene sets used in this study are provided in Supplementary Information.

### ChIP-sequencing (ChIP-seq)

20-30 million cells were collected, washed, cross-linked with 1% paraformaldehyde in PBS at room temperature for 10 min, and quenched with 125 nM glycine for 5 min. Cells were resuspended and sonicated in RIPA 0.3 buffer (10 mM Tris-HCl (pH 7.4), 0.1% SDS, 1% Triton X-100, 1 mM EDTA (pH 8.0), 0.1% NaDOC, 0.3 M NaCl, 0.25 % sarkosyl, 1 mM DTT, and protease inhibitors) for 13 min. 6ug Flag antibody (Sigma-Aldrich, #F1804-1MG) was used to enrich Flag-ENL proteins. After overnight incubation at 4 °C, the samples were washed twice with low salt wash buffer (50 mM Tris pH 8.0, 150 mM NaCl, 1 mM EDTA, 796 1% Triton X-100, and 0.1% SDS), twice with high salt buffer (50 mM Tris pH 8.0, 500 mM NaCl, 1 mM EDTA, 1% Triton X-100, and 0.1% SDS), twice

with LiCl wash buffer (50 mM Tris pH 8.0, 150 mM LiCl, 1 mM EDTA, 1% NP-40, and 0.5% Na-Deoxycholate, 0.1% SDS), and once with TE buffer. Bound DNA then was eluted with ChIP elution buffer, reverse cross-linked, and treated with RNase A at 37 °C and proteinase K at 55 °C. Finally, the DNA was purified with a PCR purification kit (Qiagen, #28106). ChIP-seq libraries were constructed using the NEBNext® Ultra™ II DNA Library Prep Kit (NEB, #E7645L) according to the manufacturer's instructions. Samples were sequenced on an Illumina NextSeq 500. Raw reads were aligned to human reference genome (hg19) using the Bowtie2 (v2.2.5)[100]. Aligned BAM files were sorted using the function samtools (v1.6) module *sort*, and the corresponding index files were generated by module *index*[101]. The sorted BAM files were then performed the duplicates removing using the samtools module *rmdup*. The peak calling was performed by MACS2 (v2.2.8)[102] with the parameters "-f BAM -g hs --nomodel -p 1E-10 --broad --keep-dup all --broad-cutoff 1E-10". Peaks were annotated using the HOMER (v4.11) module 'annotatePeaks.pl'[103]. Deeptools (v3.5.2) module *bamCoverage* was used to generate the BigWig file with the parameters "-bs 10 --normalizeUsing CPM"[104]. The corresponding heatmap was plotted by Deeptools modules *computeMatrix* and *plotHeatmap*. BigWig files were uploaded to UCSC genome browser for visualization.

## Isothermal titration calorimetry (ITC) assay

We dialyzed the recombinant ENL YEATS proteins with ITC buffer (25 mM Tris, pH 7.5, 500 mM NaCl, and 5 % glycerol). Protein solutions were centrifuged to remove aggregates. Compounds were dissolved in DMSO based on provided molecular weight and weight information. The compound was further diluted to 50X assay concentration in DMSO prior to dilution into ITC buffer just prior to titration. ITC cells were rinsed with buffer and then compound at test concentration was added to the cell. Contents were pipetted up/down several times to mix with any trace buffer in the cell. A small volume of protein solution was removed from the cell for concentration re-check using Nano-Drop. In each ITC titration, the compound was added to the main solution in 20 increments with 250 s intervals between injections. Usually, compound at 150 µmol/L was added to protein solution at 20–25 µmol/L. The resultant ITC curves were processed using the Origin (v.7.0) software (OriginLab) in accordance with the "One Set of Sites" fitting model. For data analysis, we excluded data from the first injection.

## Cellular thermal shift assay

We performed the cellular thermal shift assay according to a published protocol[88]. Briefly, cells treated with DMSO or 10 µM TDI-11055 for 1 h at 37 °C were harvested, washed with PBS, and resuspended in PBS with protease inhibitor cocktail (Roche, complete tablets, EDTA-free). Cell suspensions were aliquoted equally and heated at indicated temperatures for 3 min. After being cooled at 25 °C for 3 min, the samples were lysed in lysis buffer (50 mmol/L Tris, pH 7.4, 250 mmol/L NaCl, 5 mmol/L EDTA, 50 mmol/L NaF, 1 mmol/L Na3VO4, 1% NP-40, protease inhibitor cocktail) and were subjected to 3 freeze-thaw cycles using liquid nitrogen. To collect lysates, samples were centrifuged at 15,000 × g for 20 min at 4 °C. The supernatants were transferred to a new tube and analyzed by immunoblotting.

## Single cell isolation from E15.5 kidneys for single-cell RNA sequencing

Kidneys from E15.5 *Enl*-WT, *Enl*-T1, and *Enl*-T1 with TDI-11055 treatment embryos were harvested and incubated with digestion solution containing Enzyme P, Enzyme D, and Enzyme A from the Multi Tissue Dissociation Kit 2 (Miltenyi Biotec, #130-110-203) in a gentleMACS C tube. The tissues were digested on the gentleMACS Octo Dissociator with Heaters run program 37C_Multi_E. The samples were resuspended and filtered through 70 µm strainer, followed by one wash with 15 ml PBS. After centrifugation at 300 × g for 10 min, the cell pellet was incubated with 10 volumes of 1X Red blood cell lysis solution (Miltenyi Biotec, #130-094-183) for 2 min at RT. Then, the samples were centrifuged at 300 × g for 5 min at RT and resuspended with PBS for further steps. Cell number and viability were measured on a Countess AutoCounter (Invitrogen, #C10227). The cell concentrations were $1.21 \times 10^3$/µl with 88% viability for *Enl*-WT kidneys, $2.9 \times 10^2$/µl with 85% viability for *Enl*-T1 kidneys, and $2.5 \times 10^2$/µl with 89% viability for TDI-11055 treated *Enl*-T1 kidneys.

## Single nuclei isolation from E15.5 kidneys for single-nuclei RNA sequencing

Kidneys from E15.5 *Enl*-WT and *Enl*-T1 embryos were harvested. The single nuclei isolation was performed as described in the 10 x Genomics protocol (CG000366 RevB, Protocol 2). Briefly, the kidneys were collected into a 1.5 ml microcentrifuge tube. 500 µl 0.1X chilled lysis buffer (1X lysis buffer was diluted in lysis dilution buffer. 1X lysis buffer: Tris-Hcl (pH 7.4) (Fisher Scientific, BP1531) 10 mM, NaCl (Fisher Scientific, #BP358-1) 10 mM, Mgcl₂ (Sigma-Aldrich, #M0250) 3 mM, Tween-20 0.1%, Nonidet P40 Substitute (BioVision, #2111-100) 0.1%, Digitonin (Millipore, #300410) 0.01%, BSA 1%, DTT (Sigma-Aldrich, #D9163) 1 mM, RNase inhibitor (Thermo Fisher Scientific, #N8080119) 1 U/µl; lysis dilution buffer: Tris-Hcl (pH 7.4) 10 mM, NaCl 10 mM, Mgcl₂ 3 mM, DTT 1 mM, RNase inhibitor 1 U/µl) was added into the tube and the kidneys were immediately homogenized 15 times using a Pellet Pestle. After 5 min incubation on ice, the samples were mixed with a pipette and incubated for another 10 min on ice. To stop the reaction, 500 µl chilled wash buffer (Tris-Hcl (pH 7.4) 10 mM, NaCl 10 mM, MgCl₂ 3 mM, Tween-20 0.1%, DTT 1 mM, RNase inhibitor 1 U/µl) was added. Subsequently, the samples were sequentially passed through a 70 µm and 40 µm strainer into a 2 ml tube. After centrifugation at 500 × g for 5 min at 4 °C, the nuclei pellet was washed with 1 ml chilled wash buffer twice. The pellet was resuspended in diluted nuclei buffer (10 x Genomics, #2000207) for further steps. Nuclei concentration was calculated using a Countess AutoCounter (Invitrogen, #C10227).

## Single-cell RNA sequencing (scRNA-seq)

scRNA-seq was performed with Chromium Next GEM Single Cell 3' Reagent Kit V3.1 (10x Genomics, #PN-1000269) in accordance with the manufacturer's instructions (10x Genomics, CG000388 RevB). Briefly, 10,000 live cells were loaded by targeting 6,000 cell recovery. The GEM was generated on Chromium Next GEM Chip G in the Chromium Controller. After the GEM-RT incubation and cleanup, cDNA was amplified and cleaned up. Agilent Bioanalyzer High Sensitivity DNA kit (Agilent, #5067-4626) was used for post-cDNA quality control and quantification. Next, the library was constructed according to the manufacturer's manual. Qubit was used for DNA concentration measurement and Agilent Bioanalyzer High Sensitivity DNA kit was used for library quality control according to the manufacturer's manual. The library was sequenced on an Illumina Nextseq 500 with a NextSeq 500/550 high output sequencing kit (Illumina, #20046811) using the following read length: 28 bp Read1 for DNA fragments, 10 bp i7 index for sample index, 10 bp i5 index for cell barcodes, and 60 bp Read2 for DNA fragments.

## Single nuclei ATAC sequencing (snATAC-seq)

snATAC-seq was performed with Chromium Next GEM Single Cell ATAC Reagent Kit V2 (10 x Genomics, #PN-1000406) in accordance with the manufacturer's instructions (10 x Genomics, CG000496 RevA). Briefly, 10,000 nuclei were loaded and incubated in the transposition buffer by targeting 6,000 nuclei recovery. The GEM was generated on Chromium Next GEM Chip H in the Chromium Controller. After the GEM incubation cleanup, the library was constructed according to the manufacturer's manual. Qubit was used for DNA concentration measurement and Agilent Bioanalyzer High Sensitivity DNA kit was used for library quality control according to the

manufacturer's manual. The library was sequenced on an Illumina Nextseq 2000 with a NextSeq 1000/2000 P2 sequencing kit (100 cycles) (Illumina, #20046811) using the following read length: 50 bp Read1 for DNA fragments, 8 bp i7 index for sample index, 16 bp i5 index for cell barcodes, and 50 bp Read2 for DNA fragments.

## scRNA-seq data analysis

**Date processing and quality control.** Raw fastq data were aligned to mouse reference genome mm10 using CellRanger (v6.1.2). R package Seurat (v4.3.0.1) was used for quality control, processing, and dimensional reduction analysis[105]. For quality control, low-quality cells were excluded from further analysis based on the following criteria: (1) expressed gene number was <1000 or >6000 or (2) percentage of mitochondrial counts was >10%. 5232, 4144, and 2752 high-quality cells were obtained for *Enl*-WT, T1 and T1-TDI, respectively (Supplementary Fig. 2c and 10b). The resolution of clustering was determined as the lowest score capable of distinguishing all cell types based on previous single-cell studies of the developing kidney.

**scRNA-seq datasets integration.** scRNA-seq datasets integration was performed using R package Seurat (v4.3.0.1). Briefly, individual Seurat objects that passed quality control metrics were used for the integration. *FindIntegrationAnchors* was used for identifying the anchors within the Seurat objects we aimed to integrate. *IntegrateData* function was used to generate an integrated Seurat object. Following this, a standard dimensional reduction processing was performed for the integrated object and the corresponding UMAP was obtained by running the *RunUMAP* function.

**Differentially expressed gene identification.** The R package Seurat (v4.3.0.1) function *FindMarkers* was used to identify differentially expressed genes (DEGs) with parameters "test.use = 'MAST', min.pct = 0.2, min.diff.pct = 0, logfc.threshold = 0.485". Mitochondrial and ribosomal protein genes were removed from the DEG list by excluding the gene with name containing "Rpl", "Rps" or "mt-". All up- and down-regulated DEGs were divided for further analysis.

**scRNA-seq trajectory analysis.** The R package Monocle3 (v1.4.3) was used for scRNA-seq trajectory analysis on nephron and stroma lineages[106]. *as.cell_data_set* function from R package SeuratWrapper (v0.3.1) was used to convert a Seurat object into a Monocle3 object. Cells were not re-clustered and the same embedding of UMAP was retained. For the nephron-related trajectory analysis, NP1 cell population was defined as the initial node. Cells were colored by their pseudotime score in the UMAP. An arbitrary line was drawn based on the pseudotime score across the entire dataset. For stroma-related trajectory analysis, *Foxd1*+ SP cell population served as the initial node. Cells on the UMAP were colored by the clustering and the trajectories were generated by the package.

**Gene ontology terms analysis.** GO term analysis was performed by the functional annotation tool Database for Annotation, Visualization, and Integrated Discovery (DAVID) (https://david.ncifcrf.gov/)[107]. Biological Process (BP) module was used to represent the functional relevance and *p*-values were used for evaluating significance.

**Correlation analysis between human and mouse ENL-mut signatures for Wilms tumor patients from TARGET-WT database.** R package GSVA (v3.17) was used to assess the gene set enrichment score for clinical patients[108]. 124 Wilms tumor patients from TARGET-WT database were selected for this analysis. The corresponding Transcripts per Million (TPM) files of each patient were downloaded and merged into a single input file for the further analysis. The mouse ENL-mut signature identified in this study was converted to human signature using the R package nichenetr (v3.14) function

*convert_mouse_to_human_symbols*[109]. After obtaining the matrix of the GSVA score for human and mouse ENL-mut signatures, a scatter plot was generated for visualization and the Pearson coefficient was calculated for evaluating the correlation.

**Scoring gene signatures in scRNA-seq datasets.** R package UCell (v2.4.0) was used to generate the UCell score, which evaluates the gene signature enrichment within scRNA-seq datasets[53]. Upon utilizing its function *AddModuleScore_UCell*, corresponding scores were assigned to each cell. Violin plots and UMAP were used for the visualization.

**Cell number comparison among *Enl*-WT, T1 and T1-TDI nephrons.** For the UMAP embedding in Fig. 8i, because cell numbers were not equal across *Enl*-WT, T1, and T1-TDI nephrons, we randomly extracted an equal number of 700 cells from each sample to avoid bias caused by differing dataset size. For the statistical analysis regarding the percentage of cell types within the individual samples performed in Fig. 8j, k, we retained all the cells in the analysis.

## snATAC-seq data analysis

**Date processing and quality control.** Raw fastq data were aligned to mouse genome reference mm10 by CellRanger ATAC (v2.1.0). R package Signac (v1.10.0) was used for quality control, processing, and dimensional reduction analysis. Only the cells with the following criteria were retained for further analysis: (1) fragment number ranging from 3000 to 100,000, (2) percentage of fragments in peaks > 30%, (3) blacklist ratio < 2.5%, and (4) TSS enrichment score > 2. 8546 and 8556 high-quality cells were obtained from *Enl*-WT and T1 datasets, respectively (Supplementary Fig. 4e). The resolution of clustering was determined as the lowest score capable of distinguishing all cell types based on previous single-cell studies of the developing kidney.

**Label transfer from corresponding scRNA-seq datasets.** To better interpret the identity of snATAC-seq cells, we implemented label transfer based on the scRNA-seq data from the same sample by utilizing the R package Signac (v1.10.0)[64]. The function *FindVariableFeatures* was used for determining the features from the well-annotated scRNA-seq dataset, *FindTransferAnchors* was used for identifying the anchors between these two modalities, and *TransferData* was used for performing the label transfer. The predicted identities for each snATAC-seq cell were added to the existing Signac object via the function *AddMetaData* for further analysis.

**Cell type specific ATAC signal visualization.** To derive pseudo-bulk ATAC data for distinct cell types within the snATAC-seq dataset, we used the Sinto (v0.9.0) module *filterbarcodes* to generate bam files for each cell type[64]. Subsequently, Deeptools (v3.5.2) module *bamCoverage* was used to generate the BigWig file with the parameters "-bs 10 --normalizeUsing CPM". The cell type specific BigWig files were uploaded to UCSC genome browser for visualization.

**snATAC-seq datasets integration for *Enl*-WT and T1 kidneys.** snATAC-seq datasets integration was performed by R package Signac (v1.10.0). Peaks from filtered *Enl*-WT and T1 snATAC-seq datasets were merged to make sure the common features were measured in further analysis. Based on the new peak reference, *Enl*-WT and T1 datasets were re-quantified and new Signac objects were obtained, respectively. Then the two new objects were merged using the function *merge*. The function FindIntegrationAnchors was used to find the integration anchors between two datasets with the setting "reduction = "rlsi"" and the function *IntegrateEmbeddings* was used to integrate the embeddings based on the anchors identified previously. Finally, *RunUMAP* was used to generate the new UMAP for the integrated datasets based on the integrated embedding.

**snATAC-seq trajectory analysis.** snATAC-seq trajectory analysis performed using R package Cicero, an extension of Monocle3[110]. The SeuratWrapper (v0.3.1) function *as.cell_data_set* was used to convert the Signac object to the required format. The subsequent steps were same as the scRNA-seq trajectory analysis.

**Identification of differentially accessible regions (DARs).** Here we used a method similar with one previously reported by Miao et al. [60] For the comparison between two cell types from *Enl*-WT and T1 individually, downsampling was conducted for the raw fastq file of *Enl*-WT by the Seqtk (v1.3) module *sample* with a scale factor of 0.678. This was done to ensure the equality of the fragments per cell between these two datasets, thus avoiding the potential bias in further analysis. For comparisons within the same snATAC-seq dataset, no additional operations were needed.

After obtaining the cell-type specific pseudo-bulk ATAC bam files as described previously, peak calling was performed for both cell types by MACS2 (v2.2.8) with the parameters "-g mm --nomodel --keep-dup all --shift -100 --extsize 200 -q 1e-2 --call-summits". Peaks from the two different cell types were merged using the Bedtools (v2.26.0) module *merge*[111]. scATAC-pro (v1.5.0) module *reConstMtx* was used to generate the new matrixes with the common peaks for both cell types[112]. Considering the binary nature of the snATAC-seq peak matrix, Fisher's exact test was performed to identify the DARs with the significance threshold of adjusted *p*-value (Benjamini–Hochberg correction) <0.05. The corresponding heatmap was plotted by Deeptools (v3.5.2) modules *computeMatrix* and *plotHeatmap*.

**Motif enrichment analysis.** HOMER (v4.11) module *findMotifsGenome.pl* was used to perform the motif enrichment analysis with the setting "-size given". The module *annotatedPeaks.pl* was used to obtain the precise coordinates of specific motif sequence generated by *findMotifsGenome.pl*.

**chromVAR TF enrichment analysis for *Enl*-WT nephron.** *Enl*-WT nephron cells were categorized into three different lineages: Podo, PT, and DT-LOH. The Podo lineage comprises cells from NP1/2 and Podo populations, the PT lineage includes cells from NP1/2, IM, and PT populations, and the DT-LOH lineage contains cells from NP1/2, IM, D/L pre, DT and LOH populations. Cells were ranked by the pseudotime score obtained from the trajectory analysis for each lineage. R package Signac (v1.10.0) function *RunChromVAR* was utilized to calculate the TF enrichment score. Line plots were generated for visualization.

**Chromatin distribution analysis**
HOMER (v4.11) module *annotatePeaks.pl* was used to perform the chromatin distribution analysis. Among the obtained genomic annotations, "exon", "intron" and "TES" were grouped together as "genebody", while all other annotations were combined as "Non-coding" except for "Promoter-TSS" and "intergenic".

**The Genomic Regions Enrichment of Annotations Tool (GREAT) analysis**
The GREAT website tool (http://bejerano.stanford.edu/great/public/html) was used to conduct the analysis with setting the "Whole genome" as the background regions[113].

**Statistics and reproducibility**
Chi-Square test was used for the quantification of cell populations. Wilcoxon rank-sum test was used for comparing the signature score between different samples. Fisher's exact test was used for the gene expressing percentage of specific population and the DAR identification. Experimental data is presented as mean ± s.d., unless stated otherwise. Statistical significance was calculated by two-tailed, unpaired or paired Student's *t*-test on two experimental conditions with $P < 0.05$ considered statistically significant unless stated otherwise. The number of replicates and the statistical test used were indicated in the corresponding figure legends. No data were excluded from the analyses. No statistical methods were used to predetermine sample size. The experiments were not randomized. The investigators were not blinded to allocation during experiments and outcome assessment.

**Reporting summary**
Further information on research design is available in the Nature Portfolio Reporting Summary linked to this article.

## Data availability

Raw data, processed data, and metadata from mouse single-cell datasets have been deposited in GEO with the accession number GSE243868 and GSE243870. The ChIP-seq and RNA-seq data have been deposited in the Gene Expression Omnibus database under accession numbers GSE243866 and GSE243867. All other raw data generated or analyzed during this study are included in this published article (and its Supplementary Information files). Source data are provided with this paper.

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

## Acknowledgements

The authors thank members of the Wan laboratory for scientific input throughout the study; Dr. Kai Tan for critical reading of the manuscript; Dr. Xianyang Deng for scientific input; Dr. Yuting Guan and Dr. Yuka Sakata for technical support; Krista A. Budinich, Sylvia Tang, and Chujie Gong for manuscript editing; the CDB Microscopy Core and ULAR facility at the University of Pennsylvania for facility support. The research was supported by funds from the University of Pennsylvania (L.W.), a NIH Director's New Innovator Award (1DP2HG012443 to L.W.), a NIH Pathway to Independence Award (R00CA226399 to L.W.), and a Pew-Stewart Scholar Award (L.W.). L.W. is a scholar of The Leukemia & Lymphoma Society.

## Author contributions

L.W., L.S., and Q.L. conceived and designed the overall study. L.S., L.X., A.E.S., and Q.L. performed mouse works; L.S. performed cellular studies; Q.L. performed bioinformatic data analysis. Y.L. and H.L. performed ITC assay. H.W. and Q.Q. provided support for scRNA-seq and snATAC-seq studies; L.W., L.S., and Q.L. wrote the manuscript with critical input from K.Sasaki, H.W., K.Susztak, and Q.Q. L.W. supervised the overall study.

## Competing interests

L.W. is a co-inventor on a patent filed (US No. 62/949,160) related to the inhibitor used in this manuscript and is a consultant for Bridge Medicines. All other authors declare no competing interests.
