## [Peer review file · Nature Communications]

Single-Cell multiomics reveals ENL mutation perturbs kidney developmental trajectory by rewiring gene regulatory landscapeREVIEWER COMMENTS

Reviewer #1 (Remarks to the Author):

Using a combination of scRNA-seq and scATAC-seq, the authors showed that ENL mutations in mice result in partial commitment of nephron progenitors while preventing their further differentiation, possibly through Hox gene up-regulation. In addition, increased Wnt5a in stromal progenitors may partially contribute to the phenotype through paracrine activation of Wnt signaling in nephron progenitors. The bioinformatics analyses presented are of a high standard and have led to the identification of many features of this mutant mouse strain that will contribute to the understanding of the role of the ENL protein in kidney development. In addition, the authors treated the mutant mice with an ENL inhibitor to show the alleviation of the phenotype. On the other hand, weaknesses of the manuscript include the lack of histologic evidence for the gene alterations and functional in vivo validation of the hypothesis proposed by the bioinformatic analyses. In addition, there is no Wilms' tumor formation in the mutant mice, suggesting that additional molecular events are required for tumorigenesis.

Major comments

1. The bioinformatics analyses suggested upregulation of Hox clusters in nephron and stromal progenitors, increase of Wnt4 and Chd6 in nephron progenitors, and increase of Wnt5a in stromal progenitors. While these findings are potentially interesting, in situ hybridization or immunostaining of these genes should be presented to support the main conclusions of the manuscript.
2. Are such gene alterations also observed in human Wilms' tumors? Although Fig. 2l&m seems to show the clinical relevance, the coefficient is only 0.418. Alterations in representative genes found in the mutant mice (e.g. Hox genes, Wnt4, Cdh6, Wnt5a) should be presented in the human setting. Authors may soften their claim depending on the data.

3. For functional validation, the authors can test whether upregulation of Hox clusters in vivo leads to the similar phenotype and/or whether deletion of some Hox genes in ENL mutant mice alleviates the phenotype. The authors can provide such data if available. At the very least, the authors should discuss such experiments in the future.
4. How does the ENL mutation lead to the upregulation of Hox clusters? The authors should at least discuss the underlying mechanisms.
5. Treatment with the ENL inhibitor is promising, but the authors only treated the mice until E15.5. Can longer treatment alleviate the kidney defects and improve postnatal survival? If so, the impact of the treatment will be much greater. If not, the limitations should be described.
6. There is no Wilms' tumor formation in the mutant mice, suggesting that additional molecular events are required for tumorigenesis. Is there evidence that the blastema-like cells (or progenitor cluster T1 as identified in the UMAP plots) ultimately contribute to Wilms' tumor formation? If not, these limitations should be clearly described.
7. I suggest adding a paragraph summarizing the limitations of the study to avoid misunderstanding by the readers. Even with such a paragraph, I still believe that this study will contribute to the understanding of the role of ENL protein in kidney development and eventually the pathogenesis of Wilms' tumor.

Reviewer #2 (Remarks to the Author):

The authors have sought to examine a dominant allele of ENL to explore the link of ENL mutations with Wilms Tumor. The work is of a generally high quality. However, there is a major problem with the study and the impeded development is not necessarily the simplest expectation of a WT model in which massive kidney growth is the oncogenic outcome?

The major issue is that all the analyses were performed relatively speaking a long time after the initiation of the phenotype. By the severity of the observed phenotype, I expect that there will be a phenotype evident at E12.5 but much of the analysis compares wild-type and mutants at E15.5 and E18.5. This leaves open secondary measurement of all the parameters measured. This is a substantial issue given the tight linkage of regulation amongst multiple

cell types within the nephrogenic niche. Further, there may be evidence suggesting problems within cells – is an enhanced mitochondrial signature an indication of low quality RNA in the NP single cell profiling that might suggest “sick” cells?

Fortunately, there are two steps to rectifying this. First, determine when a phenotype is first observed, then repeat analyzes at this time. This would be expected to give the clearest insight into the initiating steps and allow a clearer interpretation of data in hand. Second, the authors show a remarkable rescue of the kidney phenotype treating with TDI-11055. The authors could also stop administration of the drug at e14.5 and examine transcription associated cell properties sometime later determined by preliminary study – this withdrawal would also give a likely much better handle on the events at play given many more nephrogenic niches to examine at this later timepoint than an earlier one. But, the earlier 11.5/12.5 (no drug) and a potentially 15.5 drug withdrawal at e14.5 study would be complementary.

There is also a concern in the genetic set up of the model which unfortunately removes a copy of a WT associated gene (WT1) and the potential for a synthetic interaction that cannot be controlled for in this model. Given the dominant effects of the modified ENL allele, the authors could cross to the Six2-TGC line available from the JAX labs (IMSR_JAX:009606) to generally see if the same outcomes are observed? This line is only active in the nephron lineage so it would also provide additional information not possible with the WT1 cre which is active in both interstitial and nephron lineages, on the lineage outcomes for ENL allele. Again, I am not recommending an extensive analysis for this but it is not too difficult or time-consuming to gather important new insight that may allay concerns with the WT1-cre strategy?

Reviewer #3 (Remarks to the Author):

In this manuscript, Song et al describe the development and characterization of a new genetic mouse model of a Wilms tumor-associated ENL mutation in kidney development. The authors utilized Wt1-Cre to drive the expression of the ENL T1 mutant and observed that the mutant

mice displayed severe kidney developmental defects and early postnatal death. They further carried out scRNA-seq and snATAC-seq analyses in E15.5 kidneys and found that the ENL mutation disrupted the kidney development trajectory, by rewiring gene regulatory landscape. Mutant ENL was found to influence nephron progenitor commitment and differentiation by dysregulating key transcription factors, including Hox genes. Additionally, ENL T1 was implicated in disrupting the normal stroma-nephron interaction critical for nephrogenesis, primarily through hyperactivation of the Wnt signaling. Notably, transient inhibition of the reader function of the ENL mutant with TDI-11055 partially restored kidney developmental defects and gene expression patterns, providing a proof-of-concept for epigenetic therapy. Overall, this is a comprehensive study providing new insights into how ENL mutation disrupts kidney development at single-cell resolution. Several concerns and suggestions are outlined below to strengthen its scientific rigor and clarity of the study.

Major concerns:

- 1.** This study centers on a new genetic mouse model but validation seems limited. It would greatly strengthen this study if the authors could provide more rigorous data validating the mouse model. For example, confirmation of mutant ENL expression upon Cre-mediated recombination, and comparison of the protein level of mutant ENL versus endogenous wildtype ENL.
- 2.** WT1 is a critical regulator of kidney development. In the mouse model used in this study, the presence of *Wt1*-GFPCre leads to concomitant inactivation of a copy of the endogenous *Wt1*. Although the authors claim that heterozygosity of *Wt1* does not cause any phenotypic defects in kidney development, one would expect some epigenetic and gene expression changes caused by *Wt1* heterozygosity. Therefore, it is necessary to include wildtype kidney samples in Figure 2 for direct comparison.
- 3.** Another issue of the mouse model is a possible synergistic effect of *Enl* mutation and *Wt1* heterozygosity, whereas in human patients, ENL mutations are not found to be concurrent with WT1 mutations. Thus, it is critical to validate some major phenotypic findings using another kidney specific Cre.

4. The scRNA-seq and snATAC data are beautiful but do not tell what the direct effects of ENL mutation are. Experimental validation is needed to distinguish direct and indirect effects of ENL mutations, for example, by performing CUT&RUN or CUT&Tag experiments for wild-type and mutant ENL.

5. It is impressive that transient inhibition of ENL mutant by TDI-11055 can partially rescue mutant ENL-induced transcriptomic and developmental alterations in vivo. Have the authors tried extending the treatment to a later stage in embryonic development (e.g. E18.5)? That would allow a better evaluation of the restoration of nephrogenesis. It would be interesting to see whether the neonatal lethality of Enl T1 can be partially rescued.

Minor issues:

1. Some Enl-T1 kidneys exhibited structures that resemble undifferentiated blastema components seen in Wilms tumors. Are they proliferative? The authors can perform vimentin and Ki-67 staining to determine their blastemal identity and proliferative status.

2. Are transcripts of wild-type and mutant ENL detectable in the scRNA-seq datasets? It is important to know in which cell lineages the T1 mutant is expressed and whether the wild-type and mutant ENL are expressed in the same sets of cells.

3. Based on the marker genes identified in scRNA-seq, is it possible to perform IHC staining to locate the abnormal NP1, NP2, and T1-ab cells in developing nephrons and kidneys?

4. The human ENL-mutant_Up genes were expressed at higher levels in mutant NP1, NP2 and T1-ab cells. How about their expression in mutant stromal progenitors?

5. Fig. 5i and 5l: consider adding gene expression in wt NP1 and NP2 for comparison.

6. Fig. 7l-m show that the numbers of HOXA11 nascent RNA FISH foci in mutant ENL expressing cells were reduced upon TDI-11055 treatment. How about the RNA FISH intensity of HOXA11 foci?

7. How does TDI-11055 treatment affect the percentage of cells in different lineages? Please provide a stacked bar plot, similar to Fig. 2b, to compare cellular compositions in WT, T1 and T1-TDI.

Reviewer #4 (Remarks to the Author):

Summary:

The authors have investigated the cellular and molecular mechanisms of the most common pediatric kidney tumor, Wilms tumor. Focusing on the epigenetic reader protein ENL, the most frequently mutated epigenetic regulator in Wilms tumors, the authors have used a combination of genetic mouse models, histology, scRNA and snATAC techniques to gain insights into the effects of ENL mutation in the context of Wilms tumors.

The authors use single cell transcriptomics in developing mouse embryos from their Enl-T1 mutant mice, to show that Enl mutation significantly affects the composition of nephrons, disrupts normal nephrogenesis, and results in abnormal undifferentiated progenitors. Their finding that Hoxa/c/d family genes are upregulated in Enl-T1 is clinically important as ENL mutant Wilms tumors in patients also show increased expression of Hox genes. One of the key novel findings was obtained by utilizing snATAC-seq and shows that there are dynamic changes in chromatin accessibility in the Enl mutant kidney vs. developing mouse kidney. Additionally, the authors show that the Enl T1 cells may have substantial chromatin alterations before transcriptional changes, thus shedding light on aspects that would otherwise be undiscoverable via scRNA-seq.

The changes in NPC commitment at various cell fate transitions and the altered differentiation trajectories comprise some interesting results. Furthermore, the authors showed that the mutant Enl induced a cluster of abnormal progenitor state with a loss of nephron chromatin identity. Particularly fascinating was the gain of potential to differentiate into other organs like mammary gland and heart, which could have implications for other related cell fate

decision fields. Finally, the authors utilized their in house small molecule Enl inhibitor TDI-11055, and showed that inhibition of chromatin function of mutant ENL can reverse its effects, which is fantastic step forward in the field of epigenetic therapies.

Overall this is a well conducted study that utilizes a wide array of techniques and technologies to tackle and provide a solution to a clinically important problem. The novel and significant results gained here will be valuable both for the advancement of knowledge in the field and in the clinical setting. However, there are several important concerns that should be resolved prior to publication.

Concerns and suggestions:

1. My major concern is the section about stroma-nephron interactions by aberrant activation of Wnt signaling. The authors show increased beta-catenin signature in nephron lineage, and increased Wnt5 in the stromal progenitors. However, a paracrine signaling connection between these two has not been shown/established in this study/system (line 549, 559). Given that Wnt4 is upregulated in the Enl-T1 NP1 and NP2 cells, it could be possible that Wnt4 increase has caused the increase in beta-catenin in an autocrine manner. There is also an established link between Tcf21 and beta-catenin in the literature, which could have caused the increased beta catenin signature.. Therefore, direct evidence such as Wnt5a from stromal cells binding to Fzd/LRP6 on the nephron lineage cells, or FRET/BRET with tagged Wnt5a and its receptors, would be necessary to conclusively show the interaction between stroma and nephrons. Discussion (page 73, paragraph 1) would need to be updated accordingly.

2. The authors have used the Wt1 promoter for generating the Enl mutant mice. Wt1 is primarily expressed in nephron and stroma. Could the authors comment on the possibility of Wt1 expression bias in inferences made in the downstream studies, which would be limited to the specific cell types where Wt1 Cre is expressed? For example, in line 342-344: the authors' argument is "the Enl-T1 kidney exhibited a reduction in specific nephron clusters and an increase in certain stromal clusters, emphasizing the influence of mutant Enl on nephron and stromal" However, the T1 mutation primarily only effects these 2 lineages, so it is difficult to say how much of the effects are from the mutation being limited to the 2 lineages vs. actually only affecting those 2 lineages. As such, it would be important to point out the bias of

Wt1/T1 in the downstream results and interpret the results accordingly throughout the manuscript.

3. The authors show that treatment with TDI partially reverted the increase of Wnt4+ cells (line 648), and partially restored expression levels of a small subset of the downregulated DEGs (line 651). I am curious as to what would be the effects of other lower/higher dosages of TDI? Would a higher concentration perform better or does the rescuing effects of TDI plateau at 100 mg/kg?

Additionally, if Wnt signaling is involved in mutant Enl induced defects, could the authors allude to the use of Wnt inhibitors like Dkk1/SOST for rescuing the effects of mutant Enl ?

4. Line 320-321: Could the authors speculate why the transition from NP2 to podocytes showed loss in chromatin accessibility, and its implications for their study? The podocyte cluster is mostly WT sample (Figure 3d), which may have interesting associations, for .e.g Could the T1-ab cells in C4/C10 be an intermediate to Podocytes (Figure 3c, Figure 5a,b)?

5. The results involving self-renewal, cell fate commitment, premature commitment vs differentiation block of NPCs, etc. can be difficult to track in terms of which data modality points to what result. Perhaps the authors could include schematics to summarize their results in Figure 2, 3, and 4, along with the data utilized to support it. Something like a detailed version of Fig 5m, 5n projected on the UMAPs or a block diagram like Fig S5c with details of genes regulating the cell fates and differentiation would be great in conveying the results clearly.

6. Line 222-223: The authors chose to focus on cluster 4, citing that cluster 10 has limited WT cells for comparison. However, from Figure 2e, it would appear that the percentage of WT vs T1 samples in C4 and C10 are similar. Could the authors include the number of cells overlapping between the WT and T1 samples in C4 and C10, and elaborate on the reasons for choosing C4?

7. Figure 2k: Cluster 7 is classified as NP2, and Wnt4 is expressed in NP2s. However, Wnt4 is not expressed in C7 like it is expressed in the other 2 NP2 clusters 1 and 5 (Figure 2k). Could the authors elaborate on why this might be? Is C7 different in some way?

8. Line 464, Figure 5j: There is a lot of ear morphogenesis and multicellular organism development pathways also – could the authors explain why this might be?

9. Line 621, 613: The dislodging of condensates in the ENL mutants upon TDI treatment is a fascinating finding. Although, I am curious how does that explain the increase in size or decrease in number of the condensates in Enl mutant? Could the authors elaborate on that?

10. Overall, I would urge the authors to use caution when interpreting results from pseudotime analysis. Without ground truth lineage tracing data to support these inferences, the results must be reported with care.

Minor:

1. Line 287-289: Could the authors include a p-value for the GSVA correlation?

2. Paragraph from line 261-271 seems out of place, and could be confusing as to why it is there. Perhaps it could be moved to the end of that section, after line 295.

3. Line 453 – should it more specifically be “T1-ab lost DARs (vs. NP1)?

4. Line 571, Figure 6m: p-values appear to be missing in the table?

Reviewer #5 (Remarks to the Author):

I co-reviewed this manuscript with one of the reviewers who provided the listed reports as part of the Nature Communications initiative to facilitate training in peer review and appropriate recognition for co-reviewers.

Responses to Reviewers' comments

NCOMMS-23-52570-T

"Single-Cell multiomics reveals ENL mutation perturbs kidney developmental trajectory by rewiring gene regulatory landscape "

Overall summary of the revision

We thank all reviewers for their insightful comments on our initial submission. We are pleased to read their positive remarks about the quality, significance, and novelty of our work, and greatly appreciate the constructive critiques and suggestions to further improve the study. Based on the collective feedback, we have revised the manuscript with additional data and text revision. In addition to many changes made to address specific comments, we highlight here responses to several major comments:

1. Validation of expression changes of key mutant ENL-regulated genes *in situ*.

- To validate key transcriptional changes revealed by scRNA-seq, we have performed RNA in situ hybridization (ISH) and/or protein immunofluorescence staining for several key targets. Our data confirmed the upregulation of Hox genes [e.g., Hoxc9 (Figure 4m)], Wnt4 (Figure 4n and Figure S7i), and CDH6 (Figure S7j) in specific cell types in Enl-T1 kidneys.
- We performed RT-qPCR analysis of whole kidneys and consistently observed higher expression levels of multiple Hox genes, Wnt4, and Wnt5a in Enl-T1 kidneys when compared with age-matched Enl-WT counterparts (Figure S4g and Figure R1, R2).

These results support our findings from scRNA-seq and highlight a potentially important role of these key genes in mediating Enl-T1-induced phenotypes in the kidney.

- #### 2. Evaluation of the impact of mutant ENL on kidney development using an additional cre strain. To address the concern that the phenotype observed in $Wt1^{GFPcre/+}/Enl^{flox-T1/+}$ mice may result from the synergistic effect of losing one $Wt1$ allele and expressing Enl-T1, we crossed $Enl^{flox-T1/+}$ mice with $Six2^{GFPcre/+}$ ($Six2$ -TGC) mice. This strategy allows the expression of Enl-T1 specifically in $Six2^+$ nephron progenitor cells (NPCs) and their progeny, which are confined to the nephron lineage. Our results (Figure S2) showed that similar to $Wt1^{GFPcre/+}/Enl^{flox-T1/+}$ mice, the $Six2^{GFPcre/+}/Enl^{flox-T1/+}$ mice also experienced early postnatal mortality, suggesting developmental defects and alleviating the concern that the function of Enl-T1 is contingent solely on losing one $Wt1$ allele. Interestingly, closer inspection revealed that kidney abnormalities in the $Six2$ -TGC model were milder compared to the $Wt1$ -GC model. Given that $Wt1^+$ precursor cells give rise to both nephron and stroma lineages, whereas $Six2^+$ progenitors give rise to only the nephron, these results suggest that Enl-T1 expression in the stroma may also contribute to nephrogenesis defects in $Wt1^{GFPcre/+}/Enl^{flox-T1/+}$ mice. This is corroborated by scRNA-seq and scATAC-seq data, which revealed significant gene expression and chromatin accessibility changes in Enl-T1-expressing stroma cells in $Wt1^{GFPcre/+}/Enl^{flox-T1/+}$ kidneys (Figure 6). Together, these results addressed the concern surrounding $Wt1$ heterozygosity and provide new insights into lineage effect of the Enl mutation.

3. Assess the impact of extended or increased dosage of TDI-11055 treatment on ENL-T1- induced kidney abnormalities:

- We extended the TDI-11055 treatment duration from 4 days (E10.5-E14.5) to 8 days (E10.5-E18.5) and allowed the pups to be born (Figure R4a). The extended 8-day treatment significantly ameliorated the small kidney size, aberrant CM/UB structures, and the lack of differentiating and differentiated structures induced by Enl-T1, similar to the 4-day treatment (Figure R4b-f). However, we found that both Enl-WT and T1 pups died shortly after birth at P0.5. These data, along with our unpublished data from an Enl-knockout mouse model, reveal a previously unknown role of the wildtype ENL in normal kidney development that we are actively investigating in the lab. With these new insights, the inability of the 8-day treatment to improve survival in Enl-T1 heterozygous mice could be due to adverse effects on the wildtype Enl allele, an incomplete blockage of Enl-T1 function, or a combination of both. Despite this, the treatment's substantial mitigation of Enl-T1-induced abnormalities and gene expression changes underscores its potential for treating Enl-mutated cancers.

- Exploring the potential of a higher TDI-11055 dosage to further improve the rescue effect, we increased the dose from 100 mg/kg to 200 mg/kg (E10.5-E14.5) and evaluated kidney phenotypes at E15.5. Our results showed that 200mg/kg was as effective as 100 mg/kg in restoring kidney size and various nephron structures (Figure R9), suggesting a plateau in the drug's efficacy beyond a certain dosage.

All together, we are confident that the inclusion of these new data, along with other specific revisions described in point-by-point response to reviewers' comments, has enhanced the overall quality and impact of our study. Our work provides new insights into how mutations in the epigenetic regulator ENL alter the gene regulatory landscape to disrupt developmental programs at single-cell resolution *in vivo*. It also offers a proof-of-concept for the use of epigenetics-targeted agents to rectify developmental defects.

Key to the rebuttal

- Reviewers' comments: *black italic text*
- Our responses: blue text
- New data figure citations: red text
- Previous data figure citations: blue text
- Reviewer only figures ("Figure RX"): purple text
- In the revised manuscript, sections revised or added in response to reviewers' comments are in red font.

Reviewer #1 (Remarks to the Author)

Using a combination of scRNA-seq and scATAC-seq, the authors showed that ENL mutations in mice result in partial commitment of nephron progenitors while preventing their further differentiation, possibly through Hox gene up-regulation. In addition, increased Wnt5a in stromal progenitors may partially contribute to the phenotype through paracrine activation of Wnt signaling in nephron progenitors. The bioinformatics analyses presented are of a high standard and have led to the identification of many features of this mutant mouse strain that will contribute to the understanding of the role of the ENL protein in kidney development. In addition, the authors treated the mutant mice with an ENL inhibitor to show the alleviation of the phenotype. On the other hand, weaknesses of the manuscript include the lack of histologic evidence for the gene alterations and functional *in vivo* validation of the hypothesis proposed by the bioinformatic analyses. In addition, there is no Wilms' tumor formation in the mutant mice, suggesting that additional molecular events are required for tumorigenesis.

We are grateful for the reviewer's recognition of our work's quality and its contribution to understanding ENL's role in kidney development. We also appreciate the constructive feedback and suggestions provided for enhancing our study.

In response to the concerns about the lack of histologic evidence for gene alterations and the need for functional *in vivo* validation of our bioinformatic analysis, we have performed *in situ* hybridization (ISH) and/or immunostaining of key genes identified by scRNA-seq. Results from these new experiments corroborate the upregulation of key genes such as *Hox* (Figure S4g and Figure 4m), *Wnt4* (Figure 4n and Figure S7i), and *cDH6* (Figure S7j) in *Enl-T1* kidneys (see response to Major Comment #1).

Regarding the absence of Wilms tumor in the *Wt1^{GFPcrei+}/Enl^{lox-T1/+}* mice, there could be several contributing factors. One is that the profound developmental defects and early lethality resulting from *Enl-T1* expression in *Wt1⁺* precursors preclude a direct assessment of its role in Wilms tumor formation, which typically manifests months after birth. Another factor is the potential need for additional molecular events, as suggested by this Reviewer. To address this point, we examined published sequencing databases for human Wilms tumors. Interestingly, ENL mutations often occur in Wilms tumor without other well-known genetic alterations, except in occasional instances with CTNNB1 mutations¹. With these considerations, future studies should consider inducing ENL mutations, either alone or in combination with CTNNB1 mutations, in a limited subset of kidney progenitors (e.g., using *Wt1-CreER* strain). This strategy would better recapitulate the sporadic nature of mutational events in human cancer and likely circumvent the detrimental developmental effects observed in the *Wt1-Cre/Enl-T1* model, allowing for a more targeted exploration of the potential role of ENL mutations in tumorigenesis.

Despite these limitations, our study reveals the role of mutant ENL on cell fate determination and differentiation during kidney development. Given the close link between Wilms tumor and disrupted kidney development, insights gained in this study will contribute to our understanding of the role of ENL protein in kidney development and eventually the pathogenesis of Wilms' tumor, as pointed out by this reviewer (Comment #7). We have discussed these insights and limitation in the revised manuscript as suggested (Line 823-832).

Major comments

1. The bioinformatics analyses suggested upregulation of Hox clusters in nephron and stromal progenitors, increase of *Wnt4* and *Chd6* in nephron progenitors, and increase of *Wnt5a* in stromal progenitors. While these findings are potentially interesting, *in situ* hybridization or immunostaining of these genes should be presented to support the main conclusions of the manuscript.

This is an excellent suggestion. We have put considerable efforts to validating the gene alterations identified in *Enl-T1* kidneys by bioinformatics analyses through RNAscope *in situ* hybridization (ISH) &

protein co-detection assay, immunofluorescence (IF) staining, and RT-qPCR analysis, as detailed below. Overall, these new results lend additional support to our main conclusions.

a. *Hox* genes: To confirm the upregulation of the *Hox* genes in nephron and stroma progenitors of *Enl-T1* kidneys, we began by comparing the mRNA levels of several *Hox* genes (*Hoxa9/11*, *Hoxc9/11*, *Hoxd9/11/12*) in *Enl-T1* kidneys to those in *Enl-WT* whole kidneys by RT-qPCR. Our data revealed that these *Hox* genes are expressed at significantly higher levels in *Enl-T1* kidneys than in *Enl-WT* (Figure S4g). In addition, we carried out RNAscope ISH for *Hoxc9* alongside protein co-detection for SIX2. We placed a particular focus on *Hoxc9*, as it is the predicted transcription factor that regulates *Wnt4*, *Cdh6*, and *Wnt5a* (Figure 4, Figure 6, Figure S7). The imaging data revealed that in *Enl-WT* kidneys, *Hoxc9* is highly expressed in SIX2⁺ NPCs, aligning with our scRNA-seq data showing high *Hoxc9* expression in NP1 and NP2 cells in *Enl-WT* kidneys (Figure S4e). Compared to WT kidneys, *Hoxc9* RNA ISH signals were elevated in SIX2⁺ NPCs of *Enl-T1* kidneys (Figure 4m), and elevated *Hoxc9* mRNA levels were also observed in SIX2⁻ cells within *Enl-T1* kidneys. To determine if any of these *Hoxc9*^{hi}SIX2⁻ cells could be stroma cells, we attempted co-detection for *Hoxc9* mRNA and FOXD1 protein. Unfortunately, two FOXD1 antibodies (Abclonal, # A20240; Thermo Fisher Scientific, # PA5-35145) we have tested didn't yield reliable results in this assay. Given that *Foxd1*⁺ stroma progenitors are typically found at the periphery of cap mesenchyme², we examined *Hoxc9* mRNA signals in the stroma cells surrounding the Six2⁺ CM structure. We noted that *Hoxc9* expression was higher in these cells in the T1 condition compared to their *Enl-WT* counterparts (Figure 4m). Together, these new results further support the upregulation of *Hox* genes in nephron and stromal progenitors revealed by scRNA-seq analysis.

b. *Wnt4*: To confirm *Wnt4* upregulation in *Enl-T1* nephron progenitors, we measured its mRNA levels in whole kidneys using RT-qPCR and observed higher expression in *Enl-T1* than *Enl-WT* kidneys (Figure R1). Our scRNA-seq data show that *Wnt4* is upregulated in Six2⁺ NP1 and NP2 cells within *Enl-T1* kidneys (Figure 4h). Thus, we performed RNAscope ISH for *Wnt4* alongside protein co-detection for SIX2 on E15.5 kidneys from WT and T1 embryos. In *Enl-WT* kidneys, *Wnt4* was found in the SIX2^{low} PA structure, which likely contains NP2 cells, and absent in the SIX2^{high} CM structure, which likely contains NP1 cells (Figure 4n and Figure S7i). This pattern aligns very well with our scRNA-seq data (Figure 4h) and previous research³. Importantly, *Wnt4* ISH signals were higher in *Enl-T1* SIX2⁺ cells (corresponding to NP1 and NP2) compared to *Enl-WT* (Figure 4n and Figure S7i), supporting conclusions from the scRNA-seq analysis.

Figure R1. mRNA expression of *Wnt4* (normalized to *GAPDH*) in *Enl-WT* and *Enl-T1* kidney. Two-tailed unpaired Student's t test. **p < 0.01. data represent mean ± SD.

c. *Cdh6*: Our scRNA-seq data revealed that during normal kidney development, *Cdh6* expression begins in intermediate (IM) cells within the renal vesicles and subsequently appears in differentiated cells (Figure 4h). CDH6 immunostaining confirmed that in *Enl-WT* kidneys, CDH6 was not detected in NPCs located in the CM and PA structures, but it was present in cells within the RV structure, which likely represent the IM cells identified in our scRNA-seq analysis. In contrast, in *Enl-T1* kidneys, *Cdh6* was aberrantly expressed in NP2 cells, as revealed by scRNA-seq (Figure 4h), and CDH6 protein was detected within the PA structure (Figure S7j). These results thus confirm the aberrant upregulation of *Cdh6* in *Enl-T1* nephron progenitors.

Figure R2. mRNA expression of *Wnt5a* (normalized to *GAPDH*) in *Enl-WT* and *Enl-T1* kidney. Two-tailed unpaired Student's t test. **p < 0.01. data represent mean ± SD.

d. *Wnt5a*: Our scRNA-seq revealed *Wnt5a* upregulation in *Foxd1*⁺ stroma progenitors in *Enl-T1* kidneys. RT-qPCR analysis of whole kidneys confirmed elevated *Wnt5a* expression in *Enl-T1* compared to *Enl-WT* (Figure R2). The modest increase observed can be attributed to *Wnt5a* being upregulated specifically in *Foxd1*⁺ stroma progenitors, a minor population within the kidney. Due to the lack of a reliable antibody for detecting FOXD1 by immunofluorescence, we performed RNAscope ISH for *Wnt5a* alongside

SIX2 protein co-detection. In *Enl*-WT kidneys, *Wnt5a* predominantly expressed in cells located at the periphery of cap mesenchyme, likely *Foxd1*⁺ stroma cells. In *Enl*-T1 kidneys, *Wnt5a* RNA signals were higher in these cells (Figure S9g), supporting the aberrant upregulation of *Wnt5a* in *Enl*-T1 stroma.

These new data are largely consistent with key gene alterations revealed by scRNA-seq data and provide additional support for our main conclusions. We have incorporated these data into the revised manuscript (Line 263-265, 434-454, 565-568, 587-588).

2. Are such gene alterations also observed in human Wilms' tumors? Although Fig. 2I&M seems to show the clinical relevance, the coefficient is only 0.418. Alterations in representative genes found in the mutant mice (e.g. *Hox* genes, *Wnt4*, *Cdh6*, *Wnt5a*) should be presented in the human setting. Authors may soften their claim depending on the data.

We thank the reviewer for this point. There could be several reasons to explain why the coefficient is not particularly strong: (1) The ENL-T1-induced gene alterations we identified in the mouse model occur during early nephrogenesis, whereas ENL-mutant-associated gene signatures in human are derived from established Wilms tumor. It is conceivable that not all transcriptional changes caused by a tumor-initiating event during nephrogenesis are retained in the established tumors. (2) The human data is from *bulk RNA-seq of tumor samples* which include heterogenous cell types, whereas the transcriptomic changes we derived from the mouse model are from *single-cell RNA-seq* which allows the identification of cell type-specific changes. Hence, the coefficient is likely negatively impacted by these differences in the two databases. Nevertheless, the positive coefficient we observed ($R = 0.418$, $p = 1.35e-6$, Figure 2k) supports that mutant ENL-induced transcriptional changes we identified from mouse models are relevant to human Wilms tumor.

As suggested, to further explore the relevance of our findings, we compared the expression of all 39 *HOX* genes, along with *WNT4*, *CDH6*, and *WNT5A* (Figure R3), which were identified as T1-UP DEGs from our analysis, between ENL-MUT (n = 8) and ENL-WT (n = 116) tumors in the TARGET-Wilms tumor cohort. We found 10 *HOX* genes with increased expression levels in ENL-MUT Wilms tumor (FC > 2 and p.adj < 0.05). While some of these *HOX* genes are not the same as those upregulated in *Enl*-T1 mouse models, likely due to species-specific functions and functional redundancies among *HOX* genes, these data support the upregulation of *HOX* genes in both murine and human contexts. Furthermore, *WNT4* and *CDH6* levels were higher in ENL-MUT than in ENL-WT tumor samples, whereas *WNT5A* levels did not differ. Note, our scRNA-seq revealed *Wnt5a* upregulation by mutant ENL in *Foxd1*⁺ stroma cells; however, given the heterogeneity of cells in Wilms tumors and the dominance of nephron cells, changes in *Wnt5A* expression in stroma might be masked.

Together, these data support that key transcriptional changes induced by mutant ENL in our mouse model are also upregulated in human Wilms tumors harboring ENL mutations. Future studies aiming at developing Wilms tumor model from mutant ENL mice will allow further comparison of mutant ENL-induced gene expression in animal models and patient samples.

Figure R3. Gene expression levels in Human Wilms tumor samples from the TARGET dataset (TARGET-WT).

3. For functional validation, the authors can test whether upregulation of *Hox* clusters *in vivo* leads to the similar phenotype and/or whether deletion of some *Hox* genes in ENL mutant mice alleviates the phenotype. The authors can provide such data if available. At the very least, the authors should discuss such experiments in the future.

To our knowledge, there are no *Hox* over-expression mouse models available for us to compare their phenotypes with ENL mutant mice. Furthermore, given that mutant ENL leads to upregulation of multiple

Hox clusters (Figure 2k) and that *Hox* genes are known to play redundant roles in nephrogenesis⁴⁻⁶, it would be technically challenging to delete these *Hox* genes simultaneously in mouse models to test if this alleviates the phenotype induced by mutant ENL. Nevertheless, we have discussed these points as suggested in the revised manuscript (Line 754–760).

4. *How does the ENL mutation lead to the upregulation of Hox clusters? The authors should at least discuss the underlying mechanisms.*

Our previous studies^{7,8} in the human embryonic kidney cell line HEK293 showed that Wilms tumor-associated ENL mutations (T1 to T8) can increase the self-association of the ENL protein, leading to the aberrant formation of submicron-sized condensates at select target genes, particularly the HOX genes. This results in increased recruitment of ENL and its associated elongation factors (e.g. AFF4, CDK9, and DOT1L), which in turn promotes the transcriptional elongation by RNA Polymerase II at these genes. Importantly, we show that perturbing mutant ENL's ability to form such condensates compromises its ability to increase chromatin targeting and drive target gene activation^{7,8}. In addition to findings in HEK293 cells, our unpublished work also shows that mutant ENL can increase the expression of *Hox* genes in hematopoietic stem/progenitor cells in vivo through a similar mechanism. We have discussed these mechanisms more clearly in the revised manuscript (Line 625-630).

5. *Treatment with the ENL inhibitor is promising, but the authors only treated the mice until E15.5. Can longer treatment alleviate the kidney defects and improve postnatal survival? If so, the impact of the treatment will be much greater. If not, the limitations should be described.*

In the original manuscript, we treated mice with 100 mg/kg TDI-11055 from E10.5 to E14.5 and collected kidneys at E15.5 for histological analyses and scRNA-seq (Figure 8). This treatment regimen led to full recovery of kidney size and partial recovery in differentiation structures and transcriptome in *Enl*-T1 kidneys. To address whether such treatment can be optimized to rescue postnatal survival, we extended the treatment (100 mg/kg) from E10.5 to E18.5 on WT and T1 embryos and allowed the pups to be born (Figure R4a). Using TDI-11055 treated *Enl*-WT kidneys as controls, we found that the extended 8-day treatment significantly ameliorated the small kidney size, aberrant CM/UB structures, and the lack of differentiating and differentiated structures induced by ENL-T1 (Figure R4b-f). Thus, compared to the previous 4-day treatment (Figure 8), the 8-day treatment exerts a slightly stronger effect on alleviating the differentiation defects caused by ENL-T1.

However, we found that both *Enl*-WT and T1 pups underwent 8-day treatment died shortly after birth at P0.5. Surprised by this data, we took a closer examination of TDI-11055-treated *Enl*-WT kidneys. We observed an aberrantly wider nephrogenic zone compared with untreated WT kidneys (Figure R4b). Normally, a single layer of CM/UB structures is located at the periphery of the kidney. However, in TDI-11055 treated *Enl*-WT kidneys, the CM/UB structures expanded towards the inside of the kidney. These aberrations may contribute to the dysfunction of the kidney and the postnatal death of *Enl*-WT pups. Given TDI-11055 can inhibit the acetyl-binding of both wildtype and mutant ENL proteins⁹, the phenotypes caused by TDI-11055 treatment in *Enl*-WT mice are presumably attributed to the inhibitor's effect on wildtype ENL. These results also reveal a previously unrecognized role of wildtype ENL in normal kidney development. To explore this hypothesis, we have begun to examine the role of wildtype ENL in normal kidney development by generating a conditional *Enl* knockout (KO) mouse model and crossing it with the *Wt1*^{GFP^{Cre}/+} strain (Figure R5a, b). Our preliminary data showed that homozygous *Enl*-KO in *Wt1*⁺ precursors resulted in a wider nephron zone in the kidneys and postnatal death (Figure R5c and d). These phenotypes were remarkably similar to those observed in *Enl*-WT mice with the 8-day TDI-11055 treatment (Figure R4b), supporting a role of wildtype ENL in kidney development that warrants further investigations.

With these new insights, the inability of the 8-day TDI-11055 treatment to improve survival in *Enl*-T1 heterozygous mice could be due to adverse effects on the wildtype *Enl* allele, an incomplete blockage of ENL-T1 function, or a combination of both. Despite this, the treatment's substantial mitigation of ENL-T1 induced abnormalities and gene expression changes (Figure 8) suggests its potential for treating ENL-

mutated cancers that warrants further exploration. However, given that ENL inhibitors also affect the wildtype ENL, a careful assessment of potential side effects is essential if this strategy is to be translated into clinical setting. We have discussed this important point in the revised manuscript (Line 838-842).

It is worth pointing out that treatments in cancer patients are typically applied post-development, and our previous studies demonstrated that long-term TDI-11055 treatment in adult mice did not result in significant adverse effects⁹, suggesting a promising therapeutic window for ENL inhibitors. Supporting this, our unpublished study found that heterozygous expression of *Enl-T1* in the hematopoietic system results in aggressive AML in mice, and TDI-11055 treatment significantly inhibited AML development and extended survival with minimal adverse effects in adult mice.

Figure R4. The impact of longer TDI-11055 treatment on *Enl-WT* and *Enl-T1* kidney development. **a**, Schematic to show the experimental dosing strategy. **b**, Left, Histology of P0.5 kidneys. Scale bar = 1 mm. Right, Immunostaining for indicated proteins on P0.5 kidney sections. E-cad, E-cadherin. Scale bar in the first column images, 200 μ m; scale bar in the other images, 100 μ m. CM, Cap mesenchyme; UB, ureteric bud; SB, S-shape body; CB, Comma-shape body; G, glomerulus; PT, proximal tubule; DT, distal tubule. **c-f**, The number of nephron structures per field. One dot indicates the number of one indicated structure per field. Data represent mean \pm s.d.; two-tailed unpaired Student's t-test. NS, no significance.

6. *There is no Wilms' tumor formation in the mutant mice, suggesting that additional molecular events are required for tumorigenesis. Is there evidence that the blastema-like cells (or progenitor cluster T1 as identified in the UMAP plots) ultimately contribute to Wilms' tumor formation? If not, these limitations should be clearly described.*

This is an excellent point. In our study, we showed that *Enf*-T1 kidneys harbored abnormal CM structures that resemble undifferentiated, proliferative blastema components seen in Wilms tumor (Figure 1c, f and Figure S1f, g), raising the possibility that they may have a role in tumorigenesis. Regarding the absence of Wilms tumor in the *Wt1*^{GFPcrei+}/*Enf*^{lox-T1/+} mice, there could be several contributing factors. One is that the profound developmental defects and early lethality resulting from *Enf*-T1 expression in *Wt1*⁺ precursors preclude a direct assessment of its role in Wilms tumor formation, which typically manifests months after birth. Another factor is the potential need for additional molecular events, as suggested by this Reviewer. To address this point, we examined published sequencing databases for human Wilms tumors. Interestingly, ENL mutations often occur in Wilms tumor without other well-known genetic alterations, except in occasional instances with CTNNB1 mutations¹. With these considerations, future studies should consider inducing ENL mutations, either alone or in combination with CTNNB1 mutations, in a limited subset of kidney progenitors (e.g., using *Wt1*-CreER strain). This strategy would better recapitulate the sporadic nature of mutational events in human cancer and likely circumvent the detrimental developmental effects observed in the *Wt1*-Cre/*Enf*-T1 model, allowing for a more targeted exploration of the potential role of ENL mutations in tumorigenesis. In the revised manuscript, we have pointed out this limitation, outlined potential reasons, and suggested directions for future research (Line 823-832).

Despite this limitation, our study reveals the role of mutant ENL on cell fate determination and differentiation during kidney development. Given the close link between Wilms tumor and disrupted kidney development, insights gained in this study will contribute to our understanding of the role of ENL in kidney development and eventually the pathogenesis of Wilms tumor, as pointed out by this reviewer (Comment #7).

7. I suggest adding a paragraph summarizing the limitations of the study to avoid misunderstanding by the readers. Even with such a paragraph, I still believe that this study will contribute to the understanding of the role of ENL protein in kidney development and eventually the pathogenesis of Wilms' tumor.

In response to feedback from reviewers, we have incorporated discussion of the study's limitations within the relevant sections of the Discussion.

Reviewer #2 (Remarks to the Author)

The authors have sought to examine a dominant allele of ENL to explore the link off ENL mutations with Wilms Tumor. The work is of a generally high quality.

We appreciate the reviewer's recognition of our study's high quality. We have thoroughly addressed concerns raised by this reviewer, as detailed below.

Comment 1 - However, there is a major problem with the study and the impeded development is not necessarily the simplest expectation of a WT model in which massive kidney growth in the oncogenic outcome?

This is an excellent point also raised by Reviewer 1. We acknowledged the limitation regarding the absence of Wilms tumor in *Wt1^{GFPcrei+}/Enl^{lox-T1/+}* mice and please see our detailed response to Reviewer 1, comment # 6. In the revised manuscript, we have pointed out this limitation, outlined potential reasons, and suggested directions for future research (Line 823-832).

Comment 2 - The major issue is that all the analyzes were performed relatively speaking a longtime after the initiation of the phenotype. By the severity of the observed phenotype, I expect that there will be a phenotype evident at E12.5 but much of the analysis compares wild-type and mutants at E15.5 and E18.5. This leaves open secondary medication of all the parameters measured. This is a substantial issue given the tight linkage of regulation amongst multiple cell types within the nephrogenic niche.

We appreciated the reviewer's insightful comment. When we dissected the embryos at E12.5, we found that *Enl-T1* kidneys are much smaller than their WT counterparts, in agreement with the reviewer's expectation that the phenotype is evident at E12.5. However, the incredibly small size of *Enl-T1* kidneys posed a great challenge for handling tissue collection and single-cell analysis. Thus, we chose to conduct our analysis at E15.5. More importantly, E15.5 represents a developmental stage when all defined nephron structures start to emerge, thus enabling us to evaluate the impact of mutant ENL on cellular composition and changes in gene expression and chromatin accessibility across all major cell types in the developing kidney. Similar time points have been chosen in the field for single-cell analysis to examine the chromatin changes in NPCs during development¹⁰ and to identify key molecular coordinates of mature stages in developing kidneys¹¹. Thus, acknowledging that the effects of the mutation manifest earlier than E15.5, we respectively think that the analyses of E15.5 kidneys provide critical insights into the mutation's specific influence on gene expression and chromatin states across various cell types, thus informing how it impacts kidney development. Our single-cell studies, especially the observed upregulation of *Hox* genes by mutant ENL, are corroborated by cellular experiments, affirming a direct role of the ENL mutation in regulating these targets (Figure 7).

Comment 3 - Further, there may be evidence suggesting problems within cells – is an enhanced mitochondrial signature an indication of low quality RNA in the NP single cell profiling that might suggest

“sick” cells? Fortunately, there are two steps to rectifying this. First, determine when a phenotype is first observed, then repeat analyzes at this time. This would be expected to give the clearest insight into the initiating steps and allow a clearer interpretation of data in hand. Second, the authors show a remarkable rescue of the kidney phenotype treating with TDI-11055. The authors could also stop administration of the drug at e14.5 and examine transcription associated cell properties sometime later determined by preliminary study – this withdrawal would also give a likely much better handle on the events at play given many more nephrogenic niches to examine at this later timepoint than an earlier one. But, the earlier 11.5/12.5 (no drug) and a potentially 15.5 drug withdrawal at e14.5 study would be complementary.

We thank the reviewer for this thoughtful question. In our scRNA-seq analysis, we carefully excluded low-quality cells, using standard criteria to filter out cells with either an abnormally low (<1000) or high (> 6000) number of expressed genes or with mitochondrial gene content exceeding 10%. Thus, the enhanced mitochondrial-related gene signatures in *Enl-T1* kidneys are unlikely due to low quality RNA resulted from apoptosis or cell lysis, etc. Nevertheless, we thank the reviewer for the two suggested strategies. For the first one, as in our response to this reviewer’s comment #2, handling E11.5/12.5 *Enl-T1* kidneys for tissue dissociation and single-cell analyses poses significant challenges due to their exceptionally small sizes. Regarding the second strategy, we have already implemented the suggested treatment regimen, administering TDI-11055 from E10.5 to E14.5 followed by histological and single-cell RNAseq analysis at E15.5. This treatment significantly rescued kidney size and nephron structures in *Enl-T1* kidneys (Figure 8b-g), which suggests that cells from the treated *Enl-T1* kidneys are functional but not merely ‘sick’. Notably, the mitochondrial-related gene signatures upregulated in *Enl-T1* kidneys persisted despite TDI-11055 treatment, further indicating that these are not solely indicators of cellular distress. Furthermore, these mitochondrial signatures are present in both ENL-T1-positive nephron and stroma compartments and in cell types not expressing *Enl-T1*, such as endothelial and UB-UE cells, suggesting these changes are not a direct consequence of ENL-T1 expression. Interestingly, there is evidence that mitochondrial-related pathways become increasingly activated during nephron progenitor differentiation¹², raising the possibility that the mitochondrial signatures in *Enl-T1* kidneys might be linked to premature commitment of these cells, a hypothesis that warrants further investigations.

Comment 4 - There is also a concern in the genetic set up of the model which unfortunately removes a copy of a WT associated gene (WT1) and the potential for a synthetic interaction that cannot be controlled for in this model. Given the dominant effects of the modified ENL allele, the authors could cross to the Six2-TGC line available from the JAX labs (IMSR_JAX:009606) to generally see if the same outcomes are observed? This line is only active in the nephron lineage so it would also provide additional information not possible with the WT1 cre which is active in both interstitial and nephron lineages, on the lineage outcomes for ENL allele. Again, I am not recommending an extensive analysis for this but it is not too difficult or time-consuming to gather important new insight that may allay concerns with the WT1-cre strategy?

This point is well-taken. We initially chose the *Wt1*-Cre line as it is one of the commonly used strains in the field for investigating the impact of a given genetic alteration in kidney development and Wilms tumor formation^{13,14}. Furthermore, mutations implicated in Wilms tumor have been found in both nephron and stroma lineages¹⁵. There have been instances where the phenotypic impact of a mutation is apparent with *Wt1*-Cre but not with other more lineage-specific Cre strains¹³. Importantly, to account for the potential impact of losing a *Wt1* allele on kidney development, we consistently used ^{*Wt1*GFP^{Cre}/+} as control throughout our study to ensure that our comparing groups have the same amount of *Wt1* and the only difference is the *Enl* allele.

We acknowledge the value of using the Six2-TGC line to garner insights specifically into *Enl-T1* activation in the nephron lineage. As suggested, we crossed ^{*Enl*lox-T1/+} mice with ^{*Six2*GFP^{Cre}/+} (*Six2*-TGC) mice to induce *Enl-T1* expression specifically in *Six2*⁺ NPCs and their progeny (Figure S2a). We found that similar to ^{*Enl*lox-T1/+} ^{*Wt1*GFP^{Cre}} mice, ^{*Enl*lox-T1/+} ^{*Six2*GFP^{Cre}} mice died shortly after birth (Figure S2b), alleviating the concern that the function of *Enl-T1* is contingent solely on losing one *Wt1* allele. However, the phenotypic manifestations of ENL-T1 expression differ between the two Cre strains. Specifically, ENL-T1 expression in *Six2*⁺ NPCs minimally impacted overall kidney size and tubular structures, but more prominently affected

the development and maturation of glomeruli in embryonic (E15.5 and E18.5) (Figure S2c-f) and neonatal (P0.5) (Figure S2g, h) kidneys, as indicated by reduced, shrunken, and fragmented glomeruli. The *Six2* promoter drives expression solely in the nephron lineage, whereas *Wt1* is expressed earlier in development in metanephric mesenchyme which gives rise to both nephron and stroma lineages^{16,17}. These observed phenotypic differences between the two cre strains might suggest impact of ENL-T1 in both the nephron and stroma compartments. Corroborating this, our scRNA-seq and snATAC analyses revealed extensive transcriptomic and chromatin accessibility changes in the stroma progenitor cells in *Enl^{lox-T1/+}Wt1^{GFPCre}* kidneys. These alterations may adversely affect nephrogenesis by disrupting nephron-stroma interactions (Figure 6 and S9). We have included these data in the revised manuscript (Line178-188), and future research is needed to further elucidate the lineage-specific function of the ENL mutation in developing kidneys.

Reviewer #3 (Remarks to the Author)

In this manuscript, Song et al describe the development and characterization of a new genetic mouse model of a Wilms tumor associated ENL mutation in kidney development. The authors utilized Wt1-Cre to drive the expression of the ENL T1 mutant and observed that the mutant mice displayed severe kidney developmental defects and early postnatal death. They further carried out scRNA-seq and snATAC-seq analyses in E15.5 kidneys and found that the ENL mutation disrupted the kidney development trajectory, by rewiring gene regulatory landscape. Mutant ENL was found to influence nephron progenitor commitment and differentiation by dysregulating key transcription factors, including Hox genes. Additionally, ENL T1 was implicated in disrupting the normal stroma-nephron interaction critical for nephrogenesis, primarily through hyperactivation of the Wnt signaling. Notably, transient inhibition of the reader function of the ENL mutant with TDI-11055 partially restored kidney developmental defects and gene expression patterns, providing a proof-of-concept for epigenetic therapy. Overall, this is a comprehensive study providing new insights into how ENL mutation disrupts kidney development at single-cell resolution. Several concerns and suggestions are outlined below to strengthen its scientific rigor and clarity of the study.

We are pleased to hear that the reviewer found our study comprehensive and that it brings new biological insights. We have addressed the concerns and suggestions raised by this reviewer, as detailed below.

Major concerns:

1. *This study centers on a new genetic mouse model but validation seems limited. It would greatly strengthen this study if the authors could provide more rigorous data validating the mouse model. For example, confirmation of mutant ENL expression upon Cre-mediated recombination, and comparison of the protein level of mutant ENL versus endogenous wildtype ENL.*

As suggested, we have provided additional description and validation of the mouse models in the revised manuscript (Figure S1a-d and Method), as detailed below:

a. The strategy of generating the mouse model has been described in further detail in the revised Methods. In addition, PCR genotyping used to distinguish the *WT* and *T1* alleles was included in the revised manuscript (Figure S1b).

b. There is no antibody available to distinguish ENL-WT and ENL-T1 proteins, preventing us from comparing the protein levels of mutant versus endogenous wildtype ENL following Cre recombinase-mediated editing. Thus, we used an alternative strategy to compare their mRNA levels in *Enl^{lox-T1/+}Wt1^{GFPCre/+}* kidneys. Briefly, RNA was extracted from an *Enl^{lox-T1/+}Wt1^{GFPCre/+}* kidney and reverse transcribed into cDNA. We then employed a primer pair that can amplify both WT and T1 *Enl* cDNA and conducted RT-qPCR. The PCR products underwent next-generation sequencing to compare the relative abundance of *Enl*-WT and *Enl*-T1 mRNA (Figure S1c). As a negative control, we also sequenced PCR

products from an *Enl*^{+/+}/*Wt1*^{GFP-Cre/+} control kidney. Our results showed that *Enl*-WT kidneys only express *Enl*-WT mRNA, as expected. In *Enl*-T1 kidneys, 58% and 41% of sequencing reads mapped to *Enl*-WT and *Enl*-T1 cDNA, respectively (Figure S1d). These results confirm that *Enl*-T1 is expressed upon Cre-mediated recombination and indicate overall comparable expression levels of *Enl*-T1 and *Enl*-WT alleles in *Enl*^{fllox-T1/+}/*Wt1*^{GFP-Cre/+} kidneys. The slightly lower level of *Enl*-T1 (41% vs 58%) compared to *Enl*-WT is likely because some cells in *Enl*-T1 kidneys (e.g. UB-UE) are not derived from *Wt1*⁺ cells and thus should not express *Enl*-T1. Lastly, as *Enl*-T1 is expressed from the endogenous *Enl* promoter and there is no difference in ENL-WT and ENL-T1 protein stability⁸, the mRNA levels could reasonably reflect the protein levels.

2. *WT1* is a critical regulator of kidney development. In the mouse model used in this study, the presence of *Wt1*-GFP-Cre leads to concomitant inactivation of a copy of the endogenous *Wt1*. Although the authors claim that heterozygosity of *Wt1* does not cause any phenotypic defects in kidney development, one would expect some epigenetic and gene expression changes caused by *Wt1* heterozygosity. Therefore, it is necessary to include wildtype kidney samples in Figure 2 for direct comparison.

We thank the Reviewer for bringing up this important point. It has been reported that *Wt1*^{GFP-Cre/+} heterozygous mice are phenotypically normal during embryonic development¹⁸, and our histological assessment aligns with previous literature. This is likely one reason that *Wt1*-Cre strain is commonly used in the field to study the impact of gene knockout or mutation on kidney development and Wilms tumor formation^{13,14}. Nevertheless, as suggested, we compared our scRNA-seq dataset on E15.5 *Enl*^{+/+}/*Wt1*^{GFP-Cre/+} kidneys with the published dataset on E15.5 C57BL/6N mouse kidneys¹¹. The UMAP of the integrated scRNA-seq datasets suggested that the transcriptomes of these two datasets closely resemble each other (Figure R6a). The integrated nephron cells were classified into different cell types based on the expression of specific markers as detailed in the manuscript (Figure R6b-c). The pseudotime trajectory analysis on these two datasets revealed largely similar results (Figure R6d). These results are consistent with the histological assessment. Importantly, to control for any epigenetic and gene expression changes caused by *Wt1* heterozygosity, we used *Enl*^{+/+}/*Wt1*^{GFP-Cre/+} as control throughout our study to ensure that our comparing groups have the same amount of *Wt1* and the only difference is the *Enl* allele.

Figure R6. Comparison of E15.5 kidney scRNA-seq data between *Wt1*^{GFP-Cre/+} mouse and C57BL/6N mouse. **a.** UMAP embedding of scRNA-seq data labeled with four main embryonic kidney lineages. UB-UE, ureteric bud-ureteric epithelium. **b.** UMAP embedding of integrated scRNA-seq cells from nephron lineage. NP, nephron progenitor; Podo, podocyte; LOH, loop-of-Henle; PT, proximal tubule; DT, distal tubule. **c.** Violin plot showing the gene expression of selected makers for each cell type in nephron. **d.** UMAP embedding of *Enl*-WT and T1 scRNA-seq nephron differentiation trajectory. Cells are colored by pseudotime. Trajectories are depicted in red.

3. Another issue of the mouse model is a possible synergistic effect of *Enl* mutation and *Wt1* heterozygosity, whereas in human patients, *ENL* mutations are not found to be concurrent with *WT1* mutations. Thus, it is critical to validate some major phenotypic findings using another kidney specific *Cre*.

This is an excellent point also raised by Reviewer 2. Please see our detailed response to Reviewer 2, comment #4.

4. The scRNA-seq and snATAC data are beautiful but do not tell what the direct effects of *ENL* mutation are. Experimental validation is needed to distinguish direct and indirect effects of *ENL* mutations, for example, by performing CUT&RUN or CUT&Tag experiments for wild-type and mutant *ENL*.

The suggested experiments would be powerful in identifying direct targets of *ENL-T1* involved in kidney development. However, such experiments are currently not feasible due to the lack of antibodies that can distinguish between WT and T1 proteins. We have collaborated with Millipore to develop antibodies that specifically recognize *ENL-T1*. While our pilot antibodies can successfully differentiate between *ENL-WT* and *ENL-T1* using highly concentrated purified recombinant proteins, they, unfortunately, have proven ineffective with cell lysates in assays such as western blot and ChIP-seq. We acknowledge the importance of such experiments and will continue our efforts in generating these antibodies for this purpose.

One of the major findings from the study is that mutant *ENL* upregulates *Hox* genes, which likely contribute to the developmental defects observed in the mouse model. Importantly, we have shown that a series of *ENL* mutants, including *ENL-T1*, bind and form aberrant condensates directly on *HOXA* genes in multiple systems including in HEK293 cells^{7,8} and hematopoietic stem/progenitor cells (unpublished data), leading to hyperactivation of these *HOX* genes. These results, together with the strong upregulation of *Hox* genes in *Enl-T1* mouse kidneys and in *ENL*-mutated Wilms tumors, provide strong evidence that *Hox* genes are likely direct targets of mutant *ENL*. That said, many transcriptomic changes observed in *Enl-T1* kidney cells could be downstream effects of the upregulated *Hox* genes, as suggested by our ATAC-seq analysis (Figures 4 and 6). It is conceivable the primary and secondary transcriptional changes induced by *ENL* mutations all together contribute to the biological consequences and possibly pathogenic function of *ENL* mutations. We have discussed these points in the revised manuscript (Line 754-760).

5. It is impressive that transient inhibition of *ENL* mutant by *TDI-11055* can partially rescue mutant *ENL*-induced transcriptomic and developmental alterations *in vivo*. Have the authors tried extending the treatment to a later stage in embryonic development (e.g. E18.5)? That would allow a better evaluation of the restoration of nephrogenesis. It would be interesting to see whether the neonatal lethality of *Enl T1* can be partially rescued.

This is an excellent question also raised by Reviewer 1. We have performed the suggested experiments. Please see our detailed response to Reviewer 1, comment # 5.

Minor issues:

1. Some *Enl-T1* kidneys exhibited structures that resemble undifferentiated blastema components seen in Wilms tumors. Are they proliferative? The authors can perform vimentin and Ki-67 staining to determine their blastemal identity and proliferative status.

As suggested, we have performed IHC for Vimentin and Ki-67 on E15.5 kidneys. We observed an increased expression of Vimentin in the cap mesenchyme (CM) cells of *Enl-T1* kidneys compared to their *Enl-WT* counterparts. Additionally, the CM cells in *Enl-T1* kidneys, including those within blastema-like structures, were positive for Ki-67, indicating active proliferation. These findings indicate that the aberrant CM in *Enl-T1* kidneys not only exhibits blastemal morphology and Vimentin positivity but is also proliferative. The results have been incorporated into the revised figure (Figure S1f).

2. Are transcripts of wild-type and mutant *ENL* detectable in the scRNA-seq datasets? It is important to

know in which cell lineages the T1 mutant is expressed and whether the wild-type and mutant ENL are expressed in the same sets of cells.

Unfortunately, the scRNA-seq database cannot distinguish the transcripts of *Enl*-WT and *Enl*-T1. UMAP of *Enl*-WT kidneys showed that endogenous *Enl* is expressed broadly in all cell lineages within the kidney (Figure R7). The *Wt1*^{GFP-Cre} mouse allows *Enl*-T1 to be expressed in *Wt1*⁺ cells and its progeny, which include cells in stromal and nephron lineages. Thus, in theory, *Enl*-T1 should be expressed in these two lineages in *Enl*^{fllox-T1/+}/*Wt1*^{GFP-Cre/+} mice. Given that the *Enl*-T1 allele is controlled under the endogenous *Enl* regulatory elements, it should be expressed in the same sets of cells as the *Enl*-WT allele in *Enl*^{fllox-T1/+}/*Wt1*^{GFP-Cre/+} kidneys. Furthermore, as

detailed in our response to comment #1 from this reviewer, we compared the expression levels of WT versus T1 mRNA in *Enl*^{fllox-T1/+}/*Wt1*^{GFP-Cre/+} kidneys using RT-PCR and next-generation sequencing, and our results showed comparable expression of WT and T1 *Enl* (Figure S1c, d).

3. Based on the marker genes identified in scRNA-seq, is it possible to perform IHC staining to locate the abnormal NP1, NP2, and T1-ab cells in developing nephrons and kidneys?

Our scRNA-seq analysis revealed that in both *Enl*-WT and *Enl*-T1 nephrons, *Six2* is expressed in both NP1 and NP2 cells (Figure 5i), with NP1 cells showing higher *Six2* expression than NP2 (Figure 5i). Furthermore, in *Enl*-WT kidneys, *Wnt4* is not expressed in NP1 cells but is expressed at a low level in NP2 cells (Figure 4h). In contrast, in *Enl*-T1, *Wnt4* is aberrantly expressed in NP1 cells and its expression is higher in NP2 (Figure 4h). Thus, in *Enl*-WT nephrons, *Six2*^{hi}*Wnt4* and *Six2*^{low}*Wnt4*^{low} cells likely represent NP1 and NP2 cells residing in the CM and PA structures, respectively. In *Enl*-T1 nephrons, *Six2*^{hi}*Wnt4*^{low} and *Six2*^{low}*Wnt4*^{hi} cells likely represent the abnormal NP1 and NP2 cells, respectively.

To spatially identify NP1 and NP2 cells in the kidneys, we performed RNA in situ hybridization for *Wnt4* mRNA coupled with IF for SIX2 protein on E15.5 *Enl*-WT and *Enl*-T1 kidneys. As predicted, in *Enl*-WT nephrons, we detected SIX2^{hi}*Wnt4* NP1 and SIX2^{low}*Wnt4*^{low} NP2 cells located in the CM and PA structures, respectively. In *Enl*-T1 nephrons, SIX2^{hi}*Wnt4*^{low} NP1 cells and SIX2^{low}*Wnt4*^{hi} NP2 cells were identified surrounding the UB tip structures (Figure 4n and Figure S7i).

As for the T1 abnormal cell clusters, our scRNA-seq analysis suggests that these cells are arrested in an aberrant progenitor state, showing a decrease in chromatin accessibility associated with kidney lineage identity and a increase in chromatin accessibility associated with other lineages (Figure 5). While *Hox* genes, *Wnt4* and *Cdh6* are upregulated in T1-ab cells (Figure 2k and 4h), there are no specific markers for staining to locate these cells. Spatial transcriptomics could be instrumental in future studies for accurately locating these cells in *Enl*-T1 kidneys.

4. The human ENL-mutant_Up genes were expressed at higher levels in mutant NP1, NP2 and T1-ab cells. How about their expression in mutant stromal progenitors?

Figure R7. *Enl* expression in scRNA-seq data. a-b. UMAP embeddings showing *Enl* expression patterns in the *Enl*-WT (a) and *Enl*-T1 (b) kidney. Cells are colored by the expression level.

We have analyzed the expression levels of human ENL-mutant_Up genes in *Enl*-T1 stromal progenitors. The results revealed a modest yet significant increase in UCell scores for *Enl*-T1 *Foxd1*⁺ stroma progenitors compared to their *Enl*-WT counterparts (Figure R8). This aligns with the observation that *Enl*-T1 upregulates a shared subset of genes in both nephron and stromal cells in our mouse model, including the *Hox* genes (Figure S2i).

Figure R8. The UCell score evaluated by human ENL_MUT_UP signature for the *Foxd1*⁺ stroma progenitor (SP) cells within *Enl*-WT and T1 datasets, respectively. Wilcoxon rank-sum test p-values are shown.

5. Fig. 5i and 5l: consider adding gene expression in wt NP1 and NP2 for comparison. We have revised the figure as suggested to include gene expression in WT NP1 and NP2 (Figure 5i, l).

6. Fig. 7l-m show that the numbers of *HOXA11* nascent RNA FISH foci in mutant ENL expressing cells were reduced upon TDI-11055 treatment. How about the RNA FISH intensity of *HOXA11* foci?

We quantified the RNA FISH foci intensity for *HOXA11* (ENLmut target gene) and *GAPDH* (non-ENLmut target gene) in HEK293 cells expressing ENL-T1 or ENL-T2 under DMSO and TDI-11055 treatment conditions. Our results revealed a reduction in *HOXA11* RNA FISH intensity following TDI-11055 treatment (Figure S10m), while *GAPDH* intensity remained unchanged (Figure S10p). This decrease, alongside a lower frequency of *HOXA11* RNA-FISH foci in cells treated with TDI-11055 (Figure 7n, o), strongly indicates that *HOXA11* gene expression is downregulated by TDI-11055 treatment. These new data have been incorporated into the revised Figure (Figure S10m, S10p).

7. How does TDI-11055 treatment affect the percentage of cells in different lineages? Please provide a stacked bar plot, similar to Fig. 2b, to compare cellular compositions in WT, T1 and T1-TDI.

As suggested, we quantified the distribution of different cell lineages of *Enl*-T1 kidneys after TDI-11055 treatment. We found that the proportion of the nephron lineage decreased, whereas that of the stroma lineage increased in *Enl*-T1 kidneys relative to *Enl*-WT. TDI-11055 treatment partially reversed these changes, indicating that the treatment effectively mitigated the T1-induced compositional abnormalities in both nephron and stroma lineages. The proportion of endothelial cells remained unchanged with TDI-11055 treatment. We have incorporated this data into the revised manuscript (Figure S11c, Line 695-698).

Reviewer #4 (Remarks to the Author)

Summary:

The authors have investigated the cellular and molecular mechanisms of the most common pediatric kidney tumor, Wilms tumor. Focusing on the epigenetic reader protein ENL, the most frequently mutated epigenetic regulator in Wilms tumors, the authors have used a combination of genetic mouse models, histology, scRNA and snATAC techniques to gain insights into the effects of ENL mutation in the context of Wilms tumors.

The authors use single cell transcriptomics in developing mouse embryos from their *Enl*-T1 mutant mice, to show that *Enl* mutation significantly affects the composition of nephrons, disrupts normal nephrogenesis, and results in abnormal undifferentiated progenitors. Their finding that *Hoxa/c/d* family genes are upregulation in *Enl*-T1 is clinically important as ENL mutant Wilms tumors in patients also show increased expression of *Hox* genes. One of the key novel findings was obtained by utilizing snATAC-seq and shows that there are dynamic changes in chromatin accessibility in the *Enl* mutant kidney vs. developing mouse

kidney. Additionally, the authors show that the *Enl* T1 cells may have substantial chromatin alterations before transcriptional changes, thus shedding light on aspects that would otherwise be undiscoverable via scRNA-seq.

The changes in NPC commitment at various cell fate transitions and the altered differentiation trajectories comprise some interesting results. Furthermore, the authors showed that the mutant *Enl* induced a clusters of abnormal progenitor state with a loss of nephron chromatin identity. Particularly fascinating was the gain of potential to differentiate into other organs like mammary gland and heart, which could have implications for other related cell fate decision fields. Finally, the authors utilized their in house small molecule *Enl* inhibitor TDI-11055, and showed that inhibition of chromatin function of mutant *ENL* can reverse its effects, which is fantastic step forward in the field of epigenetic therapies.

Overall this is a well conducted study that utilizes a wide array of techniques and technologies to tackle and provide a solution to a clinically important problem. The novel and significant results gained here will be valuable both for the advancement of knowledge in the field and in the clinical setting. However, there are several important concerns that should be resolved prior to publication.

We are very pleased that this reviewer finds our work well conducted, novel, and significant. We greatly appreciated this reviewer's constructive feedback and suggestions to improve the study, and we have addressed all comments as detailed below.

Concerns and suggestions:

1. My major concern is the section about stroma-nephron interactions by aberrant activation of *Wnt* signaling. The authors show increased beta-catenin signature in nephron lineage, and increased *Wnt5* in the stromal progenitors. However, a paracrine signaling connection between these two has not been shown/established in this study/system (line 549, 559). Given that *Wnt4* is upregulated in the *Enl*-T1 NP1 and NP2 cells, it could be possible that *Wnt4* increase has caused the increase in beta-catenin in an autocrine manner. There is also an established link between *Tcf21* and beta-catenin in the literature, which could have caused the increased beta catenin signature. Therefore, direct evidence such as *Wnt5a* from stromal cells binding to *Fzd*/*LRP6* on the nephron lineage cells, or FRET/BRET with tagged *Wnt5a* and its receptors, would be necessary to conclusively show the interaction between stroma and nephrons. Discussion (page 73, paragraph 1) would need to be updated accordingly.

This is an excellent point. We agree that upregulation of *Wnt4*, a key factor in NPC commitment and MET initiation, likely contributes to the observed increase in 13-catenin gene signatures in *Enl*-T1 NP1 and NP2 cells, potentially through an autocrine mechanism. The slight increase in *Tcf21* expression in these cells (Figure 5i) could also contribute to the increased 13-catenin signatures, as pointed out by the reviewer. These insights have been integrated into the discussion on 13-catenin gene signatures in NPCs and stroma progenitors (Line 602-613).

Furthermore, the elevated *Wnt5a* in *Foxd1*⁺ stroma progenitors suggests but does not definitely prove, potential paracrine signaling to adjacent NPCs. This hypothesis stems from the following observations: (1) Studies in genetic mouse models have shown that alterations in 13-catenin activity in *Foxd1*⁺ progenitors can significantly impact NPC differentiation¹⁹. (2) Prior studies have suggested that stroma signals could amplify *Wnt*/13-catenin activity in NPCs²⁰. (3) In addressing comments from Reviewer 2 and 3, we crossed *Enl*^{fllox-T1/+} mice with *Six2*^{GFP^{Cre}/+} strain to express *Enl*-T1 specifically in the nephron lineage. Our results revealed more severe developmental defects in *Enl*^{fllox-T1/+}/*Wt1*^{GFP^{Cre}/+} mice compared to *Enl*^{flloxT1/+}/*Six2*^{GFP^{Cre}/+} mice, underscoring the critical contribution of *Enl*-T1 stroma cells to the observed kidney developmental defects in *Enl*^{fllox-T1/+}/*Wt1*^{GFP^{Cre}/+} mice. We acknowledge the necessity of direct evidence demonstrating *Wnt5a* interaction with NPC receptors to substantiate this hypothesis. Considering the challenges of conducting such experiments *in vivo*, we propose that future research could explore *ex vivo* co-culture systems for a more definitive examination of this hypothesis.

These considerations and a more cautious interpretation of nephron-stroma interactions have been incorporated into the revised manuscript (Line 43-44, 573-576, 594-596, 602-613, 772-773, 778-781).

2. The authors have used the *Wt1* promoter for generating the *Enl* mutant mice. *Wt1* is primarily expressed in nephron and stroma. Could the authors comment on the possibility of *Wt1* expression bias in inferences made in the downstream studies, which would be limited to the specific cell types where *Wt1* Cre is expressed? For example, in line 342-344: the authors' argument is "the *Enl*-T1 kidney exhibited a reduction in specific nephron clusters and an increase in certain stromal clusters, emphasizing the influence of mutant *Enl* on nephron and stroma" However, the T1 mutation primarily only effects these 2 lineages, so it is difficult to say how much of the effects are from the mutation being limited to the 2 lineages vs. actually only affecting those 2 lineages. As such, it would be important to point out the bias of *Wt1*/T1 in the downstream results and interpret the results accordingly throughout the manuscript.

This is an excellent point. Wilms tumors are believed to arise from nephron and possibly stroma cells^{13,15,19,21}. Thus, these two lineages are likely more relevant for examining the impact of Wilms tumor mutations on kidney development and tumorigenesis. We agree on the *Wt1* expression bias in inferences made in the downstream studies and have modified the statement to remove 'emphasizing the influence of mutant *Enl* on nephron and stroma' (Line 363-365). We also pointed out the need for future studies to determine whether mutant ENL can also affect other lineages within the kidney (Line 795-800).

3. The authors show that treatment with TDI partially reverted the increase of *Wnt4*+ cells (line 648), and partially restored expression levels of a small subset of the downregulated DEGs (line 651). I am curious as to what would be the effects of other lower/higher dosages of TDI? Would a higher concentration perform better or does the rescuing effects of TDI plateau at 100 mg/kg?

Exploring the potential of a higher TDI-11055 dosage to further improve rescue effect, we increased the dose from 100mg/kg to 200 mg/kg (E10.5-E14.5) and evaluated kidney phenotypes at E15.5 (Figure R9a). Our results showed that 200mg/kg was as effective as 100 mg/kg in restoring kidney size and various nephron structures (Figure R9b), suggesting a plateau in the drug's efficacy beyond a certain dosage.

Figure R9. The impact of higher dosage TDI-11055 treatment on *Enl*-WT and *Enl*-T1 kidney development. a, Schematic to show the experimental dosing strategy. **b**, Left, Histology and Immunostaining for indicated proteins of E15.5 kidneys. CM, Cap mesenchyme; UB, ureteric bud; SB, S-shape body; G, glomerulus; PT, proximal tubule.

Additionally, if Wnt signaling is involved in mutant *Enl* induced defects, could the authors allude to the use of Wnt inhibitors like *Dkk1/SOST* for rescuing the effects of mutant *Enl*?

Thanks for the reviewer bringing up the point. In our study, we found that the Wnt/ β -catenin gene signature is upregulated in *Enl* T1 nephron and stroma progenitor cells, which could result from *Enl* T1 induced upregulation of *Wnt4* and *Wnt5a*. Nonetheless, beyond *Wnt4* and *Wnt5a*, other transcriptional changes, particularly the marked upregulation of *Hox* cluster genes, likely also contribute to *Enl* T1-induced developmental defects (Figure S6g and 6h). Thus, using Wnt inhibitors might not effectively mitigate these defects compared to TDI 11055, which targets mutant ENL directly. Moreover, inhibiting Wnt signaling could negatively impact kidney development, as evidenced by previous studies²².

4. Line 320-321: Could the authors speculate why the transition from NP2 to podocytes showed loss in chromatin accessibility, and its implications for their study? The podocyte cluster is mostly WT sample (Figure 3d), which may have interesting associations, for .e.g Could the T1 ab cells in C4/C10 be an intermediate to Podocytes (Figure 3c, Figure 5a,b)?

The cell fate commitment process is often accompanied by shutting down of gene programs associated with alternate lineages. Thus, the decrease in chromatin accessibility we observed during the transition from NP2 to podocytes in *Enl*-WT nephrons may reflect the suppression of gene programs for other cell types, such as tubule cells.

Despite the spatial proximity of T1-ab cells (C4/10) to podocytes on the UMAP plot (Figure 2c) we do not think that T1-ab cells serve as an intermediate state towards podocyte differentiation for several reasons:

- 1) Our scRNA-seq analysis (Figure 5) suggests that T1 ab cells do not enter the proliferative stage observed in NP2 cells, an important phase marking NPC commitment to differentiation. Moreover, T1-ab cells gain open chromatin accessibility associated with other tissue types indicating an aberrant chromatin state that make them improbable precursor to podocytes
- 2) Trajectory analysis (Figure S8b) suggests that T1-ab cells might originate from uncommitted NP1, while podocyte precursors in *Enl* WT nephrons emerge from committed NP2 or downstream intermediate cells.
- 3) Upon comparing podocyte precursors in WT nephrons and T1 ab cells using established markers, we found that podocyte precursors display high levels of *Mafb* and low levels of mature podocyte marker *Nphs1*. However, these markers are not present at all in T1 ab cells (Figure R10a c), further distinguishing them from traditional podocyte precursors.

a E15.5 WT nephron scRNA-seq

b

c

Figure R10. Comparison between wildtype podocyte and T1-ab cells. a, UMAP embedding of scRNA-seq data of *Enl*-WT nephron. b, The expression levels of *Mafb* and *Nphs1* in all nephron culters. c, The expression levels of *Mafb* and *Nphs1* in podocyte precursor and T1-ab cells.

5. The results involving self renewal, cell fate commitment, premature commitment vs differentiation block of NPCs, etc. can be difficult to track in terms of which data modality points to what result. Perhaps the

authors could include schematics to summarize their results in Figure 2, 3, and 4, along with the data utilized to support it. Something like a detailed version of Fig 5m, 5n projected on the UMAPs or a block diagram like Fig S5c with details of genes regulating the cell fates and differentiation would be great in conveying the results clearly.

We thank the review for this suggestion. We have added schematics in the revised figures (Figure 2n and Figure 4o).

6. Line 222-223: The authors chose to focus on cluster 4, citing that cluster 10 has limited WT cells for comparison. However, from Figure 2e, it would appear that the percentage of WT vs T1 samples in C4 and C10 are similar. Could the authors include the number of cells overlapping between the WT and T1 samples in C4 and C10, and elaborate on the reasons for choosing C4?

When performing the differential analysis for the scRNA-seq data comparing WT and T1, we aimed for a minimum cell number of 50 per cluster. However, in WT C10, the cell number is only 38 (Figure R11), comprising merely 1.7% of the whole WT nephron population. As this number falls below the established threshold, we decided to exclude C10 from our differential analysis. We have made this point clearer in the revised manuscript (Line 243-244).

Figure R11. The number of cells grouped into *Enl*-WT and *Enl*-T1 nephron clusters.

7. Figure 2k: Cluster 7 is classified as NP2, and *Wnt4* is expressed in NP2s. However, *Wnt4* is not expressed in C7 like it is expressed in the other 2 NP2 clusters 1 and 5 (Figure 2k). Could the authors elaborate on why this might be? Is C7 different in some way?

The heatmap (now in revised Figure 2j) represents the log2 fold change of individual genes between T1 and WT in indicated clusters, rather than their absolute expression. This clarification has been added to the revised Figure 2 legend (Line 1142-1143). Analysis of *Wnt4* expression levels across different clusters shows that Cluster 7 has *Wnt4* expression levels comparable to Clusters 1 and 5 in WT kidneys (Figure R12). Comparing *Wnt4* between WT and T1 within each cluster reveals a consistent upregulation of *Wnt4* across *Enl*-T1 cluster 1/5/7, though the increase is less prominent in cluster 7 (FC = 1.10) compared to cluster 1 (FC = 1.52) and 5 (FC = 1.50). We have adjusted the color scale of the updated heatmap (Figure 2j) to more accurately reflect these fold changes.

Figure R12. Violin plot showing the gene expression of *Six2* and *Wnt4* for each cell type in *Enl*-WT and *Enl*-T1 nephron.

8. Line 464, Figure 5j: There is a lot of ear morphogenesis and multicellular organism development pathways also – could the authors explain why this might be?

The functional analysis is evaluated based on the degree of overlap between the provided gene list and the gene signature associated with specific biological functions in the GO term database. In some cases, biological processes or pathways across different contexts may share common key genes, such as developmental-related pathways in various tissue types. In our case, the genes involved in the ear morphogenesis pathway (*Eya1*, *Fgfr1*, *Gas1*, *Osr1*, and *Six2*) as shown in Figure 5j, and those involved in the multicellular organism development pathway (*Cited1*, *Eya1*, *Meis2*, *Uncx*, *Fgfr1*, *Meox2*, *Nnat*, *Nrp1*, *Robo2*, and *Six2*), are largely correlated with those involved in nephrogenesis. This may be the main reason why these pathways show up in the GO analysis.

9. Line 621, 613: The dislodging of condensates in the ENL mutants upon TDI treatment is a fascinating finding. Although, I am curious how does that explain the increase in size or decrease in number of the condensates in Enl mutant? Could the authors elaborate on that?

There could be a few reasons to explain the increase in size and decrease in number of the condensates upon TDI treatment. As the condensates are displaced from chromatin, they are no longer constrained by chromatin and may therefore undergo fusion events, which can lead to an increase in size and a decrease in number. In addition, it is worth noting that besides enriching in the condensates, mutant ENL proteins also bind to other genomic regions without forming detectable condensates, as evident in our ChIP-seq experiments (Figure 7f). The inhibitor treatment decreases the overall chromatin occupancy of mutant ENL, which presumably led to an increase in the amount of 'free' mutant ENL proteins in the nucleus. These proteins can be integrated into the condensates, leading to a large size. Similar observations were noted with other chromatin regulators such as ARID1A and NUP98-HOXA9, where their condensates become bigger when the DNA binding domain is disrupted^{23,24}.

10. Overall, I would urge the authors to use caution when interpreting results from pseudotime analysis. Without ground truth lineage tracing data to support these inferences, the results must be reported with care.

This is a good point. We have revised the manuscript to tone down the interpretation of the differentiation trajectory analysis and also underscore the necessity for lineage tracing data to verify the hypotheses derived from this analysis (Lines 229-238, 457-463, 533-534).

Minor comments:

1. Line 287-289: Could the authors include a p-value for the GSVA correlation?

We have added a p-value for the GSVA correlation (Figure 2l).

2. Paragraph from line 261-271 seems out of place and could be confusing as to why it is there. Perhaps it could be moved to the end of that section, after line 295.

Thanks for the suggestion. We have moved this section to the end of the section to enhance the logic flow of the manuscript and revised the manuscript accordingly (Line 304-317).

3. Line 453 – should it more specifically be “T1-ab lost DARs (vs. NP1)?

We have modified the phrase from 'T1-ab lost DARs (vs. NP1)' to 'T1-ab lost DARs when compared with NP1 cells' for enhanced clarity in the revised manuscript (Line 480-481)

4. Line 571, Figure 6m: p-values appear to be missing in the table?

The table in Figure 6m has been moved to Figure S9k. This data delineates the presence or absence of motif sequences for specific Hox TFs within the corresponding T1-gained DARs, thus there was no P-

values for this analysis. We provided below a comprehensive list of all Hox motif sequences identified within these six DARs.

T1-gained DAR index						
	1	2	3	4	5	6
Hoxa9	chr14:28,441,519-28,441,528 (TTTTATGAGA,-) chr14:28,441,634-28,441,643 (AGCTATAAAA,+)	chr14:28475486-28475495 (ACCAATGAGA,+)			chr14:28613432-28613441 (ACCCATAACA,+)	chr14:28622338-28622347 (CTTTAGTGCT,-) chr14:28,622,545-28,622,554 (AACAAATAAG,+) chr14:28622553-28622562 (AGTAATAAAA,+)
Hoxc9						chr14:28622277-28622286 (GGATTTATGTCT,-)
Hoxa11	chr14:28,441,519-28,441,528 (TTTTATGAGA,-) chr14:28441634-28441643 (AGCTATAAAA,-)				chr14:28613408-28613417 (TTTTATTATG,+)	chr14:28622328-28622337 (TTTTATGTGT,+) chr14:28622553-28622562 (AGTAATAAAA,-)
Hoxd11	chr14:28441519-28441528 (TTTTATGAGA,-) chr14:28441634-28441643 (AGCTATAAAA,+)	chr14:28475343-28475352 (CTTACAGCT,-)			chr14:28613408-28613417 (TTTTATTATG,-)	chr14:28622328-28622337 (TTTTATGTGT,-) chr14:28622553-28622562 (AGTAATAAAA,+)
Hoxd12	chr14:28441633-28441644 (AAGCTATAAAA,+)				chr14:28613407-28613418 (ATTTTATTATGG,-)	chr14:28622337-28622348 (TCTTTAGTGCTG,-) chr14:28622341-28622352 (TAGTGCTGAAAT,+) chr14:28622552-28622563 (AAGTAATAAAC,+)

Reviewer #5 (Remarks to the Author):

I co-reviewed this manuscript with one of the reviewers who provided the listed reports as part of the Nature Communications initiative to facilitate training in peer review and appropriate recognition for co-reviewers.

References

1. Perlman, E. J. *et al.* MLLT1 YEATS domain mutations in clinically distinctive Favourable Histology Wilms tumours. *Nat Commun* 6, (2015).
2. Kobayashi, A. *et al.* Identification of a Multipotent Self-Renewing Stromal Progenitor Population during Mammalian Kidney Organogenesis. *Stem Cell Reports* 3, 650-662 (2014).
3. Stark, K., Vainio, S., Vassileva, G. & McMahon, A. P. Epithelial transformation of metanephric mesenchyme in the developing kidney regulated by Wnt-4. *Nature* 372, 679683 (1994).
4. Drake, K. A., Adam, M., Mahoney, R. & Potter, S. S. Disruption of Hox9,10,11 function results in cellular level lineage infidelity in the kidney. *Sci Rep* 8, 6306 (2018).
5. Mugford, J. W., Sipilä, P., Kobayashi, A., Behringer, R. R. & McMahon, A. P. Hoxd11 specifies a program of metanephric kidney development within the intermediate mesoderm of the mouse embryo. *Dev Biol* 319, 396-405 (2008).
6. Wellik, D. M., Hawkes, P. J. & Capecchi, M. R. *Hox11* paralogous genes are essential for metanephric kidney induction. *Genes Dev* 16, 1423-1432 (2002).
7. Song, L. *et al.* Hotspot mutations in the structured ENL YEATS domain link aberrant transcriptional condensates and cancer. *Mol Cell* 82, 4080-4098.e12 (2022).
8. Wan, L. *et al.* Impaired cell fate through gain-of-function mutations in a chromatin reader. *Nature* 577, 121-126 (2020).
9. Liu, Y. *et al.* Small-Molecule Inhibition of the Acyl-Lysine Reader ENL as a Strategy against Acute Myeloid Leukemia. *Cancer Discov* 12, 2684-2709 (2022).
10. Hilliard, S., Tortelote, G., Liu, H., Chen, C.-H. & El-Dahr, S. S. Single-Cell Chromatin and Gene-Regulatory Dynamics of Mouse Nephron Progenitors. *Journal of the American Society of Nephrology* 33, 1308-1322 (2022).
11. Naganuma, H. *et al.* Molecular detection of maturation stages in the developing kidney. *Dev Biol* 470, 62-73 (2021).
12. Wang, G. *et al.* Spatial dynamic metabolomics identifies metabolic cell fate trajectories in human kidney differentiation. *Cell Stem Cell* 29, 1580-1593.e7 (2022).
13. Urbach, A. *et al.* Lin28 sustains early renal progenitors and induces Wilms tumor. *Genes Dev* 28, 971-982 (2014).

14. Kruber, P. *et al.* Loss or oncogenic mutation of DROSHA impairs kidney development and function, but is not sufficient for Wilms tumor formation. *Int J Cancer* 144, 1391-1400 (2019).
15. Li, C.-M. *et al.* CTNNB1 Mutations and Overexpression of Wnt/13-Catenin Target Genes in WT1-Mutant Wilms' Tumors. *Am J Pathol* 165, 1943-1953 (2004).
16. Hastie, N. D. Wilms' tumour 1 (WT1) in development, homeostasis and disease. *Development (Cambridge)* 144, 2862-2872 (2017).
17. McMahon, A. P. Development of the Mammalian Kidney. in *Current Topics in Developmental Biology* vol. 117 31-64 (Academic Press Inc., 2016).
18. Paris, N. D., Coles, G. L. & Ackerman, K. G. Wt1 and 13-catenin cooperatively regulate diaphragm development in the mouse. *Dev Biol* 407, 40-56 (2015).
19. Drake, K. A. *et al.* Stromal 13-catenin activation impacts nephron progenitor differentiation in the developing kidney and may contribute to Wilms tumor. *Development (Cambridge)* 147, (2020).
20. Ramalingam, H. *et al.* Disparate levels of beta-catenin activity determine nephron progenitor cell fate. *Dev Biol* 440, 13-21 (2018).
21. Huang, L. *et al.* Nephron Progenitor But Not Stromal Progenitor Cells Give Rise to Wilms Tumors in Mouse Models with 13-Catenin Activation or Wt1 Ablation and Igf2 Upregulation. *Neoplasia (United States)* 18, 71-81 (2016).
22. Pietilä, I. *et al.* Secreted Wnt antagonist Dickkopf-1 controls kidney papilla development coordinated by Wnt-7b signalling. *Dev Biol* 353, 50-60 (2011).
23. Chandra, B. *et al.* Phase Separation Mediates NUP98 Fusion Oncoprotein Leukemic Transformation. *Cancer Discov* 12, 1152-1169 (2022).
24. Patil, A. *et al.* A disordered region controls cBAF activity via condensation and partner recruitment. *Cell* 186, 4936-4955.e26 (2023).

REVIEWERS' COMMENTS

Reviewer #1 (Remarks to the Author):

The authors have significantly improved the manuscript and addressed most of my concerns.

The rebuttal letter is well written, but there are many figures only for the reviewers. I suggest that at least some of them be included in the manuscript as supplementary figures. In particular, Figure R4 will be informative for the readers, even though prolonged treatment with TDI-11055 did not rescue the perinatal lethality. The authors could mention this result in the text as one of the limitations of the study.

Reviewer #2 (Remarks to the Author):

The authors have made a thorough response addressing key concerns raised by the reviewers. adding additional clarity and in the process revealing an interesting area (difference between nephron/interstitial and nephron only modification of Enl) for a future study.

I would suggest removing this somewhat vague statement from the abstract: "Furthermore, the mutant ENL might modulate stroma-nephron interactions via paracrine Wnt signaling." This one of many "mights" in the data. The abstract should focus on conclusive statements.

Reviewer #3 (Remarks to the Author):

In the revised version of the manuscript, authors have added quite some new data to address my original concerns. These additional data and revisions have substantially strengthened the manuscript to support their conclusions.

Reviewer #4 (Remarks to the Author):

The authors have satisfactorily responded to our comments. They made the wording/phrasing changes to interpret their results more carefully, and included some new analysis in their reviewer only figures to address our concerns. They have also tested increased dosage of TDI (200mg/kg) to confirm that the rescue effects of their drug plateau at their original dosage (100mg/kg). It might be useful to include this dosage result in the main manuscript also, even if just a line of text in the results with no data shown.

Where we asked for more direct evidence to show Wnt5a interaction with NPC receptors, the authors proposed future research to examine that in ex-vivo system, but they have revised the wording in the revised manuscript to address this, which is acceptable for the scope of this paper.

Reviewer #5 (Remarks to the Author):

Responses to Reviewers' comments

NCOMMS-23-52570A

"Single-Cell multiomics reveals ENL mutation perturbs kidney developmental trajectory by rewiring gene regulatory landscape "

Overall summary of the revision

We are pleased to learn that the reviewers found their previous comments well addressed and that the manuscript has been significantly improved. We have further revised the manuscript to address the remaining minor comments as detailed below.

Reviewer #1 (Remarks to the Author):

The authors have significantly improved the manuscript and addressed most of my concerns. The rebuttal letter is well written, but there are many figures only for the reviewers. I suggest that at least some of them be included in the manuscript as supplementary figures. In particular, Figure R4 will be informative for the readers, even though prolonged treatment with TDI-11055 did not rescue the perinatal lethality. The authors could mention this result in the text as one of the limitations of the study.

We are pleased to hear that the reviewer found our manuscript significantly improved and that our revisions addressed most of their concerns. We agree that Figure R4 would be informative for the readers. Therefore, we have added it as Supplementary Figure 12 in the revised manuscript and discussed the results in the revised text (Line 727-735, 860-864).

Reviewer #2 (Remarks to the Author):

The authors have made a thorough response addressing key concerns raised by the reviewers. adding additional clarity and in the process revealing an interesting area (difference between nephron/interstitial and nephron only modification of Enl) for a future study. I would suggest removing this somewhat vague statement from the abstract: "Furthermore, the mutant ENL might modulate stroma-nephron interactions via paracrine Wnt signaling." This one of many "mights" in the data. The abstract should focus on conclusive statements.

We are glad that the reviewer found our revisions thorough and that they revealed interesting areas for future studies. As suggested, we have revised the abstract to focus on conclusive statements.

Reviewer #3 (Remarks to the Author):

In the revised version of the manuscript, authors have added quite some new data to address my original concerns. These additional data and revisions have substantially strengthened the manuscript to support their conclusions.

We are pleased to hear that the reviewer found the additional data and revisions satisfactory and that they have substantially strengthened the manuscript.

Reviewer #4 (Remarks to the Author):

The authors have satisfactorily responded to our comments. They made the wording/phrasing changes to interpret their results more carefully, and included some new analysis in their reviewer only figures to address our concerns. They have also tested increased dosage of TDI (200mg/kg) to confirm that the rescue effects of their drug plateau at their original dosage (100mg/kg). It might be useful to include this dosage result in the main manuscript also, even if just a line of text in the results with no data shown. Where we asked for more direct evidence to show Wnt5a interaction with NPC receptors, the authors proposed future research to examine that in ex-vivo system, but they have revised the wording in the revised manuscript to address this, which is acceptable for the scope of this paper.

We are pleased to hear that the reviewer found the revisions satisfactory. As suggested, we have now included the results with increased TDI dosage in Supplementary Figure 12 and address the data in the main text (Line 723-727).

Reviewer #5 (Remarks to the Author):
